# BCE vs. CE in Deep Feature Learning

## Abstract

When training classification models, it expects that the leaned features are compact within classes, and can well separate different classes. As a dominant loss function to train classification models, the minimization of CE (Cross-entropy) loss can maximize the compactness and distinctiveness, i.e., reaching neural collapse. The recently published works show that BCE (Binary CE) loss performs also well in multi-class tasks. In this paper, we compare BCE and CE in the context of deep feature learning. For the first time, we prove that BCE can also maximize the intra-class compactness and inter-class distinctiveness when reaching its minimum, i.e., leading to neural collapse. We point out that CE measures the relative values of decision scores in the model training, implicitly enhancing the feature properties by classifying samples one-by-one. In contrast, BCE measures the absolute values of decision scores and adjust the positive/negative decision scores across all samples to uniform high/low levels. Meanwhile, the classifier bias in BCE presents a substantial constraint on the samples' decision scores. Thereby, BCE explicitly enhances the feature properties in the training. The experimental results are aligned with above analysis, and show that BCE consistently and significantly improve the classification performance and leads to better compactness and distinctiveness among sample features.

## 1 Introduction

Cross-entropy (CE) loss is the most commonly used loss function for classifications and feature learning. In a multi-class classification with $K$ categories, for any sample $\boldsymbol{X}^{(k)}$ from category $k$, a model $\mathcal{M}$ first extracts its feature $\boldsymbol{h}^{(k)} = \mathcal{M}(\boldsymbol{X}^{(k)}) \in \mathbb{R}^d$, which is output from the penultimate hidden layer in deep model. Then a linear classifier with weight $\boldsymbol{W} = [\boldsymbol{w}_1, \cdots, \boldsymbol{w}_K]^T \in \mathbb{R}^{K \times d}$ and bias $\boldsymbol{b} = [b_1, \cdots, b_K]^T \in \mathbb{R}^K$ transforms the feature into $K$ logits/decision scores, $\{\boldsymbol{w}_j^T \boldsymbol{h}^{(k)} - b_j\}_{j=1}^K$, which are finally converted into predicted probabilities by Softmax, and computed the loss using cross-entropy,

$$\mathcal{L}_{\mathrm{ce}}\big(\boldsymbol{z}^{(k)}\big) = -\boldsymbol{y}_k^T \cdot \log\Big(\mathrm{Softmax}\big(\boldsymbol{z}^{(k)}\big)\Big) = \log\left(1 + \sum_{\substack{\ell=1 \\ \ell \neq k}}^K \frac{\mathrm{e}^{\boldsymbol{w}_\ell^T \boldsymbol{h}^{(k)} - b_\ell}}{\mathrm{e}^{\boldsymbol{w}_k^T \boldsymbol{h}^{(k)} - b_k}}\right), \tag{1}$$

where $\boldsymbol{z}^{(k)} = \boldsymbol{W}\boldsymbol{h}^{(k)} - \boldsymbol{b}$ and $\boldsymbol{y}_k$ is the one-hot label, i.e., the vector with one only in the $k$th entry.

In the multi-class classification, binary CE (BCE) loss is deduced by decomposing the task into $K$ binary tasks and predicting whether the sample $\boldsymbol{X}^{(k)}$ belongs to the $j$th category, for $\forall j \in [K]$,

$$\mathcal{L}_{\mathrm{bce}}\big(\boldsymbol{z}^{(k)}\big) = -\boldsymbol{y}_k^T \cdot \log\Big(\mathrm{Sigmoid}\big(\boldsymbol{z}^{(k)}\big)\Big) - (\mathbf{1} - \boldsymbol{y}_k)^T \cdot \log\Big(\mathbf{1} - \mathrm{Sigmoid}\big(\boldsymbol{z}^{(k)}\big)\Big)$$

$$= \log\Big(1 + \mathrm{e}^{-\boldsymbol{w}_k^T \boldsymbol{h}^{(k)} + b_k}\Big) + \sum_{\substack{j=1 \\ j \neq k}}^K \log\Big(1 + \mathrm{e}^{\boldsymbol{w}_j^T \boldsymbol{h}^{(k)} - b_j}\Big), \tag{2}$$

which has been widely used in the multi-label classification (Kobayashi, 2023) and attracted increasing attentions in the multi-class classification (Beyer et al., 2020; Wightman et al., 2021; Touvron et al., 2022; Fang et al., 2023; Wen et al., 2022; Zhou et al., 2023).

The pre-trained classification models can be used as feature extractors for downstream tasks that request well intra-class compactness and inter-class distinctiveness across the sample features, such

as person re-identification (He et al., 2021), object tracking (Cai et al., 2023), image segmentation (Guo et al., 2022), and facial recognition (Wen et al., 2022), etc. For CE, a remarkable theoretical result is that when it reaches its minimum, both the compactness and distinctiveness on the training samples will be maximized, which refers to neural collapse found by Papyan et al. (2020). Neural collapse gives peace of mind in training classification models by using CE, and it was extended to the losses satisfying contrastive property by Zhu et al. (2021) and Zhou et al. (2022), including CE, focal loss, and label smoothing loss. However, though BCE in Eq. (2) is a combination of multiple CE, it does not satisfy the contrastive property due to its classifier bias, and whether BCE can lead to neural collapse, or not, is not answered.

Besides that, in the practical training of classification models, the classifier vectors $\{\boldsymbol{w}_k\}_{k=1}^{K}$ play the role of proxy for each category (Wen et al., 2022). Intuitively, when the distances between the sample features and their class proxy are closer, or the *positive* decision scores between them are larger, it usually leads to better intra-class compactness. Similarly, when the distances between sample features and the proxy of different classes are farther, or the *negative* decision scores between them are smaller, it could results in better inter-class distinctiveness. However, according to Eq. (1), CE measures the *relative* value between the exponential positive and negative decision scores using Softmax and logarithmic functions, to pursue that the positive decision score is greater than all its negative ones for each sample, making it unable to explicitly and directly enhance the intra-class compactness and inter-class distinctiveness across samples. In contrast, BCE in Eq. (2) respectively measures the *absolute* values of the exponential positive decision score and the exponential negative ones using Sigmoid and logarithmic functions, which makes it could explicitly and directly enhance the compactness and distinctiveness of features in the training.

In this paper, we compare BCE and CE in deep feature learning. We primarily address two questions: **Q1**. Can BCE result in the neural collapse, i.e., maximizing the compactness and distinctiveness in theoretical? **Q2**. In practical training of classification models, does BCE perform better than CE in terms of the feature compactness and distinctiveness? Our contributions are summarized as follows.

(1) We provide the first theoretical proof that BCE can also lead to the neural collapse, i.e., maximizing the compactness and distinctiveness, even when the loss does not satisfy the contrastive property, broadening the range of losses that can lead to neural collapse.

(2) We find that BCE performs better than CE in enhancement of intra-class compactness and inter-class distinctiveness across sample features, and, BCE can explicitly enhance the feature properties, while CE only implicitly enhance them.

(3) We point out that when training models with BCE, the classifier bias plays a substantial role in enhancing the feature properties, while in the training with CE, it almost does not work.

(4) We conduct extensive experiments, and find that, compared to CE, BCE can more quickly lead to the neural collapse on the training dataset and achieves better feature compactness and distinctiveness, resulting in higher classification performance on the test dataset.

## 2    RELATED WORKS

### 2.1    CE VS. BCE

The CE loss is the most popular loss used in the multi-class classification and feature learning, which has been evolved into many variants in different scenarios, such as focal loss (Lin et al., 2017), label smoothing loss (Szegedy et al., 2016), normalized Softmax loss (Wang et al., 2017), and marginal Softmax loss (Liu et al., 2016), etc. The classification models are often applied to the downstream tasks, such as image segmentation (Guo et al., 2022), person re-identification (He et al., 2021), object tracking (Cai et al., 2023), etc., which request well intra-class compactness and inter-class distinctiveness among the sample features. In the multi-class classification task, the BCE loss can be deduced by decomposing the task into $K$ binary tasks and adding the $K$ binary cross-entropy losses, which has been widely applied in the multi-label classification (Kobayashi, 2023).

The CE and BCE losses are expected to train the models to fit the sample distribution in the multi-class and multi-label classifications. When Wightman et al. (2021) applied BCE to the training of ResNets for a multi-class task, they considered that this loss is consistent with Mixup (Zhang et al., 2018) and CutMix (Yun et al., 2019) augmentations, which mix multiple objects from different samples into one sample. DeiT III (Touvron et al., 2022) adopted this approach and achieved a

significant improvement in the multi-class task on ImageNet-1K by using the BCE loss. Currently, though the CE loss dominates the training of multi-class and feature learning models, the BCE loss is also gaining more attention and is increasingly being applied in these fields (Fang et al., 2023; Wang et al., 2023; Xu et al., 2023; Mehta & Rastegari, 2023; Chun, 2024; Hao et al., 2024). However, none of these works reveals the essential advantages of BCE over CE.

## 2.2 NEURAL COLLAPSE

The neural collapse was first found by Papyan et al. (2020), which refers to four properties about the sample features $\{\boldsymbol{h}_i^{(k)}\}$ and the classifier vectors $\{\boldsymbol{w}_k\}$ at the terminal phase of training.

- **NC1**, within-class variability collapse. Each feature $\boldsymbol{h}_i^{(k)}$ collapse to its class center $\bar{\boldsymbol{h}}^{(k)} = \frac{1}{n_k}\sum_{i'=1}^{n_k} \boldsymbol{h}_{i'}^{(k)}$, indicating the *maximal intra-class compactness*

- **NC2**, convergence to simplex equiangular tight frame. The set of class centers $\{\bar{\boldsymbol{h}}^{(k)}\}_{k=1}^K$ form a simplex equiangular tight frame (ETF), with equal and maximized cosine distance between every pair of feature means, i.e., the *maximal inter-class distinctiveness*.

- **NC3**, convergence to self-duality. The class center $\bar{\boldsymbol{h}}^{(k)}$ is ideally aligned with the classifier vector $\boldsymbol{w}_k, \forall k \in [K]$.

- **NC4**, simplification to nearest class center. The classifier is equivalent to a nearest class center decision.

The current works about neural collapse (Kothapalli, 2023) are focused on the CE loss (Lu & Steinerberger, 2022; Graf et al., 2021; Zhu et al., 2021) and mean squared error (MSE) loss (Han et al., 2022; Tirer & Bruna, 2022). It has been proved that the models will fall to the neural collapse when the loss reaches its minimum. A comprehensive analysis (Zhou et al., 2022) for various losses, including CE loss, focal loss (Lin et al., 2017), and label smoothing loss (Szegedy et al., 2016), shows that these losses perform equally as any global minimum point of the loss satisfies the neural collapse. The neural collapse has also been investigated in the imbalanced datasets (Fang et al., 2021; Yang et al., 2022; Wang et al., 2024), out-of-distribution data (Ammar et al., 2024), and models with fixed classifiers (Yang et al., 2022; Kim & Kim, 2024). All these studies are conducted using CE or MSE losses; and whether BCE can lead to neural collapse remains unexplored.

## 3 MAIN RESULTS

In this section, we first theoretically prove that BCE can maximize the compactness and distinctiveness when reaching its minimums (**Q1**). Then, through in-depth analyzing the decision scores in the training with BCE and CE, we explain that BCE can better enhance the compactness and distinctiveness of sample features in practical training (**Q2**).

## 3.1 PRELIMINARY

Let $\mathcal{D} = \bigcup_{k=1}^K \bigcup_{i=1}^{n_k} \{\boldsymbol{X}_i^{(k)}\}$ be a sample set, where $\boldsymbol{X}_i^{(k)}$ denotes the $i$th sample of category $k$, $n_k$ is the number of samples in this category, and $\boldsymbol{h}_i^{(k)} = \mathcal{M}(\boldsymbol{X}_i^{(k)})$. In classification tasks, a linear classifier with vectors $\boldsymbol{W} = [\boldsymbol{w}_1, \cdots, \boldsymbol{w}_K]^T \in \mathbb{R}^{K \times d}$ and bias $\boldsymbol{b} = [b_1, \cdots, b_K]^T \in \mathbb{R}^K$ predicts the category for each sample according to its feature. For the well predication results, the CE or BCE loss is applied to tune the parameters of the model $\mathcal{M}$ and classifier.

Following the previous works (Fang et al., 2021; Lu & Steinerberger, 2022; Graf et al., 2021; Zhu et al., 2021; Han et al., 2022; Tirer & Bruna, 2022) for neural collapse, we compare CE and BCE in training of unconstrained model or layer-peeled model in this paper, i.e, treating the features $\bigcup_{k=1}^K \{\boldsymbol{h}_i^{(k)}\}_{i=1}^{n_k}$, classifier vectors $\{\boldsymbol{w}_k\}_{k=1}^K$, and classifier bias $\{b_k\}_{k=1}^K$ as free variables, without considering the sophisticated structure or the parameters of the model $\mathcal{M}$. Then, taking the regularization terms on the variables, the CE or BCE loss in the training is

$$f_\mu(\boldsymbol{W}, \boldsymbol{H}, \boldsymbol{b}) := \frac{1}{nK}\sum_{k=1}^K \sum_{i=1}^{n_k} \mathcal{L}_\mu\big(\boldsymbol{W}\boldsymbol{h}_i^{(k)} - \boldsymbol{b}\big) + \frac{\lambda_{\boldsymbol{W}}}{2}\|\boldsymbol{W}\|_F^2 + \frac{\lambda_{\boldsymbol{H}}}{2}\|\boldsymbol{H}\|_F^2 + \frac{\lambda_{\boldsymbol{b}}}{2}\|\boldsymbol{b}\|_2^2, \quad (3)$$

where $\mathcal{L}_\mu$ is presented in Eqs. (1-2), $\mu \in \{\text{ce}, \text{bce}\}$,

$$\boldsymbol{W} = \begin{bmatrix} \boldsymbol{w}_1, \boldsymbol{w}_2, \cdots, \boldsymbol{w}_K \end{bmatrix}^T \in \mathbb{R}^{K \times d}, \tag{4}$$

$$\boldsymbol{H} = \begin{bmatrix} \boldsymbol{h}_1^{(1)}, \boldsymbol{h}_2^{(1)}, \cdots, \boldsymbol{h}_{n_1}^{(1)}, \cdots, \boldsymbol{h}_1^{(K)}, \boldsymbol{h}_2^{(K)}, \cdots, \boldsymbol{h}_{n_K}^{(K)} \end{bmatrix} \in \mathbb{R}^{d \times (\sum_{k=1}^K n_k)}, \tag{5}$$

$$\boldsymbol{b} = [b_1, b_2, \cdots, b_K]^T \in \mathbb{R}^K, \tag{6}$$

and $\lambda_{\boldsymbol{W}}, \lambda_{\boldsymbol{H}} > 0, \lambda_{\boldsymbol{b}} \geq 0$ are weight decay parameters for the regularization terms.

## 3.2 NEURAL COLLAPSE WITH CE AND BCE LOSSES

On the balanced dataset, i.e., $n = n_k, \forall k \in [K]$, Zhu et al. (2021) proved that the CE loss can result in neural collapse, and in Theorem 1, Zhou et al. (2022) extended the proof to the losses satisfying the contrastive property (see Definition S-1 in supplementary), such as focal loss and label smoothing loss. Though BCE loss is a combination of CE loss, it fails to satisfy the contrastive property, as that the classifier bias parameters present substantial constraint within its components. Despite that, we find that the BCE loss can also result in the neural collapse, i.e., Theorem 2. The primary difference between BCE and CE losses lies in the bias parameter $\boldsymbol{b}$ of their classifiers.

**Theorem 1** *(Zhou et al., 2022) Assume that the feature dimension $d$ is larger than the category number $K$, i.e., $d \geq K - 1$, and $\mathcal{L}_\mu$ is satisfying the contrastive property. Then any global minimizer $(\boldsymbol{W}^\star, \boldsymbol{H}^\star, \boldsymbol{b}^\star)$ of $f_\mu(\boldsymbol{W}, \boldsymbol{H}, \boldsymbol{b})$ defined using $\mathcal{L}_\mu$ with Eq. (3) obeys the following properties,*

$$\|\boldsymbol{w}^\star\| = \|\boldsymbol{w}_1^\star\| = \|\boldsymbol{w}_2^\star\| = \cdots = \|\boldsymbol{w}_K^\star\|, \tag{7}$$

$$\boldsymbol{h}_i^{(k)\star} = \sqrt{\frac{\lambda_{\boldsymbol{W}}}{n\lambda_{\boldsymbol{H}}}} \boldsymbol{w}_k^\star, \ \forall \, k \in [K], \ i \in [n], \tag{8}$$

$$\tilde{\boldsymbol{h}}_i^\star := \frac{1}{K} \sum_{k=1}^K \boldsymbol{h}_i^{(k)\star} = \boldsymbol{0}, \ \forall \, i \in [n], \tag{9}$$

$$\boldsymbol{b}^\star = b^\star \boldsymbol{1}_K, \tag{10}$$

*where either $b^\star = 0$ or $\lambda_{\boldsymbol{b}} = 0$. The matrix $\boldsymbol{W}^{\star T}$ forms a $K$-simplex ETF in the sense that*

$$\frac{1}{\|\boldsymbol{w}^\star\|_2^2} \boldsymbol{W}^{\star T} \boldsymbol{W}^\star = \frac{K}{K-1} \Big( \boldsymbol{I}_K - \frac{1}{K} \boldsymbol{1}_K \boldsymbol{1}_K^T \Big), \tag{11}$$

*where $\boldsymbol{I}_K \in \mathbb{R}^{K \times K}$ denotes the identity matrix, and $\boldsymbol{1}_K \in \mathbb{R}^K$ denotes the all ones vector.* ∎

**Theorem 2** *Assume that the feature dimension $d$ is larger than the category number $K$, i.e., $d \geq K - 1$. Then any global minimizer $(\boldsymbol{W}^\star, \boldsymbol{H}^\star, \boldsymbol{b}^\star)$ of $f_{\text{bce}}(\boldsymbol{W}, \boldsymbol{H}, \boldsymbol{b})$ defined using $\mathcal{L}_{\text{bce}}$ with Eq. (3) obeys the properties (7) - (11), where $b^\star$ is the solution of equation*

$$0 = -\frac{K-1}{K \left( 1 + \exp \left( b + \sqrt{\frac{\lambda_{\boldsymbol{W}}}{n\lambda_{\boldsymbol{H}}}} \frac{\rho}{K(K-1)} \right) \right)} + \frac{1}{K \left( 1 + \exp \left( \sqrt{\frac{\lambda_{\boldsymbol{W}}}{n\lambda_{\boldsymbol{H}}}} \frac{\rho}{K} - b \right) \right)} + \lambda_{\boldsymbol{b}} b, \tag{12}$$

*and $\rho = \|\boldsymbol{W}^\star\|_F^2$ is the squared Frobenius norm of $\boldsymbol{W}^\star$.*

**Proof** *The detailed proof is presented in the supplementary, i.e., Theorem S-4, which similar to that of Zhu et al. (2021); Zhou et al. (2022); Lu & Steinerberger (2022), studies lower bounds for the BCE loss in Eq. (3) and finds the conditions for achieving the lower bounds.* ∎

Theorem 2 significantly broadens the range of losses that can lead to neural collapse, i.e., the contrastive property (Zhou et al., 2022) is not necessarily satisfied.

**The decision scores.** According to Theorems 1 and 2, when training a classification model with CE or BCE losses, if the loss reaches its minimum and results in the neural collapse, the sample feature $\boldsymbol{h}_i^{(k)}$ will converge to its class center $\bar{\boldsymbol{h}}^{(k)} = \frac{1}{n} \sum_{i=1}^n \boldsymbol{h}_i^{(k)}$, indicating the maximum intra-class compactness. Furthermore, the class center $\bar{\boldsymbol{h}}^{(k)}$ becomes a multiple of the corresponding classifier vector $\boldsymbol{w}_k$, and the $K$ classifier vectors $\{\boldsymbol{w}_k\}_{k=1}^K$ will form an ETF, indicating the maximum inter-class distinctiveness. In addition, the positive and negative decision scores without the biases of all

samples will respectively converge to fixed values, i.e., for $\forall j \neq k \in [K],\ i \in [n]$,

$$s_{\text{pos}}^{(kk,i)} = \boldsymbol{w}_k^T \boldsymbol{h}_i^{(k)} \to \sqrt{\frac{\lambda_{\boldsymbol{W}}}{n\lambda_{\boldsymbol{H}}}} \frac{\rho}{K} \quad \text{and} \quad s_{\text{neg}}^{(jk,i)} = \boldsymbol{w}_j^T \boldsymbol{h}_i^{(k)} \to -\sqrt{\frac{\lambda_{\boldsymbol{W}}}{n\lambda_{\boldsymbol{H}}}} \frac{\rho}{K(K-1)}. \qquad (13)$$

**The classifier bias.** Comparing Theorems 1 and 2, one can find that the primary difference between CE and BCE losses lies in their classifier bias parameter. According to Theorem 1, when $\lambda_{\boldsymbol{b}} > 0$, the minimum point of CE loss satisfies $\boldsymbol{b} = \boldsymbol{0}$, separating the final positive and negative scores in Eq. (13); when $\lambda_{\boldsymbol{b}} = 0$, any point that satisfies properties (7) - (11) and $\boldsymbol{b} = b^\star \boldsymbol{1}$ is a minimum point of CE loss, which implies that the minimum points of CE loss form a ridge line in term of $\boldsymbol{b}$. In contrast, the classifier bias $\boldsymbol{b}$ at the minimum points of BCE loss satisfy Eq. (12) whenever $\lambda_{\boldsymbol{b}} = 0$ or not. According to Lemma 5 in the supplementary, Eq. (12) has only one solution, indicating that the BCE loss has only one minimum point in term of $\boldsymbol{b}$. This optimal classifier bias $\boldsymbol{b} = b^\star \boldsymbol{1}$ will separate the positive and negative decision scores if it satisfies the Eq. (165) (see Lemma 6 in supplementary for details). Both of the optimal points of CE and BCE losses are associated with the unified classifier bias $\boldsymbol{b} = b^\star \boldsymbol{1}$, which is aligned with the unified bias integrated loss designed by Wen et al. (2022) and Zhou et al. (2023) for facial recognition.

## 3.3 THE DECISION SCORES IN TRAINING WITH BCE AND CE

Though the intra-class compactness and inter-class distinctiveness of sample features can be theoretically maximized by both CE and BCE, the two losses perform very different in practical training of classification models. We here compare the decision scores in the model training by using BCE and CE, to explain their difference in enhancing the feature properties.

**A geometric comparison for CE and BCE.** In practical training with CE or BCE, to minimize the loss, it is desirable for their exponential function variables to be as small as possible, and less than zero at least. For CE in Eq. (1), it is desirable that, for $\forall \ell \neq k \in [K]$,

$$\underbrace{\boldsymbol{w}_k \boldsymbol{h}^{(k)} - b_k}_{\text{positive decision score}} > \underbrace{\boldsymbol{w}_\ell \boldsymbol{h}^{(k)} - b_\ell}_{\text{negative decision score}}, \qquad (14)$$

while, for BCE in Eq. (2), it is desirable that, for $\forall j \neq k \in [K]$,

$$\boldsymbol{w}_k \boldsymbol{h}^{(k)} - b_k > 0, \qquad (15)$$

$$\boldsymbol{w}_j \boldsymbol{h}^{(k)} - b_j < 0. \qquad (16)$$

As Fig. 1 shows, we apply the distance of vectors to reflect their inner product or similarity in the metric space. Without considering the bias $\boldsymbol{b}$, the CE loss push feature vector $\boldsymbol{h}^{(k)}$ closer to its classifier vector $\boldsymbol{w}_k$ compared to others $\{\boldsymbol{w}_\ell\}_{\ell \neq k}$, implying a *unbounded* feature space for each category and bad intra-class compactness. In addition, any two unbounded feature spaces introduced by CE could share the same decision boundary, indicating bad inter-class distinctiveness. In the training with CE, the bias $b_k$ acts as a com-

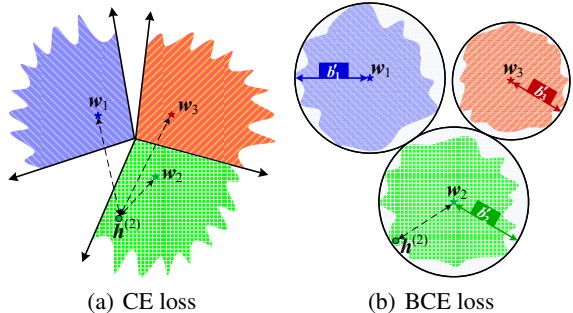

(a) CE loss       (b) BCE loss

Figure 1: The feature distributions of CE and BCE losses in the distance space. We apply the blue, red, and green shading to indicate the feature space of three categories, respectively. The pentagrams represent their classifier vectors, and the solid dot represents a general feature vector $\boldsymbol{h}^{(2)}$ in the second category. Since the distance between two vectors is inversely proportional to their similarity/inner product, CE loss requires the distance from the feature to its classifier vector to be less than the distance to other classifier vectors, while BCE loss requires the distance to be less than its corresponding bias. Small $b_k'$ implies large $b_k$ in Eqs. (15-16).

pensation to adjust the distance/decision score between the sample features and the classifier vector, introducing indirect constraint across sample features by Eq. (14). This constraint will vanish if $b_k = b_\ell$ for $\forall k, \ell \in [K]$, which could be reached at the minimum points of CE according to Theorem 1. Overall, in the training of classification models, CE does not require absolutely large positive decision scores or absolutely small negative ones, but only requires the positive one to be relatively greater than the negative ones for each sample, thereby implicitly enhancing the features' properties by correctly classifying samples one-by-one.

In contrast, for BCE, Eq. (15) requires the feature $\boldsymbol{h}^{(k)}$ to fall within a *closed* hypersphere centered at its classifier vector $\boldsymbol{w}_k$ with a "radius" of $b_k$, while Eq. (16) requires that any two hypersphere

do not intersect, indicating well intra-class compactness and inter-class distinctiveness. In other words, BCE presents explicitly constraint across-samples in the training. While Eq. (15) requires the positive decision scores of all samples are uniformly larger than threshold $t = 0$, Eq. (16) requires the negative ones of all samples are uniformly smaller than the unified threshold, i.e.,

$$\min \bigcup_{k=1}^{K} \bigcup_{i=1}^{n} \{\boldsymbol{w}_k^T \boldsymbol{h}_i^{(k)} - b_k\} > t \geq \max \bigcup_{k=1}^{K} \bigcup_{i=1}^{n} \{\boldsymbol{w}_j^T \boldsymbol{h}_i^{(k)} - b_j\}_{\substack{j=1 \\ j \neq k}}^{K}, \tag{17}$$

while the unified threshold $t$ might be not exactly zero in practice. As Eq. (17), BCE expects the positive decision scores to be uniformly high and the negative ones to be uniformly low, which could result in better compactness and distinctiveness than that introduced by Eq. (14). For the $k$th category, the bias $b_k$ would be absorbed into the threshold. In contrary, the bias $b_k$ explicitly reflect the intra-class compactness of its corresponding category and the inter-class distinctiveness between its category with other different categories. Therefore, BCE can explicitly enhance the compactness and distinctiveness across sample features by learning well biases.

**The decision scores in practical training.** In deep learning, gradient descent and back propagation are the most commonly used techniques for the model training. We here analyze the gradients in terms of the positive decision score $(\boldsymbol{w}_k \boldsymbol{h}_i^{(k)} - b_k)$ and negative one $(\boldsymbol{w}_j \boldsymbol{h}_i^{(k)} - b_j)$ for any sample $\boldsymbol{X}_i^{(k)}$ from category $k$ with $\forall j \neq k$

$$\frac{\partial f_{\mathrm{ce}}(\boldsymbol{W}, \boldsymbol{H}, \boldsymbol{b})}{\partial(\boldsymbol{w}_k \boldsymbol{h}_i^{(k)} - b_k)} = \frac{\mathrm{e}^{\boldsymbol{w}_k \boldsymbol{h}_i^{(k)} - b_k}}{\sum_{\ell} \mathrm{e}^{\boldsymbol{w}_\ell \boldsymbol{h}_i^{(k)} - b_\ell}} - 1, \quad \frac{\partial f_{\mathrm{ce}}(\boldsymbol{W}, \boldsymbol{H}, \boldsymbol{b})}{\partial(\boldsymbol{w}_j \boldsymbol{h}_i^{(k)} - b_j)} = \frac{\mathrm{e}^{\boldsymbol{w}_j \boldsymbol{h}_i^{(k)} - b_j}}{\sum_{\ell} \mathrm{e}^{\boldsymbol{w}_\ell \boldsymbol{h}_i^{(k)} - b_\ell}}, \quad \text{and} \tag{18}$$

$$\frac{\partial f_{\mathrm{bce}}(\boldsymbol{W}, \boldsymbol{H}, \boldsymbol{b})}{\partial(\boldsymbol{w}_k \boldsymbol{h}_i^{(k)} - b_k)} = \frac{1}{1 + \mathrm{e}^{-\boldsymbol{w}_k \boldsymbol{h}_i^{(k)} + b_k}} - 1, \quad \frac{\partial f_{\mathrm{bce}}(\boldsymbol{W}, \boldsymbol{H}, \boldsymbol{b})}{\partial(\boldsymbol{w}_j \boldsymbol{h}_i^{(k)} - b_j)} = \frac{1}{1 + \mathrm{e}^{-\boldsymbol{w}_j \boldsymbol{h}_i^{(k)} + b_j}}. \tag{19}$$

According to Eq. (18), for any two samples $\boldsymbol{X}_i^{(k)}, \boldsymbol{X}_{i'}^{(k)}$ from the same category $k$ with diverse initial positive scores, if their predicted probabilities are equal, i.e., $\frac{\mathrm{e}^{\boldsymbol{w}_k \boldsymbol{h}_i^{(k)} - b_k}}{\sum_{\ell} \mathrm{e}^{\boldsymbol{w}_\ell \boldsymbol{h}_i^{(k)} - b_\ell}} = \frac{\mathrm{e}^{\boldsymbol{w}_k \boldsymbol{h}_{i'}^{(k)} - b_k}}{\sum_{\ell} \mathrm{e}^{\boldsymbol{w}_\ell \boldsymbol{h}_{i'}^{(k)} - b_\ell}}$, which is somewhat likely to occur during the practical training, then their positive scores will experience the same update of amplitude during back propagation. Consequently, it will be difficult to update the positive scores to the uniformly high level, impeding the enhancement of intra-class compactness within the same category. A similar phenomenon can also occur with the negative decision scores, resulting in unsatisfactory inter-class distinctiveness in the training with CE loss.

In contrast, according to Eq. (19), during training with BCE loss, the large positive decision scores $(\boldsymbol{w}_k \boldsymbol{h}_i^{(k)} - b_k)$ were updated for the small amplitude $1 - \frac{1}{1 + \exp(-\boldsymbol{w}_k \boldsymbol{h}_i^{(k)} + b_k)}$, while the small ones were updated for the large update amplitude, facilitating a more rapid adjustment of positive scores across different samples to a uniform high level, to enhance the intra-class compactness of sample features. For the negative decision scores, similarly, the large/small score will be updated with large/samll amplitudes in the training with BCE loss to adjust them to a uniform low level, enhancing the inter-class distinctiveness.

**The classifier bias in practice.** During the model training, the classifier bias is also updated through the gradient descent, and the positive and negative decision scores are constrained by approaching the stable point of the bias. For CE, the gradient of bias $b_k$ is

$$\frac{\partial f_{\mathrm{ce}}}{\partial b_k} = \frac{1}{nK} \left( n - \sum_{j=1}^{K} \sum_{i=1}^{n} \frac{\mathrm{e}^{\boldsymbol{w}_k \boldsymbol{h}_i^{(j)} - b_k}}{\sum_{\ell} \mathrm{e}^{\boldsymbol{w}_\ell \boldsymbol{h}_i^{(j)} - b_\ell}} \right) + \lambda_{\boldsymbol{b}} b_k \to \lambda_{\boldsymbol{b}} b. \tag{20}$$

As approaching the stable point of the bias, i.e., the points satisfying $\frac{\partial f_{\mathrm{ce}}}{\partial b_k} = 0$, Eq. (20) presents constraint on the relative value of the exponential decision scores. This constraint will vanish as reaching the minimum of CE, and the bias gradient $\frac{\partial f_{\mathrm{ce}}}{\partial b_k}$ approaches $\lambda_{\boldsymbol{b}} b$, according to Eq. (13). At the minimum points, the update amplitude of bias is $\eta \lambda_{\boldsymbol{b}} b$, where $\eta$ denotes the learning rate. If $\lambda_{\boldsymbol{b}} = 0$, the update is zero, and the final bias can locate at any point on the ridge line $\boldsymbol{b} = b\mathbf{1}$, where $b$ is depended on some other factors, such as the bias initial value, but not the relationship between the bias and the decision scores. If $\lambda_{\boldsymbol{b}} > 0$, one can concluded $b = 0$; however, in practice, this

theoretical value might be not reached due to that $\eta\lambda_b$ will be very small at the terminal phase of practical training. The above analysis implies that the classifier bias, in the training with CE, cannot provide consistent and explicit constraints on the decision scores, and thus almost does not affect the final features' properties.

In contrast, for BCE, the gradient of bias $b_k$ is

$$\frac{\partial f_{\text{bce}}}{\partial b_k} = \frac{1}{nK}\left(n - \sum_{j=1}^{K}\sum_{i=1}^{n}\frac{1}{1 + e^{-\boldsymbol{w}_k \boldsymbol{h}_i^{(j)} + b_k}}\right) + \lambda_b b_k \rightarrow \text{RHS of Eq. (12)}, \qquad (21)$$

which presents clear constraint on the absolute value of the exponential decision scores for the all samples. The constraint evolve into Eq. (12) when BCE reaches its minimum points. Therefore, as approaching the stable point, the classifier bias consistently and explicitly constrain the decision scores, regardless $\lambda_b = 0$ or not, and it will separate the final positive and negative decision scores if Eq. (165) holds. In other words, the classifier bias in BCE plays a substantial role in enhancing the final features' properties.

## 4 EXPERIMENTS

To compare CE and BCE in deep feature learning, we train deep classification models, ResNet18 (He et al., 2016), ResNet50 (He et al., 2016), and DenseNet121 (Huang et al., 2017), using the two losses respectively, on three popular datasets, including MNIST (LeCun et al., 1998), CIFAR10 (Krizhevsky et al., 2009), and CIFAR100 (Krizhevsky et al., 2009). We train the models using SGD and AdamW for 100 epochs with batch size of 128. The initial learning rate is set to 0.01 and 0.001 for SGD and AdamW, which is respectively decayed in "step" and "cosine" schedulers.

### 4.1 MAXIMIZING COMPACTNESS AND DISTINCTIVENESS BY BCE AND CE

We first experimentally illustrate that both BCE and CE can maximize the intra-class compactness and inter-class distinctiveness among sample features, i.e., resulting in neural collapse (NC). Similar to (Zhu et al., 2021; Zhou et al., 2022), we do not apply any data augmentation in the experiments of NC, and adopt the metrics, $\mathcal{NC}_1, \mathcal{NC}_2$, and $\mathcal{NC}_3$ (see supplementary for their definitions), to measure the properties of NC1, NC2, and NC3. The lower metrics reflect the better NC properties.

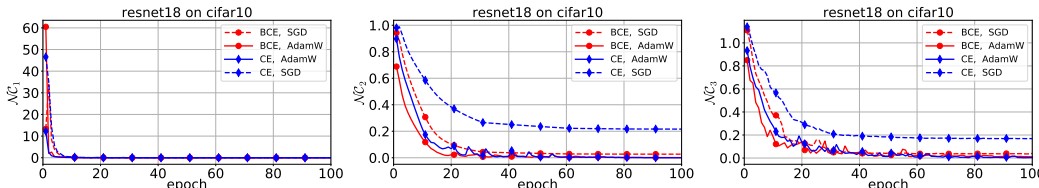

Figure 2: NC metrics of ResNet18 trained on CIFAR10 with CE and BCE using SGD and AdamW, respectively. The NC metrics of CE and BCE approach zero at the terminal phase of training, while the NC metrics of BCE decrease faster than that of CE in the first 20 epochs.

In the training, we set $\lambda_W = \lambda_H = \lambda_b = 5 \times 10^{-4}$, and no weight decay is applied on the other parameters of the model $\mathcal{M}$. Fig. 2 shows the NC results of ResNet18 trained by CE and BCE with two optimizers on CIFAR10, and the other results are presented in the supplementary. In the figure, the red curves with dot markers exhibit the evolution of the metrics $\mathcal{NC}_1, \mathcal{NC}_2, \mathcal{NC}_3$ of BCE, and the blue curves with diamond markers exhibit that of CE. All the three NC metrics consistently approach zero in the training with different losses and optimizers, which matches the conclusions of Theorem 1 and 2. In the initial training stage (the first 20 epochs), the NC metrics of BCE usually decrease faster than that of CE, implying that BCE is easier to result in NC.

**The final classifier bias and decision scores**. As reaching NC and maximizing the feature compactness and distinctiveness, the final classifier biases and the positive/negative decision scores will converge to fixed values. We compute their means and standard deviations for the different models on the training set. Table 1 shows the results on CIFAR10. Except for that of ResNet50 and DenseNet121 trained by SGD, the standard deviations of the final decision scores and classifier biases are very small, and less than 0.3 when the models are trained using AdamW, indicating that the

diverse classifier biases are almost equal, so are all the final positive/negative decision scores. As $\lambda_b > 0$, the final classifier biases are near zero ($\hat{b} \approx 0$) in the CE-trained models, while, in the BCE-trained models, the biases make that the function ($\alpha(b)$ in Table 1) on the RHS of Eq. (12) almost degenerate to zero, i.e., $\alpha(\hat{b}) \approx 0$, aligning with our analysis.

**The failures in NC.** According to the results in Table 1, one can find that NC might be not caused by the ResNet50 and DenseNet121 trained by SGD. Though the classification accuracy ($\mathcal{A}$) of the two models are higher than $99.00\%$ (almost $100\%$), their positive and negative decision scores have large standard deviations, implying that the decision scores have not converged to the fixed values and conflict with the results (Eq. (13)) of neural collapse. It still requires a long time training to reach the neural collapse, after zero classification error. More discussions can be found in the supplementary.

Table 1: The means and standard deviations of positive/negative decision scores ($s_{\text{pos}}, s_{\text{neg}}$, without bias) and the classifier biases ($\hat{b}$) of the models trained by CE and BCE on CIFAR10 with $\lambda_W = \lambda_H = \lambda_b = 5 \times 10^{-4}$. The score values are computed on the training set, and $\mathcal{A}$ denotes the classification accuracy on the training set.

| $\mathcal{M}$ | | CIFAR10 | | | |
| --- | --- | --- | --- | --- | --- |
| | | SGD | | AdamW | |
| | | CE | BCE | CE | BCE |
| ResNet18 | $s_{\text{pos}}$ | $5.71 \pm 0.23$ | $6.56 \pm 0.20$ | $5.64 \pm 0.06$ | $7.50 \pm 0.05$ |
| | $s_{\text{neg}}$ | $-0.64 \pm 0.36$ | $-3.46 \pm 0.19$ | $-0.63 \pm 0.01$ | $-2.36 \pm 0.02$ |
| | $\hat{b}$ | $-0.01 \pm 0.04$ | $2.26 \pm 0.07$ | $-0.00 \pm 0.00$ | $3.29 \pm 0.01$ |
| | $\alpha(\hat{b})$ | — | $-0.03$ | — | $-0.01$ |
| | $\mathcal{A}$ | $99.99$ | $100.0$ | $100.0$ | $100.0$ |
| ResNet50 | $s_{\text{pos}}$ | $5.74 \pm 8.21$ | $6.56 \pm 4.39$ | $5.64 \pm 0.11$ | $7.44 \pm 0.28$ |
| | $s_{\text{neg}}$ | $-0.64 \pm 14.1$ | $-3.57 \pm 7.01$ | $-0.63 \pm 0.02$ | $-2.45 \pm 0.22$ |
| | $\hat{b}$ | $0.00 \pm 0.14$ | $2.40 \pm 0.15$ | $-0.00 \pm 0.01$ | $3.21 \pm 0.03$ |
| | $\alpha(\hat{b})$ | — | $-0.02$ | — | $-0.01$ |
| | $\mathcal{A}$ | $99.61$ | $99.65$ | $99.99$ | $100.0$ |
| DenseNet121 | $s_{\text{pos}}$ | $5.72 \pm 1.72$ | $6.24 \pm 0.84$ | $5.62 \pm 0.29$ | $7.67 \pm 0.12$ |
| | $s_{\text{neg}}$ | $-0.63 \pm 0.87$ | $-3.62 \pm 1.63$ | $-0.62 \pm 0.03$ | $-2.17 \pm 0.06$ |
| | $\hat{b}$ | $0.00 \pm 0.04$ | $2.09 \pm 0.12$ | $0.00 \pm 0.01$ | $3.46 \pm 0.02$ |
| | $\alpha(\hat{b})$ | — | $-0.03$ | — | $-0.01$ |
| | $\mathcal{A}$ | $99.40$ | $99.72$ | $99.87$ | $100.0$ |

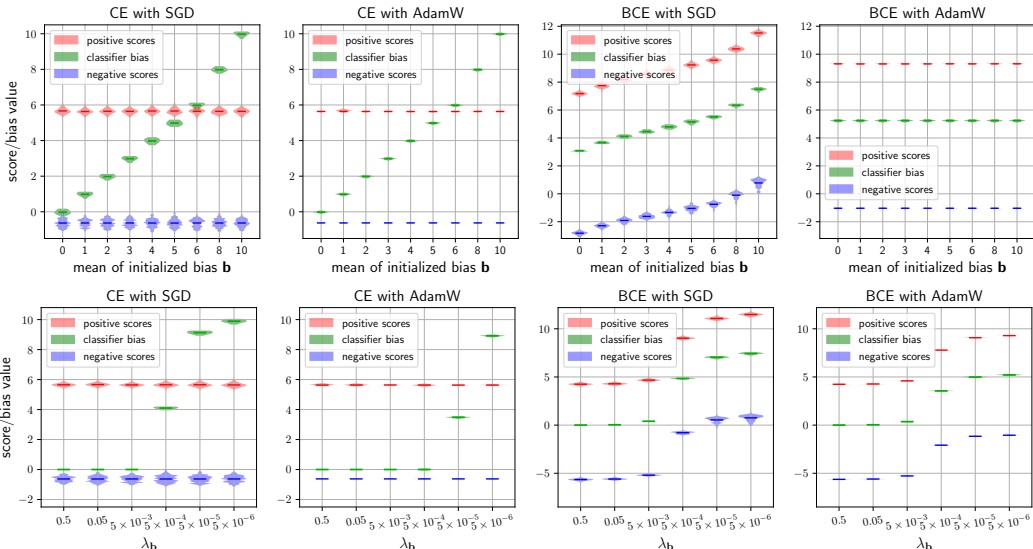

Figure 3: The distributions of the final classifier bias and positive/negative decision scores for ResNet18 trained on MNIST with fixed weight decay factor $\lambda_b$ (top) and varying $\lambda_b$ (bottom), while $\lambda_W = \lambda_H = 5 \times 10^{-4}$. The mean of initialized bias $b$ is respectively set as $0, 1, 2, 3, 4, 5, 6, 8, 10$ in the experiments with fixed $\lambda_b = 0$, and the bias mean is set as $10$ in the experiments with varying $\lambda_b$.

**The bias decay factor $\lambda_b$.** To illustrate the different effects of classifier bias of CE and BCE on the decision scores, we conduct two groups of experiments by respectively applying fixed and varying classifier bias decay factor $\lambda_b$ in the training of ResNet18 on MNIST: (1) with fixed $\lambda_b = 0$ and default other hyper-parameters, respectively, setting the mean of the initialized classifier bias to $0, 1, 2, 3, 4, 5, 6, 8$, and $10$; (2) with varying $\lambda_b = 0.5, 0.05, 5 \times 10^{-3}, 5 \times 10^{-4}, 5 \times 10^{-5}$, and $5 \times 10^{-6}$, respectively, setting the mean of initialized classifier bias to $10$. Fig. 3 shows the distributions of final classifier bias and positive/negative decision scores (without bias) using violin plots for the models in these experiments, while the numerical results are presented in Tables S-8 and S-9 in supplementary. One can find from Fig. 3(top), for the CE-trained models with $\lambda_b = 0$, the final classifier bias values are almost entirely determined by their initial values, no matter which

optimizer was applied. For the CE-trained models in Fig. 3(bottom), the means of the final classifier bias reach to zero from the initial mean of 10 with appropriate lager $\lambda_b$ ($\geq 5 \times 10^{-3}$ for SGD and $\geq 5 \times 10^{-4}$ for AdamW), and they do not achieve the theoretical value when $\lambda_b$ is too small. As a comparison, for the all CE-trained models, their final positive and negative decision scores respectively converge to around $5.64$ and $-0.63$ (see supplementary for details). In total, in CE-trained models, the classifier bias hardly affects the decision scores, and thus almost does not affect the final feature properties.

In contrast, for the BCE-trained models in Fig. 3, their final positive and negative decision scores are always separated by the final classifier biases, no matter what the initial mean of classifier bias, $\lambda_b$, or optimizer are, and clear correlation exists between the bias and positive/negative decision scores. These results imply that, in the training with BCE, the classifier bias has a substantial impact on the sample feature distribution, thereby enhancing the compactness and distinctiveness across samples.

## 4.2 BCE performing better than CE in enhancing feature properties

To further demonstrate that BCE performs better than CE in enhancing the intra-class compactness and inter-class distinctiveness of sample features in the practical training, we train the three classification models by applying two different data augmentation techniques, (1) DA1: random cropping and horizontal flipping, (2) DA2: Mixup and CutMix, on CIFAR10 and CIFAR100 using SGD and AdamW, respectively. In the experiments, we take a global weight decay factor $\lambda$ for the all parameters in the models, including the classifier weight and bias, and $\lambda = 5 \times 10^{-4}$ for SGD, $\lambda = 0.05$ for AdamW. The other hyper-parameters are presented in the supplementary. To compare the results of BCE and CE, besides the classification accuracy ($\mathcal{A}$), we define and apply three other metrics, uniform accuracy ($\mathcal{A}_{\mathrm{Uni}}$), compactness ($\mathcal{E}_{\mathrm{com}}$), and distinctiveness ($\mathcal{E}_{\mathrm{dis}}$), seeing Eqs. (43,47,48) in supplementary for the definitions. While $\mathcal{A}_{\mathrm{Uni}}$ is evolved from Eq. (17), it is calculated on the decision scores across samples, simultaneously reflecting the feature compactness and distinctiveness; as their name implies, $\mathcal{E}_{\mathrm{com}}$ and $\mathcal{E}_{\mathrm{dis}}$ respectively measure the intra-class compactness and inter-class distinctiveness among sample features.

Table 2: The classification on the test set of CIFAR10 and CIFAR100. The accuracy ($\mathcal{A}$) of most BCE-trained models is higher than that of CE-trained ones, while BCE-trained models perform consistently and significantly better than CE-trained models in terms of uniform accuracy ($\mathcal{A}_{\mathrm{Uni}}$).

| $\mathcal{D}$ | $\mathcal{M}$ | Loss | SGD | | | | AdamW | | | |
|---|---|---|---|---|---|---|---|---|---|---|
| | | | DA1 | | DA1+DA2 | | DA1 | | DA1+DA2 | |
| | | | $\mathcal{A}$ | $\mathcal{A}_{\mathrm{Uni}}$ | $\mathcal{A}$ | $\mathcal{A}_{\mathrm{Uni}}$ | $\mathcal{A}$ | $\mathcal{A}_{\mathrm{Uni}}$ | $\mathcal{A}$ | $\mathcal{A}_{\mathrm{Uni}}$ |
| CIFAR10 | R18 | CE | 92.82 | 85.20 | 92.71 | 89.08 | 93.36 | 88.97 | 95.72 | 94.34 |
| | | BCE | 93.22 | 91.92 | 93.64 | 91.87 | 93.95 | 93.37 | 95.57 | 95.16 |
| | | $\Delta$ | +0.40 | +6.72 | +0.93 | +2.79 | +0.59 | +4.40 | −0.15 | +0.82 |
| | R50 | CE | 92.69 | 85.23 | 92.74 | 89.58 | 94.48 | 87.86 | 96.00 | 94.31 |
| | | BCE | 93.40 | 92.48 | 93.20 | 91.50 | 94.02 | 93.55 | 96.15 | 95.72 |
| | | $\Delta$ | +0.71 | +7.25 | +0.46 | +1.92 | −0.46 | +5.69 | +0.15 | +1.41 |
| | D121 | CE | 87.87 | 78.67 | 86.65 | 81.54 | 90.42 | 83.62 | 92.55 | 90.70 |
| | | BCE | 88.66 | 87.58 | 87.78 | 84.95 | 90.55 | 89.91 | 92.59 | 91.78 |
| | | $\Delta$ | +0.79 | +8.91 | +1.13 | +3.41 | +0.13 | +6.29 | +0.04 | +1.08 |
| CIFAR100 | R18 | CE | 71.16 | 43.21 | 71.76 | 56.66 | 71.69 | 49.17 | 76.53 | 64.43 |
| | | BCE | 72.16 | 63.33 | 72.34 | 62.89 | 73.15 | 66.27 | 76.70 | 69.96 |
| | | $\Delta$ | +1.00 | +20.1 | +0.58 | +6.23 | +1.46 | +17.1 | +0.17 | +5.53 |
| | R50 | CE | 71.60 | 44.20 | 70.32 | 55.17 | 74.95 | 48.79 | 78.58 | 67.79 |
| | | BCE | 71.75 | 64.07 | 71.87 | 62.82 | 75.25 | 68.84 | 78.47 | 72.68 |
| | | $\Delta$ | +0.15 | +19.9 | +1.55 | +7.65 | +0.30 | +20.1 | −0.11 | +4.89 |
| | D121 | CE | 60.79 | 32.93 | 57.23 | 39.82 | 63.65 | 38.76 | 68.99 | 57.15 |
| | | BCE | 61.10 | 53.47 | 58.35 | 47.68 | 63.56 | 57.28 | 69.40 | 63.52 |
| | | $\Delta$ | +0.21 | +20.5 | +1.12 | +7.85 | −0.09 | +18.5 | +0.41 | +6.37 |

Table 2 shows the classification results of the three models ("R18","R50", and "D121" respectively stand for ResNet18, ResNet50, and DenseNet121) on the test set of CIFAR10 and CIFAR100. From the table, one can find that, BCE is better than CE in term of accuracy ($\mathcal{A}$) in most cases, and it performs consistently and significantly superior to CE in term of uniform accuracy ($\mathcal{A}_{\mathrm{Uni}}$). Taking CIFAR10 for example, among the twelve pairs of models trained by CE and BCE, BCE slightly reduced the accuracy of two pairs of models, while the gain of uniform accuracy introduced by BCE is $0.82\%$ at least for the all models. For CIFAR100, the gain of BCE in uniform accuracy could be more than $20\%$, and the classification accuracy of BCE is still higher than that of CE in most cases. These results illustrate that BCE can usually achieve better classification results than CE, which is likely resulted from its enhancement in compactness and distinctiveness among sample features.

Furthermore, similar to BCE, the better data augmentation techniques and optimizer can simultaneously improve the classification results of models. For example, Mixup, CutMix, and AdamW

increase $\mathcal{A}$ and $\mathcal{A}_{\text{Uni}}$ from 92.82% and 85.20% to 95.72% and 94.34%, respectively, for ResNet18 trained on CIFAR10. In addition, the higher performance of BCE than CE with only DA1 implies that the superiority of BCE is not resulted from the alignment with Mixup and CutMix, which is not consistent with the statements about BCE by Wightman et al. (2021).

As the uniform accuracy simultaneously reflect the intra-class compactness and inter-class distinctiveness, the higher uniform accuracy $\mathcal{A}_{\text{Uni}}$ of BCE-trained models implies their better feature properties. Table 3 presents the compactness ($\mathcal{E}_{\text{com}}$) and distinctiveness ($\mathcal{E}_{\text{dis}}$) of the trained models. One can clearly observe that, in most cases, BCE improves the compactness and distinctiveness of the sample features extracted by the models compared to CE,

Table 3: The feature properties on the test set of CIFAR10 and CIFAR100. The feature compactness ($\mathcal{E}_{\text{com}}$) and distinctiveness ($\mathcal{E}_{\text{dis}}$) of BCE-trained models are usually better than that of CE-trained models. See supplementary for the definitions of $\mathcal{E}_{\text{com}}$ and $\mathcal{E}_{\text{dis}}$.

| $\mathcal{D}$ | $\mathcal{M}$ | Loss | SGD | | | | AdamW | | | |
|---|---|---|---|---|---|---|---|---|---|---|
| | | | DA1 | | DA1+DA2 | | DA1 | | DA1+DA2 | |
| | | | $\mathcal{E}_{\text{com}}$ | $\mathcal{E}_{\text{dis}}$ | $\mathcal{E}_{\text{com}}$ | $\mathcal{E}_{\text{dis}}$ | $\mathcal{E}_{\text{com}}$ | $\mathcal{E}_{\text{dis}}$ | $\mathcal{E}_{\text{com}}$ | $\mathcal{E}_{\text{dis}}$ |
| CIFAR10 | R18 | CE | 0.8541 | 0.2553 | 0.8148 | 0.2088 | 0.8546 | 0.2694 | 0.8929 | 0.3307 |
| | | BCE | **0.9056** | **0.3049** | **0.8438** | **0.2387** | **0.9140** | **0.3254** | **0.9178** | **0.3669** |
| | R50 | CE | 0.8351 | 0.1782 | 0.8564 | 0.2027 | 0.8547 | 0.2332 | **0.9529** | **0.3782** |
| | | BCE | **0.8990** | **0.2322** | **0.8693** | **0.2161** | **0.8912** | **0.2720** | 0.9168 | 0.3569 |
| | D121 | CE | 0.7874 | 0.3123 | 0.7672 | 0.2805 | 0.7463 | 0.3070 | 0.8201 | **0.3194** |
| | | BCE | **0.8458** | **0.3319** | **0.8089** | **0.2973** | **0.8302** | **0.3371** | **0.8371** | 0.3190 |
| CIFAR100 | R18 | CE | 0.7234 | **0.2699** | 0.7127 | 0.2575 | 0.6923 | 0.2895 | 0.7140 | **0.3073** |
| | | BCE | **0.7331** | 0.2624 | **0.7289** | **0.2688** | **0.7265** | **0.2930** | **0.7422** | 0.2906 |
| | R50 | CE | 0.7084 | 0.2002 | 0.7101 | 0.1866 | 0.6886 | 0.2581 | 0.7229 | **0.3646** |
| | | BCE | **0.7326** | **0.2196** | **0.7400** | **0.2184** | **0.7517** | **0.2783** | **0.7631** | 0.3254 |
| | D121 | CE | 0.7120 | **0.3097** | 0.7280 | **0.3171** | 0.6472 | 0.2981 | 0.6998 | **0.3403** |
| | | BCE | **0.7324** | 0.2947 | **0.7363** | 0.3049 | **0.7091** | **0.3008** | **0.7262** | 0.3259 |

which is consistent with our expectations and provides a solid and reasonable explanation for the higher performance of BCE in tasks that require feature comparison, such as facial recognition and verification (Wen et al., 2022; Zhou et al., 2023).

Fig. 4 visually shows the feature distributions on the testing data of CIFAR10 for ResNet18 trained by CE (left) and BCE (right) with "DA1+DA2" and AdamW. One can find that, for CE-trained model, its feature distributions for categories 3 and 5 (i.e., "cat" and "dog") overlap with each other, and the sample features of the third category exhibit clear internal dispersion; in contrast,

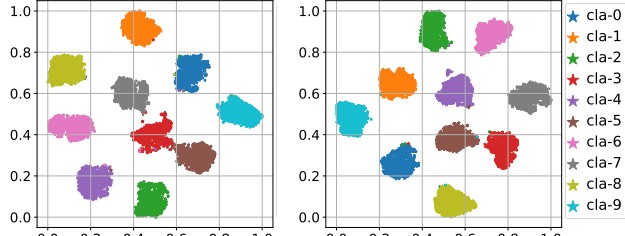

Figure 4: Distributions of features extracted by ResNet18 trained on CIFAR10 using CE (left) and BCE (right) in t-SNE.

the features of BCE-trained ResNet18 for these categories are distributed in more compact areas and have significant gaps between them, implying better feature compactness and distinctiveness.

## 5 CONCLUSIONS

This paper compares CE and BCE losses in deep feature learning. Both the losses can maximize the intra-class compactness and inter-class distinctiveness among sample features, i.e., leading to neural collapse when reaching their minimums. In the training, CE implicitly enhances the feature properties by correctly classifying samples one-by-one. In contrast, BCE can adjust the positive and negative decision scores across samples, and, in this process, its classifier bias plays a substantial and consistent role, making it explicitly enhance the intra-class compactness and inter-class distinctiveness of features. Therefore, BCE can usually achieve better classification performance.

**Limitations**. The decision scores measure the inner product/similarity of sample features to each classifier vector, which reflect the feature properties. However, it does not directly calculate the measurements among samples, nor can it be used to directly measure the compactness and distinctiveness of sample features. In the future, we will analyze the CE and BCE losses used for feature contrastive learning and compare the feature properties brought by them.

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
