# BCE VS. CE IN DEEP FEATURE LEARNING

## ABSTRACT

When training classification models, it expects that the leaned features are compact within classes, and can well separate different classes. As a dominant loss function to train classification models, the minimization of CE (Cross-entropy) loss can maximize the compactness and distinctiveness, i.e., reaching neural collapse. The recently published works show that BCE (Binary CE) loss performs also well in multi-class tasks. In this paper, we compare BCE and CE in the context of deep feature learning. For the first time, we prove that BCE can also maximize the intra-class compactness and inter-class distinctiveness when reaching its minimum, i.e., leading to neural collapse. We point out that CE measures the relative values of decision scores in the model training, implicitly enhancing the feature properties by classifying samples one-by-one. In contrast, BCE measures the absolute values of decision scores and adjust the positive/negative decision scores across all samples to uniform high/low levels. Meanwhile, the classifier bias in BCE presents a substantial constraint on the samples' decision scores. Thereby, BCE explicitly enhances the feature properties in the training. The experimental results are aligned with above analysis, and show that BCE consistently and significantly improve the classification performance and leads to better compactness and distinctiveness among sample features.

## 1 INTRODUCTION

Cross-entropy (CE) loss is the most commonly used loss function for classifications and feature learning. In a multi-class classification with $K$ categories, for any sample $\boldsymbol{X}^{(k)}$ from category $k$, a model $\mathcal{M}$ first extracts its feature $\boldsymbol{h}^{(k)} = \mathcal{M}(\boldsymbol{X}^{(k)}) \in \mathbb{R}^d$, which is output from the penultimate hidden layer in deep model. Then a linear classifier with weight $\boldsymbol{W} = [\boldsymbol{w}_1, \cdots, \boldsymbol{w}_K]^T \in \mathbb{R}^{K \times d}$ and bias $\boldsymbol{b} = [b_1, \cdots, b_K]^T \in \mathbb{R}^K$ transforms the feature into $K$ logits/decision scores, $\{\boldsymbol{w}_j^T \boldsymbol{h}^{(k)} - b_j\}_{j=1}^K$, which are finally converted into predicted probabilities by Softmax, and computed the loss using cross-entropy,

$$\mathcal{L}_{\text{ce}}\big(\boldsymbol{z}^{(k)}\big) = -\boldsymbol{y}_k^T \cdot \log\Big(\text{Softmax}\big(\boldsymbol{z}^{(k)}\big)\Big) = \log\left(1 + \sum_{\substack{\ell=1 \\ \ell \neq k}}^K \frac{e^{\boldsymbol{w}_\ell^T \boldsymbol{h}^{(k)} - b_\ell}}{e^{\boldsymbol{w}_k^T \boldsymbol{h}^{(k)} - b_k}}\right), \tag{1}$$

where $\boldsymbol{z}^{(k)} = \boldsymbol{W}\boldsymbol{h}^{(k)} - \boldsymbol{b}$ and $\boldsymbol{y}_k$ is the one-hot label, i.e., the vector with one only in the $k$th entry.

In the multi-class classification, binary CE (BCE) loss is deduced by decomposing the task into $K$ binary tasks and predicting whether the sample $\boldsymbol{X}^{(k)}$ belongs to the $j$th category, for $\forall j \in [K]$,

$$\mathcal{L}_{\text{bce}}\big(\boldsymbol{z}^{(k)}\big) = -\boldsymbol{y}_k^T \cdot \log\Big(\text{Sigmoid}\big(\boldsymbol{z}^{(k)}\big)\Big) - (\boldsymbol{1} - \boldsymbol{y}_k)^T \cdot \log\Big(\boldsymbol{1} - \text{

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

# BCE vs. CE in Deep Feature Learning

## Supplementary Material

## S-1 NEURAL COLLAPSE AND FEATURE PROPERTY

### S-1.1 NEURAL COLLAPSE

The neural collapse was first found by Papyan et al. (2020), which refers to four properties about the sample features $\{\boldsymbol{h}_i^{(k)}\}$ and the classifier vectors $\{\boldsymbol{w}_k\}$ at the terminal phase of training (Han et al., 2022), as list in Sec. 2.2. These four properties can be formulized as follows.

- **NC1**, within-class variability collapse, $\boldsymbol{\Sigma}_B^{\dagger} \boldsymbol{\Sigma}_W \to \boldsymbol{0}$, where

$$\boldsymbol{\Sigma}_B = \frac{1}{K} \sum_{k=1}^{K} \left(\bar{\boldsymbol{h}}^{(k)} - \bar{\boldsymbol{h}}\right)\left(\bar{\boldsymbol{h}}^{(k)} - \bar{\boldsymbol{h}}\right)^T \tag{22}$$

$$\boldsymbol{\Sigma}_W = \frac{1}{\sum_k n_k} \sum_{k=1}^{K} \sum_{i=1}^{n_k} \left(\boldsymbol{h}_i^{(k)} - \bar{\boldsymbol{h}}^{(k)}\right)\left(\boldsymbol{h}_i^{(k)} - \bar{\boldsymbol{h}}^{(k)}\right)^T \tag{23}$$

$$\bar{\boldsymbol{h}}^{(k)} = \frac{1}{n_k} \sum_{i=1}^{n_k} \boldsymbol{h}_i^{(k)}, \tag{24}$$

$$\bar{\boldsymbol{h}} = \frac{1}{\sum_k n_k} \sum_{k=1}^{K} \sum_{i=1}^{n_k} \boldsymbol{h}_i^{(k)} \tag{25}$$

and † denotes the Mooer-Penrose pseudo-inverse;

- **NC2**, convergence to simplex equiangular tight frame,

$$\left\|\bar{\boldsymbol{h}}^{(k)} - \bar{\boldsymbol{h}}\right\|_2 - \left\|\bar{\boldsymbol{h}}^{(k')} - \bar{\boldsymbol{h}}\right\|_2 \to 0, \tag{26}$$

$$\frac{\left\langle \bar{\boldsymbol{h}}^{(k)} - \bar{\boldsymbol{h}}, \, \bar{\boldsymbol{h}}^{(k')} - \bar{\boldsymbol{h}} \right\rangle}{\left\|\bar{\boldsymbol{h}}^{(k)} - \bar{\boldsymbol{h}}\right\|_2 \left\|\bar{\boldsymbol{h}}^{(k')} - \bar{\boldsymbol{h}}\right\|_2} \to \begin{cases} 1, & k = k', \\ -\frac{1}{K-1}, & k \neq k'; \end{cases} \tag{27}$$

- **NC3**, convergence to self-duality,

$$\frac{\boldsymbol{w}_k}{\left\|\boldsymbol{w}_k\right\|_2} - \frac{\bar{\boldsymbol{h}}^{(k)} - \bar{\boldsymbol{h}}}{\left\|\bar{\boldsymbol{h}}^{(k)} - \bar{\boldsymbol{h}}\right\|_2} \to 0; \tag{28}$$

- **NC4**, simplification to nearest class center,

$$\arg\max_j \left\{\boldsymbol{w}_j \boldsymbol{h} - b_j\right\}_{j=1}^{K} \to \arg\min_j \left\{\|\boldsymbol{h} - \bar{\boldsymbol{h}}^{(j)}\|_2\right\}_{j=1}^{K}. \tag{29}$$

In Sec. 4, we applied three metrics, $\mathcal{NC}_1, \mathcal{NC}_2, \mathcal{NC}_3$, to measure the above properties, similar to that defined in (Zhu et al., 2021; Zhou et al., 2022):

$$\mathcal{NC}_1 := \frac{1}{K}\text{trace}\left(\boldsymbol{\Sigma}_W \boldsymbol{\Sigma}_B^{\dagger}\right), \tag{30}$$

$$\mathcal{NC}_2 := \left\|\frac{\tilde{\boldsymbol{W}}\tilde{\boldsymbol{W}}^T}{\|\tilde{\boldsymbol{W}}\tilde{\boldsymbol{W}}^T\|_F} - \frac{1}{\sqrt{K-1}}\left(\boldsymbol{I}_K - \frac{1}{K}\boldsymbol{1}_K\boldsymbol{1}_K^T\right)\right\|_F, \tag{31}$$

$$\mathcal{NC}_3 := \left\|\frac{\boldsymbol{W}\tilde{\boldsymbol{H}}}{\|\boldsymbol{W}\tilde{\boldsymbol{H}}\|_F} - \frac{1}{\sqrt{K-1}}\left(\boldsymbol{I}_K - \frac{1}{K}\boldsymbol{1}_K\boldsymbol{1}_K^T\right)\right\|_F, \tag{32}$$

where

$$\tilde{\boldsymbol{W}} = [\boldsymbol{w}_1 - \bar{\boldsymbol{w}}, \boldsymbol{w}_2 - \bar{\boldsymbol{w}}, \cdots, \boldsymbol{w}_K - \bar{\boldsymbol{w}}]^T \in \mathbb{R}^{K \times d}, \tag{33}$$

$$\tilde{\boldsymbol{H}} = [\bar{\boldsymbol{h}}^{(1)} - \bar{\boldsymbol{h}}, \bar{\boldsymbol{h}}^{(2)} - \bar{\boldsymbol{h}}, \cdots, \bar{\boldsymbol{h}}^{(K)} - \bar{\boldsymbol{h}}] \in \mathbb{R}^{d \times K}, \tag{34}$$

$$\bar{\boldsymbol{w}} = \frac{1}{K} \sum_{k=1}^{K} \boldsymbol{w}_k. \tag{35}$$

When defining $\mathcal{NC}_2$, Zhu et al. (2021) and Zhou et al. (2022) did not subtract the classifier vectors with their mean, i.e., the original $\mathcal{NC}_2$ is defined as $\left\| \frac{\boldsymbol{W}\boldsymbol{W}^T}{\|\boldsymbol{W}\boldsymbol{W}^T\|_F} - \frac{1}{\sqrt{K-1}}\left(\boldsymbol{I}_K - \frac{1}{K}\mathbf{1}_K\mathbf{1}_K^T\right)\right\|_F$, with $\boldsymbol{W} = [\boldsymbol{w}_1, \boldsymbol{w}_2, \cdots, \boldsymbol{w}_K]^T \in \mathbb{R}^{K \times d}$.

As mentioned by Zhu et al. (2021) and Zhou et al. (2022), due to the "ReLU" activation functions before the FC classifiers in the deep models, the feature mean $\tilde{\boldsymbol{h}}_i = \frac{1}{K}\sum_{k=1}^{K}\boldsymbol{h}_i^{(k)}$ will be non-negative, which conflicts with $\tilde{\boldsymbol{h}}_i = \mathbf{0}$ required by Theorems 1 and 2. Then, the average features/class centers of $K$ categories do not directly form an ETF, while the globally-centered average features form ETF, i.e., NC2 properties described by Eqs. (26) and (27). As the proof of Theorems 1 and 2, in the neural collapse, the features of each category will be parallel to its classifier vector, i.e., $\boldsymbol{h}_i^{(k)} = \sqrt{\frac{\lambda_W}{n\lambda_H}}\boldsymbol{w}_k$ in Eqs (126,127). Therefore, the classifier vectors $\{\boldsymbol{w}_k\}$ should also subtract their global mean before form an ETF. In other words, the third NC property should be

$$\textbf{NC3'}: \quad \frac{\boldsymbol{w}_k - \bar{\boldsymbol{w}}}{\left\|\boldsymbol{w}_k - \bar{\boldsymbol{w}}\right\|_2} - \frac{\bar{\boldsymbol{h}}^{(k)} - \bar{\boldsymbol{h}}}{\left\|\bar{\boldsymbol{h}}^{(k)} - \bar{\boldsymbol{h}}\right\|_2} \to 0. \tag{36}$$

As our analysis, when a model falling to the neural collapse, its classification accuracy $\mathcal{A}$ and uniform accuracy $\mathcal{A}_{\text{Uni}}$ must be 100% on the training dataset.

## S-1.2 Feature Property

In the experiments, we applied four metrics to compare the performance of CE and BCE, i.e., classification accuracy $\mathcal{A}$, uniform accuracy $\mathcal{A}_{\text{Uni}}$, feature compactness $\mathcal{E}_{\text{com}}$, and distinctiveness $\mathcal{E}_{\text{dis}}$. These metrics on the training data will be maximized when the model, classifier, and loss in the neural collapse.

In a classification task, suppose a dataset $\mathcal{D} = \bigcup_{k=1}^{K}\mathcal{D}_k = \bigcup_{k=1}^{K}\bigcup_{i=1}^{n_k}\left\{\boldsymbol{X}_i^{(k)}\right\}$ from $K$ categories, where $\boldsymbol{X}_i^{(k)}$ denotes the $i$th sample from the category $k$. For the sample $\boldsymbol{X}_i^{(k)}$ in $\mathcal{D}$, a model $\mathcal{M}$ converts it into its feature $\boldsymbol{h}_i^{(k)} = \mathcal{M}(\boldsymbol{X}_i^{(k)}) \in \mathbb{R}^d$, where $d$ is the length of the feature vector. A linear, full connection (FC) classifier $\mathcal{C} = \left\{(\boldsymbol{w}_k, b_k)\right\}_{k=1}^{K}$ transform the feature into $K$ decision scores $\left\{\boldsymbol{w}_j\boldsymbol{h}_i^{(k)} - b_j\right\}_{j=1}^{K}$. Then, the sample can be correctly classified if

$$\boldsymbol{w}_k\boldsymbol{h}_i^{(k)} - b_k = \max\left\{\boldsymbol{w}_j\boldsymbol{h}_i^{(k)} - b_j\right\}_{j=1}^{K}, \tag{37}$$

which is equivalent to

$$k = \arg\max_{\ell}\{\boldsymbol{w}_\ell^T\boldsymbol{h}^{(k)} - b_\ell\}. \tag{38}$$

The the commonly used **classification accuracy** can be defined as

$$\mathcal{A}(\mathcal{M}, \mathcal{C}) = \frac{|\mathcal{D}(\mathcal{M}, \mathcal{C})|}{|\mathcal{D}|} \times 100\%, \tag{39}$$

where

$$\mathcal{D}(\mathcal{M}, \mathcal{C}) = \bigcup_{k=1}^{K}\left\{\boldsymbol{X}^{(k)} : k = \arg\max_{\ell}\{\boldsymbol{w}_\ell^T\boldsymbol{h}^{(k)} - b_\ell\}, \boldsymbol{X}^{(k)} \in \mathcal{D}_k, \boldsymbol{h}^{(k)} = \mathcal{M}(\boldsymbol{X}^{(k)})\right\}, \tag{40}$$

consisting of the all samples correctly classified by $\mathcal{M}$ and $\mathcal{C}$ in $\mathcal{D}$.

Eq. (37) implies a dynamic threshold $t_{\boldsymbol{X}}$ separating the positive and negative decision scores. Inspired by Eq. (17), we define uniform accuracy by using a unified threshold. Firstly, for given dataset $\mathcal{D}$ and model $\mathcal{M}$, classifier $\mathcal{C}$ with a fixed threshold $t$, we denote a subset of $\mathcal{D}$ as

$$\mathcal{D}(\mathcal{M}, \mathcal{C}; t) = \bigcup_{k=1}^{K}\left\{\boldsymbol{X}^{(k)} \in \mathcal{D}_k : \boldsymbol{w}_k\boldsymbol{h}^{(k)} - b_k > t \geq \max\left\{\boldsymbol{w}_j^T\boldsymbol{h}^{(k)} - b_j\right\}_{\substack{j=1 \\ j \neq k}}^{K}, \boldsymbol{h}^{(k)} = \mathcal{M}(\boldsymbol{X}^{(k)})\right\}$$

$$\tag{41}$$

which is the biggest subset of $\mathcal{D}$ uniformly classified by $\mathcal{M}$ and $\mathcal{C}$ with $t$. Then the ratio

$$\mathcal{A}_{\text{Uni}}(\mathcal{M}, \mathcal{C}; t) = \frac{|\mathcal{D}(\mathcal{M}, \mathcal{C}; t)|}{|\mathcal{D}|} \times 100\%, \tag{42}$$

is the corresponding uniform accuracy, and the maximum ratio with varying thresholds, i.e.,

$$\mathcal{A}_{\text{Uni}}(\mathcal{M}, \mathcal{C}) = \max_{t \in \mathbb{R}} \mathcal{A}_{\text{Uni}}(\mathcal{M}, \mathcal{C}; t), \tag{43}$$

is defined as the final **uniform accuracy**.

In practice, to calculate the uniform accuracy $\mathcal{A}_{\text{Uni}}$, the sets of positive and negative decision scores for the all samples

$$\mathcal{S}_{\text{pos}} = \bigcup_{k=1}^{K} \left\{ \boldsymbol{w}_k \boldsymbol{h}_i^{(k)} - b_k : i = 1, 2, \cdots, n_k \right\}, \tag{44}$$

$$\mathcal{S}_{\text{neg}} = \bigcup_{k=1}^{K} \bigcup_{\substack{j=1 \\ j \neq k}}^{K} \left\{ \boldsymbol{w}_j \boldsymbol{h}_i^{(k)} - b_j : i = 1, 2, \cdots, n_k \right\} \tag{45}$$

are first computed, and denote

$$s_{\text{pos-min}} = \min(\mathcal{S}_{\text{pos}}) \qquad \text{and} \qquad s_{\text{neg-max}} = \max(\mathcal{S}_{\text{pos}}). \tag{46}$$

If $s_{\text{pos-min}} \geq s_{\text{neg-max}}$, the classification accuracy $\mathcal{A}$ and the uniform one $\mathcal{A}_{\text{Uni}}$ must be 100%, otherwise, $N = 200$ thresholds $\{t_i\}_{i=1}^{N}$ are evenly taken from the interval $[s_{\text{pos-min}}, s_{\text{neg-max}}]$, and $N = 200$ uniform accuracy $\mathcal{A}_{\text{Uni}}(\mathcal{M}, \mathcal{C}; t_i)$ are figured out, while the best one $\max\left\{ A_{\text{Uni}}(\mathcal{M}, \mathcal{C}; t_i) \right\}_{i=1}^{N}$ is chosen as the final uniform accuracy $\mathcal{A}_{\text{Uni}}$. In this calculation, the final results will be slightly different when different numbers ($N$) of thresholds are taken in the score interval.

By Eqs. (17), a model with higher uniform accuracy, it would lead to more samples from category $k, \forall k \in [K]$, whose inner products (positive similarities/decision scores without bias) with the classifier vector $\boldsymbol{w}_k$ are greater than $b_k + t$, implying higher intra-class compactness in each category, and requires more samples whose inner products (negative similarities/decision scores without bias) with the classifier vectors of other categories are less than $b_j + t$, implying higher inter-class distinctiveness among all categories. For the intra-class **compactness** $\mathcal{E}_{\text{com}}$ and inter-class **distinctiveness** $\mathcal{E}_{\text{dis}}$ among sample features, we define them as

$$\mathcal{E}_{\text{com}} = \frac{1}{2}\left[ \frac{1}{K} \sum_{k=1}^{K} \left( \frac{1}{n_k^2} \sum_{i=1}^{n_k} \sum_{i'=1}^{n_k} \frac{\langle \boldsymbol{h}_i^{(k)} - \bar{\boldsymbol{h}}, \boldsymbol{h}_{i'}^{(k)} - \bar{\boldsymbol{h}} \rangle}{\|\boldsymbol{h}_i^{(k)} - \bar{\boldsymbol{h}}\| \|\boldsymbol{h}_{i'}^{(k)} - \bar{\boldsymbol{h}}\|} \right) + 1 \right], \tag{47}$$

$$\mathcal{E}_{\text{dis}} = \frac{1}{2}\left[ 1 - \frac{1}{K(K-1)} \sum_{k=1}^{K} \sum_{\substack{k'=1 \\ k' \neq k}}^{K} \left( \frac{1}{n_k} \frac{1}{n_{k'}} \sum_{i=1}^{n_k} \sum_{i'=1}^{n_{k'}} \frac{\langle \boldsymbol{h}_i^{(k)}, \boldsymbol{h}_{i'}^{(k')} \rangle}{\|\boldsymbol{h}_i^{(k)}\| \|\boldsymbol{h}_{i'}^{(k')}\|} \right) \right], \tag{48}$$

where $\bar{\boldsymbol{h}} = \frac{1}{|\mathcal{D}|} \sum_{k=1}^{K} \sum_{i=1}^{n_k} \boldsymbol{h}_i^{(k)}$ is the global feature center.

Due to the neural collapse, the compactness $\mathcal{E}_{\text{com}}$ might be higher than $\frac{1}{2} - \frac{1}{2(K-1)}$, and the distinctiveness $\mathcal{E}_{\text{dis}}$ might be lower than $\frac{1}{2} + \frac{1}{2(K-1)}$, for the model $\mathcal{M}$ and classifier $\mathcal{C}$ which have been well trained on the dataset $\mathcal{D}$.

## S-2 EXPERIMENTAL SETTINGS AND RESULTS

### S-2.1 EXPERIMENTAL SETTINGS

Table S-4: Experimental settings in our experiments.

| | | Neural collapse | | Classification | | | |
|---|---|---|---|---|---|---|---|
| | | setting-1 | setting-2 | setting-3 | setting-4 | setting-5 | setting-6 |
| Hyper-parameter | epochs | 100 | 100 | 100 | 100 | 100 | 100 |
| | optimizer | SGD | AdamW | SGD | AdamW | SGD | AdamW |
| | batch size | 128 | 128 | 128 | 128 | 128 | 128 |
| | learning rate | 0.01 | 0.001 | 0.01 | 0.001 | 0.01 | 0.001 |
| | learning rate decay | step | cosine | step | cosine | step | cosine |
| | weight decay $\lambda$ | ✗ | ✗ | $5 \times 10^{-4}$ | 0.05 | $5 \times 10^{-4}$ | 0.05 |
| | weight decay $\lambda_W$ | $5 \times 10^{-4}$ | $5 \times 10^{-4}$ | ✗ | ✗ | ✗ | ✗ |
| | weight decay $\lambda_H$ | $5 \times 10^{-4}$ | $5 \times 10^{-4}$ | ✗ | ✗ | ✗ | ✗ |
| | weight decay $\lambda_b$ | $5 \times 10^{-4}$ | $5 \times 10^{-4}$ | ✗ | ✗ | ✗ | ✗ |
| | warmup epochs | 0 | 0 | 0 | 0 | 0 | 0 |
| Data Aug. | random cropping | ✗ | ✗ | ✓ | ✓ | ✓ | ✓ |
| | horizontal flipping | ✗ | ✗ | 0.5 | 0.5 | 0.5 | 0.5 |
| | label smoothing | ✗ | ✗ | ✗ | ✗ | 0.1 | 0.1 |
| | mixup alpha | ✗ | ✗ | ✗ | ✗ | 0.8 | 0.8 |
| | cutmix alpha | ✗ | ✗ | ✗ | ✗ | 1.0 | 1.0 |
| | mixup prob. | ✗ | ✗ | ✗ | ✗ | 0.8 | 0.8 |
| | normalization | mean $= [0.4914, 0.4822, 0.4465]$, std $= [0.2023, 0.1994, 0.2010]$ | | | | | |

In Sec. 4, we train ResNet18, ResNet50, and DenseNet121 on MNIST, CIFAR10, and CIFAR100, respectively. Table S-4 shows the experimental settings. In default, we train the models using setting-1 and setting-2 in the experiments of neural collapse, and apply setting-3, setting-4, setting-5, and setting-6 in the experiments of classification.

Table S-5: The numerical results of the three models trained on MNIST, with $\lambda_W = \lambda_H = \lambda_b = 5 \times 10^{-4}$.

| | | MNIST | | | |
|---|---|---|---|---|---|
| | | SGD | | AdamW | |
| | | CE | BCE | CE | BCE |
| ResNet18 | $\hat{\rho}$ | 219.0960 | 407.1362 | 212.2180 | 357.9696 |
| | $s_{\text{pos}}$ | $5.6439 \pm 0.1437$ | $6.4008 \pm 0.1236$ | $5.6331 \pm 0.0120$ | $7.7460 \pm 0.0113$ |
| | $s_{\text{neg}}$ | $-0.6302 \pm 0.2073$ | $-3.4987 \pm 0.1137$ | $-0.6259 \pm 0.0127$ | $-2.1233 \pm 0.0291$ |
| | $\hat{b}$ | $-0.0074 \pm 0.0852$ | $2.2170 \pm 0.0308$ | $0.0001 \pm 0.0328$ | $3.5134 \pm 0.0337$ |
| | $\alpha(\hat{b})$ | — | $-0.0268$ | — | $-0.0086$ |
| | $\mathcal{A}/\mathcal{A}_{\text{Uni}}$ for training | 100.00/100.00 | 100.00/100.00 | 100.00/100.00 | 100.00/100.00 |
| | $\mathcal{A}/\mathcal{A}_{\text{Uni}}$ for testing | 99.43/99.31 | 99.59/99.52 | 99.62/99.57 | 99.65/99.61 |
| ResNet50 | $\hat{\rho}$ | 217.7276 | 396.7711 | 212.2304 | 357.2365 |
| | $s_{\text{pos}}$ | $5.6383 \pm 0.6400$ | $6.5393 \pm 1.6509$ | $5.6389 \pm 0.0380$ | $7.7706 \pm 0.0573$ |
| | $s_{\text{neg}}$ | $-0.6271 \pm 0.5978$ | $-3.2512 \pm 1.9658$ | $-0.6266 \pm 0.0220$ | $-2.1029 \pm 0.0429$ |
| | $\hat{b}$ | $0.0039 \pm 0.0733$ | $2.4674 \pm 0.0492$ | $0.0001 \pm 0.0328$ | $3.5322 \pm 0.0329$ |
| | $\alpha(\hat{b})$ | — | $-0.0217$ | — | $-0.0084$ |
| | $\mathcal{A}/\mathcal{A}_{\text{Uni}}$ for training | 99.68/99.64 | 99.79/99.76 | 100.00/100.00 | 100.00/100.00 |
| | $\mathcal{A}/\mathcal{A}_{\text{Uni}}$ for testing | 98.98/98.79 | 99.01/98.88 | 99.60/99.57 | 99.53/99.52 |
| DenseNet121 | $\hat{\rho}$ | 224.1426 | 414.7491 | 212.2337 | 355.5479 |
| | $s_{\text{pos}}$ | $5.5774 \pm 0.1217$ | $6.1977 \pm 0.0987$ | $5.6318 \pm 0.1132$ | $7.8030 \pm 0.0377$ |
| | $s_{\text{neg}}$ | $-0.6193 \pm 0.1221$ | $-3.6421 \pm 0.1048$ | $-0.6258 \pm 0.3427$ | $-2.0508 \pm 0.0314$ |
| | $\hat{b}$ | $0.0010 \pm 0.0570$ | $2.0705 \pm 0.0264$ | $0.0002 \pm 0.0324$ | $3.5767 \pm 0.0344$ |
| | $\alpha(\hat{b})$ | — | $-0.0302$ | — | $-0.0081$ |
| | $\mathcal{A}/\mathcal{A}_{\text{Uni}}$ for training | 100.00/99.99 | 100.00/100.00 | 99.63/99.62 | 100.00/100.00 |
| | $\mathcal{A}/\mathcal{A}_{\text{Uni}}$ for testing | 99.45/99.40 | 99.54/99.52 | 99.29/99.22 | 99.64/99.60 |

### S-2.2 EXPERIMENTAL RESULTS OF NEURAL COLLAPSE

In this section, we show the experimental results of neural collapse. Most of these results are calculated on the training data of the three datasets.

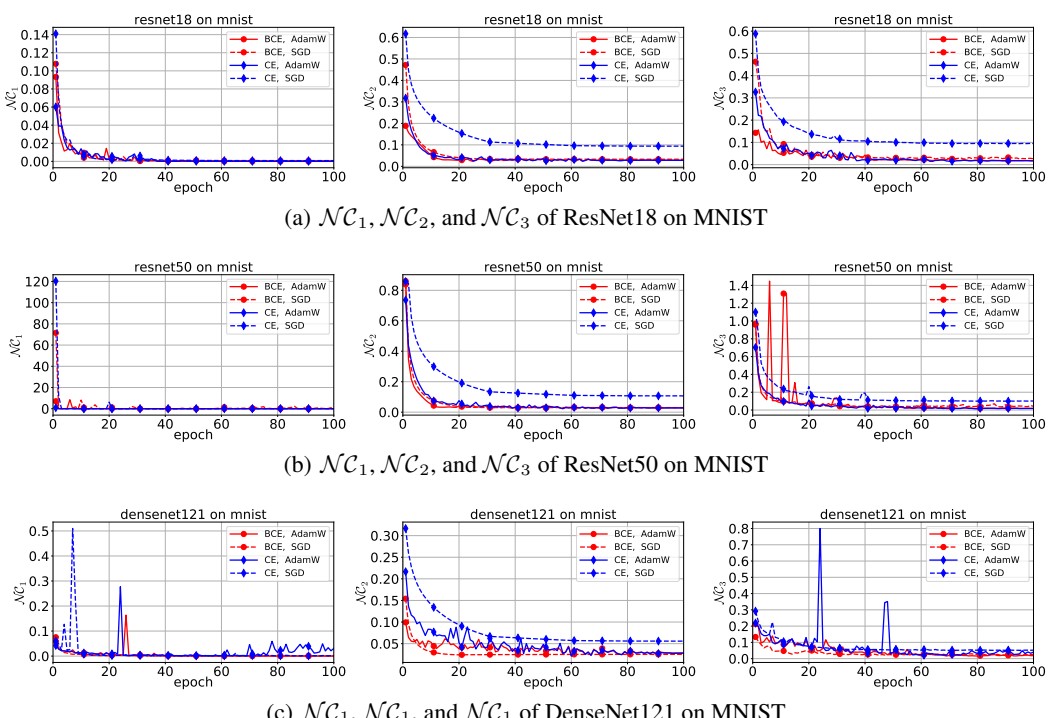

(a) $\mathcal{NC}_1, \mathcal{NC}_2$, and $\mathcal{NC}_3$ of ResNet18 on MNIST

(b) $\mathcal{NC}_1, \mathcal{NC}_2$, and $\mathcal{NC}_3$ of ResNet50 on MNIST

(c) $\mathcal{NC}_1, \mathcal{NC}_1$, and $\mathcal{NC}_1$ of DenseNet121 on MNIST

Figure S-5: The evolution of the three NC metrics in the training of ResNet18 (top), ResNet50 (middle), DenseNet121 (bottom) on MNIST with CE and BCE using SGD and AdamW, respectively, with $\lambda_{\boldsymbol{W}} = \lambda_{\boldsymbol{H}} = \lambda_{\boldsymbol{b}} = 5 \times 10^{-4}$.

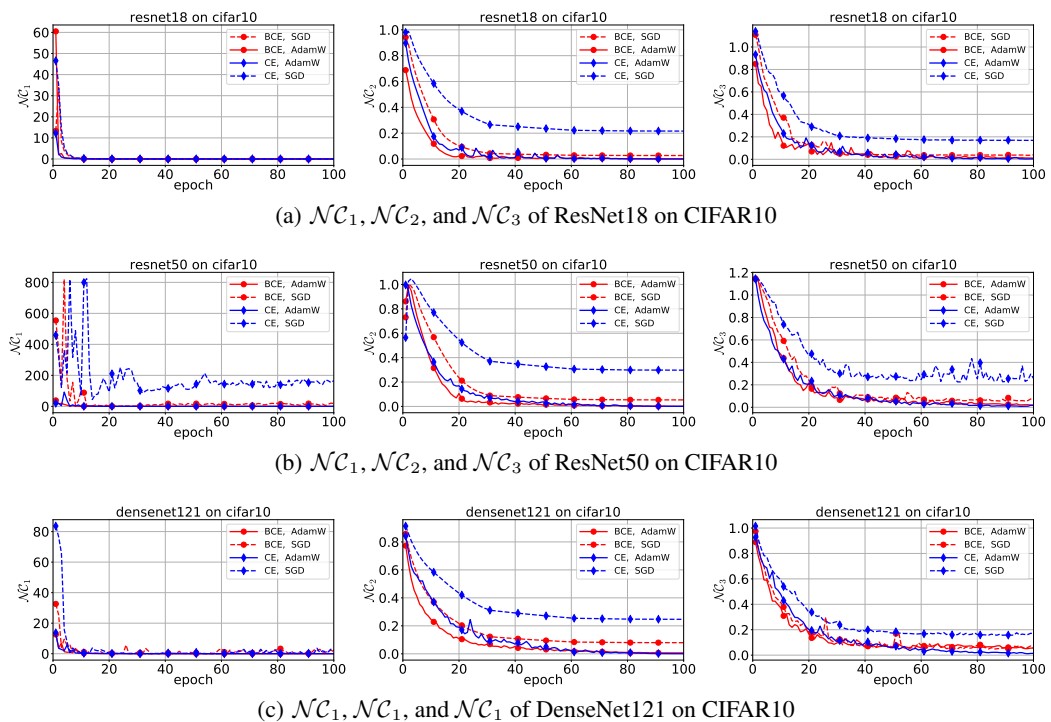

(a) $\mathcal{NC}_1, \mathcal{NC}_2$, and $\mathcal{NC}_3$ of ResNet18 on CIFAR10

(b) $\mathcal{NC}_1, \mathcal{NC}_2$, and $\mathcal{NC}_3$ of ResNet50 on CIFAR10

(c) $\mathcal{NC}_1, \mathcal{NC}_1$, and $\mathcal{NC}_1$ of DenseNet121 on CIFAR10

Figure S-6: The evolution of the three NC metrics in the training of ResNet18 (top), ResNet50 (middle), DenseNet121 (bottom) on CIFAR10 with CE and BCE using SGD and AdamW, respectively, with $\lambda_{\boldsymbol{W}} = \lambda_{\boldsymbol{H}} = \lambda_{\boldsymbol{b}} = 5 \times 10^{-4}$.

Table S-6: The numerical results of the three models trained on CIFAR10, with $\lambda_{\boldsymbol{W}} = \lambda_{\boldsymbol{H}} = \lambda_{\boldsymbol{b}} = 5 \times 10^{-4}$.

| | | CIFAR10 | | | |
| --- | --- | --- | --- | --- | --- |
| | | SGD | | AdamW | |
| | | CE | BCE | CE | BCE |
| ResNet18 | $\hat{\rho}$ | 221.7685 | 395.3918 | 212.4173 | 366.6813 |
| | $s_{\text{pos}}$ | $5.7103 \pm 0.2252$ | $6.5627 \pm 0.2042$ | $5.6393 \pm 0.0568$ | $7.5025 \pm 0.0549$ |
| | $s_{\text{neg}}$ | $-0.6386 \pm 0.3574$ | $-3.4557 \pm 0.1939$ | $-0.6265 \pm 0.0066$ | $-2.3582 \pm 0.0225$ |
| | $\hat{b}$ | $-0.0085 \pm 0.0430$ | $2.2618 \pm 0.0678$ | $-0.0001 \pm 0.0038$ | $3.2905 \pm 0.0080$ |
| | $\alpha(\hat{b})$ | — | $-0.0266$ | — | $-0.0105$ |
| | $\mathcal{A}/\mathcal{A}_{\text{Uni}}$ for training | 99.99/99.98 | 100.00/100.00 | 100.00/100.00 | 100.00/100.00 |
| | $\mathcal{A}/\mathcal{A}_{\text{Uni}}$ for testing | 79.22/75.71 | 81.19/78.78 | 86.66/84.72 | 86.58/85.07 |
| ResNet50 | $\hat{\rho}$ | 220.8594 | 382.4440 | 212.3374 | 369.2447 |
| | $s_{\text{pos}}$ | $5.7365 \pm 8.2056$ | $6.5614 \pm 4.3923$ | $5.6386 \pm 0.1062$ | $7.4351 \pm 0.2787$ |
| | $s_{\text{neg}}$ | $-0.6439 \pm 14.1340$ | $-3.5695 \pm 7.0134$ | $-0.6266 \pm 0.0150$ | $-2.4493 \pm 0.2165$ |
| | $\hat{b}$ | $0.0045 \pm 0.1430$ | $2.4002 \pm 0.1496$ | $-0.0000 \pm 0.0053$ | $3.2051 \pm 0.0309$ |
| | $\alpha(\hat{b})$ | — | $-0.0242$ | — | $-0.0114$ |
| | $\mathcal{A}/\mathcal{A}_{\text{Uni}}$ for training | 99.61/99.52 | 99.65/99.32 | 99.99/99.99 | 100.00/100.00 |
| | $\mathcal{A}/\mathcal{A}_{\text{Uni}}$ for testing | 76.28/73.08 | 78.41/76.35 | 85.73/84.33 | 85.76/84.98 |
| DenseNet121 | $\hat{\rho}$ | 225.0609 | 392.8198 | 212.7966 | 360.5613 |
| | $s_{\text{pos}}$ | $5.7225 \pm 1.7228$ | $6.2376 \pm 0.8437$ | $5.6150 \pm 0.2851$ | $7.6743 \pm 0.1239$ |
| | $s_{\text{neg}}$ | $-0.6348 \pm 0.8664$ | $-3.6171 \pm 1.6284$ | $-0.6240 \pm 0.0330$ | $-2.1715 \pm 0.0604$ |
| | $\hat{b}$ | $0.0012 \pm 0.0364$ | $2.0875 \pm 0.1229$ | $0.0003 \pm 0.0061$ | $3.4612 \pm 0.0203$ |
| | $\alpha(\hat{b})$ | — | $-0.0318$ | — | $-0.0090$ |
| | $\mathcal{A}/\mathcal{A}_{\text{Uni}}$ for training | 99.40/99.03 | 99.72/99.62 | 99.87/99.86 | 100.00/100.00 |
| | $\mathcal{A}/\mathcal{A}_{\text{Uni}}$ for testing | 77.30/74.41 | 79.16/77.95 | 81.54/80.15 | 82.34/81.70 |

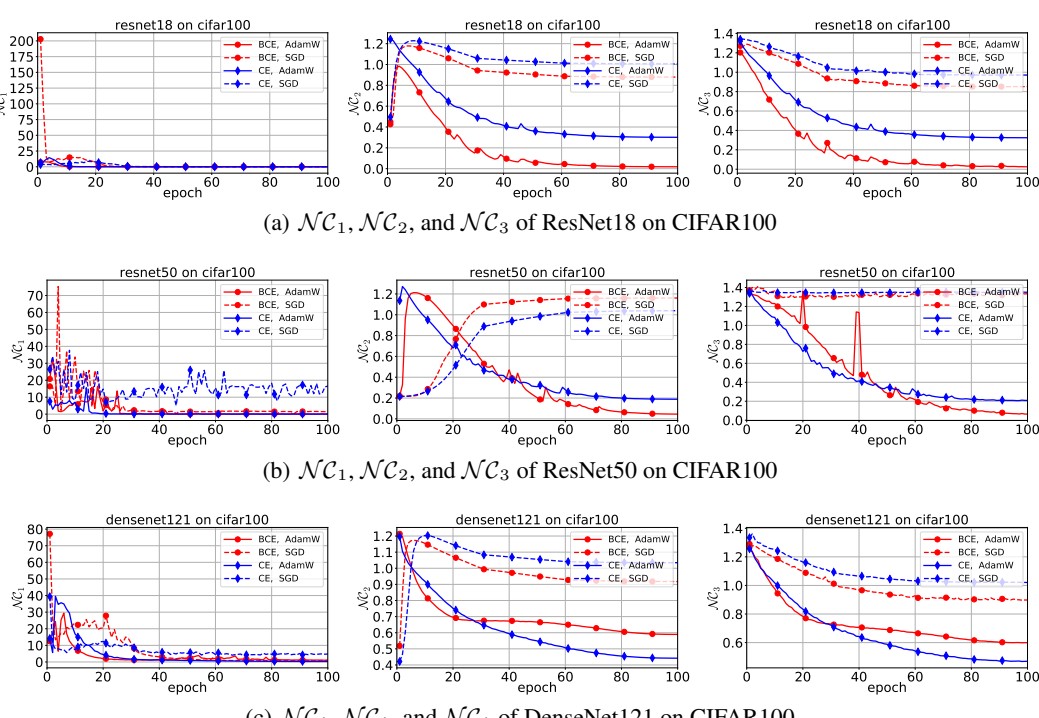

(a) $\mathcal{NC}_1$, $\mathcal{NC}_2$, and $\mathcal{NC}_3$ of ResNet18 on CIFAR100

(b) $\mathcal{NC}_1$, $\mathcal{NC}_2$, and $\mathcal{NC}_3$ of ResNet50 on CIFAR100

(c) $\mathcal{NC}_1$, $\mathcal{NC}_1$, and $\mathcal{NC}_1$ of DenseNet121 on CIFAR100

Figure S-7: The evolution of the three NC metrics in the training of ResNet18 (top), ResNet50 (middle), DenseNet121 (bottom) on CIFAR100 with CE and BCE using SGD and AdamW, respectively, with $\lambda_{\boldsymbol{W}} = \lambda_{\boldsymbol{H}} = \lambda_{\boldsymbol{b}} = 5 \times 10^{-4}$.

Table S-7: The numerical results of the three models trained on CIFAR100, with $\lambda_{\boldsymbol{W}} = \lambda_{\boldsymbol{H}} = \lambda_{\boldsymbol{b}} = 5 \times 10^{-4}$.

| | | CIFAR100 | | | |
| --- | --- | --- | --- | --- | --- |
| | | SGD | | AdamW | |
| | | CE | BCE | CE | BCE |
| ResNet18 | $\hat{\rho}$ | 954.3918 | 1732.6035 | 846.4734 | 1708.9231 |
| | $s_{\text{pos}}$ | $8.3613 \pm 0.4316$ | $3.5152 \pm 0.2392$ | $7.5183 \pm 0.0997$ | $4.0202 \pm 0.0696$ |
| | $s_{\text{neg}}$ | $-0.0848 \pm 1.3897$ | $-6.5934 \pm 1.2718$ | $-0.0754 \pm 0.2580$ | $-5.6834 \pm 0.0438$ |
| | $\hat{b}$ | $0.0004 \pm 0.2356$ | $0.8407 \pm 0.0678$ | $0.0005 \pm 0.0097$ | $1.1317 \pm 0.0007$ |
| | $\alpha(\hat{b})$ | — | $-0.2672$ | — | $-0.2147$ |
| | $\mathcal{A}/\mathcal{A}_{\text{Uni}}$ for training | 99.95/99.81 | 99.98/99.97 | 99.98/99.96 | 99.98/99.97 |
| | $\mathcal{A}/\mathcal{A}_{\text{Uni}}$ for testing | 34.61/17.99 | 42.06/30.61 | 56.58/47.29 | 60.48/43.04 |
| ResNet50 | $\hat{\rho}$ | 36.2794 | 289.5987 | 838.0098 | 1710.3754 |
| | $s_{\text{pos}}$ | $0.5404 \pm 9.8551$ | $-4.6656 \pm 16.0695$ | $7.3906 \pm 0.3560$ | $3.9356 \pm 1.5798$ |
| | $s_{\text{neg}}$ | $0.6182 \pm 11.4828$ | $-6.2663 \pm 29.8421$ | $-0.0745 \pm 0.1935$ | $-5.7441 \pm 1.1971$ |
| | $\hat{b}$ | $0.0006 \pm 0.0592$ | $0.3210 \pm 0.0241$ | $0.0005 \pm 0.0073$ | $1.1239 \pm 0.0044$ |
| | $\alpha(\hat{b})$ | — | $-0.4090$ | — | $-0.2160$ |
| | $\mathcal{A}/\mathcal{A}_{\text{Uni}}$ for training | 2.52/0.05 | 7.67/0.44 | 99.83/99.76 | 99.77/99.62 |
| | $\mathcal{A}/\mathcal{A}_{\text{Uni}}$ for testing | 2.48/0.06 | 7.16/0.39 | 55.51/50.77 | 53.55/49.18 |
| DenseNet121 | $\hat{\rho}$ | 894.4895 | 1597.8596 | 900.5263 | 1761.0126 |
| | $s_{\text{pos}}$ | $8.4473 \pm 0.8321$ | $3.0569 \pm 1.6496$ | $8.1030 \pm 0.4805$ | $4.0875 \pm 0.2246$ |
| | $s_{\text{neg}}$ | $-0.0842 \pm 1.6340$ | $-6.6552 \pm 2.6035$ | $-0.0800 \pm 0.4365$ | $-5.8613 \pm 0.7152$ |
| | $\hat{b}$ | $-0.0012 \pm 0.2239$ | $0.8313 \pm 0.0983$ | $0.0016 \pm 0.0948$ | $1.1306 \pm 0.0145$ |
| | $\alpha(\hat{b})$ | — | $-0.2714$ | — | $-0.2141$ |
| | $\mathcal{A}/\mathcal{A}_{\text{Uni}}$ for training | 99.15/94.38 | 99.38/99.23 | 99.80/99.78 | 99.98/99.97 |
| | $\mathcal{A}/\mathcal{A}_{\text{Uni}}$ for testing | 37.48/24.20 | 39.93/35.19 | 50.31/37.87 | 52.41/49.81 |

**NC metrics, the final classifier bias, and the final decision scores**. Figs. S-5 - S-7 shows the evolution of the three NC metrics in the training of ResNet18, ResNet50, DenseNet121 on MNIST, CIFAR10, and CIFAR100 with CE and BCE. In the training on MNIST and CIFAR10, the NC metrics of both CE and BCE approach zero at the terminal phase of training, and that of BCE decrease faster than that of CE at the first 20 epochs. In the training on CIFAR100, which is a more challenging dataset than MNIST and CIFAR10, the NC metrics of models trained by SGD do not decrease to zero, while that of models trained by AdamW approach zero, and the NC metrics of BCE decrease faster than that of CE in most cases. Table S-5 - S-7 present the numerical results of the final models at the 100th epoch. In these tables, $\hat{\rho} = \|\hat{\boldsymbol{W}}\|_F^2$, where $\hat{\boldsymbol{W}} = [\hat{\boldsymbol{w}}_1, \hat{\boldsymbol{w}}_2, \cdots, \hat{\boldsymbol{w}}_K]^T \in \mathbb{R}^{K \times d}$ is the final trained classifier weight; "$s_{\text{pos}}$" rows list the mean and standard deviations of the final positive decision scores without biases, i.e.,

$$\text{Mean}(s_{\text{pos}}) = \frac{1}{nK} \sum_{k=1}^{K} \sum_{i=1}^{n} \hat{\boldsymbol{w}}_k \boldsymbol{h}_i^{(k)}, \tag{49}$$

$$\text{Std}(s_{\text{pos}}) = \sqrt{\sum_{k=1}^{K} \sum_{i=1}^{n} \frac{\left(\hat{\boldsymbol{w}}_k \boldsymbol{h}_i^{(k)} - \text{Mean}(s_{\text{pos}})\right)^2}{nK}}, \tag{50}$$

"$s_{\text{neg}}$" rows list that of the final negative decision scores without biases, i.e.,

$$\text{Mean}(s_{\text{neg}}) = \frac{1}{nK(K-1)} \sum_{k=1}^{K} \sum_{\substack{j=1 \\ j \neq k}}^{K} \sum_{i=1}^{n} \hat{\boldsymbol{w}}_j \boldsymbol{h}_i^{(k)}, \tag{51}$$

$$\text{Std}(s_{\text{neg}}) = \sqrt{\sum_{k=1}^{K} \sum_{\substack{j=1 \\ j \neq k}}^{K} \sum_{i=1}^{n} \frac{\left(\hat{\boldsymbol{w}}_j \boldsymbol{h}_i^{(k)} - \text{Mean}(s_{\text{neg}})\right)^2}{nK(K-1)}}, \tag{52}$$

and "$\hat{b}$" rows list that of the final classifier bias $\hat{\boldsymbol{b}} = [\hat{b}_1, \hat{b}_2, \cdots, \hat{b}_K]^T \in \mathbb{R}^K$, i.e.,

$$\text{Mean}(\hat{b}) = \frac{1}{K} \sum_{k=1}^{K} \hat{b}_k, \tag{53}$$

$$\text{Std}(\hat{b}) = \sqrt{\frac{\sum_{k=1}^{K} \left( \hat{b}_k - \text{Mean}(\hat{b}) \right)^2}{K}}. \tag{54}$$

"$\alpha(\hat{b})$" rows list the value of function $\alpha(b)$ at point $\text{Mean}(\hat{b})$, where

$$\alpha(b) = -\frac{K-1}{K \left( 1 + \exp \left( b + \sqrt{\frac{\lambda_W}{n\lambda_H}} \frac{\rho}{K(K-1)} \right) \right)} + \frac{1}{K \left( 1 + \exp \left( \sqrt{\frac{\lambda_W}{n\lambda_H}} \frac{\rho}{K} - b \right) \right)} + \lambda_b b, \tag{55}$$

is the function at the RHS of Eq. (12).

Besides the classification accuracy $\mathcal{A}$ and uniform accuracy $\mathcal{A}_{\text{Uni}}$ of the final models on the training data, Tables S-5, S-6, and S-7 have also presented that on the testing data.

Table S-8: The numerical results of ResNet18 trained on MNIST with fixed weight decay $\lambda_b$ for the classifier bias.

| Loss | Opt. | $\bar{b}$ | $\hat{\rho}$ | $s_{\text{pos}}$ | $s_{\text{neg}}$ | $\hat{b}$ | $\alpha(\hat{b})$ |
|------|------|-----|----------|-------------------|--------------------|---------------------|-----------|
| CE | SGD | 0 | 218.9428 | $5.6648 \pm 0.1673$ | $-0.6323 \pm 0.2360$ | $-0.0179 \pm 0.1228$ | — |
| | | 1 | 218.8023 | $5.6337 \pm 0.1473$ | $-0.6290 \pm 0.2097$ | $0.9821 \pm 0.1149$ | — |
| | | 2 | 218.3450 | $5.6456 \pm 0.1556$ | $-0.6318 \pm 0.2213$ | $1.9821 \pm 0.1122$ | — |
| | | 3 | 218.3319 | $5.6399 \pm 0.1521$ | $-0.6295 \pm 0.2132$ | $2.9821 \pm 0.1163$ | — |
| | | 4 | 219.2994 | $5.6628 \pm 0.1600$ | $-0.6321 \pm 0.2281$ | $3.9820 \pm 0.1307$ | — |
| | | 5 | 219.5797 | $5.6611 \pm 0.1780$ | $-0.6329 \pm 0.2411$ | $4.9820 \pm 0.1279$ | — |
| | | 6 | 220.0522 | $5.6458 \pm 0.1598$ | $-0.6301 \pm 0.2245$ | $5.9820 \pm 0.1312$ | — |
| | | 8 | 219.4256 | $5.6410 \pm 0.1608$ | $-0.6311 \pm 0.2284$ | $7.9821 \pm 0.1194$ | — |
| | | 10 | 219.2911 | $5.6411 \pm 0.1601$ | $-0.6300 \pm 0.2152$ | $9.9821 \pm 0.1250$ | — |
| | AdamW | 0 | 212.2146 | $5.6360 \pm 0.0250$ | $-0.6262 \pm 0.0189$ | $-0.0180 \pm 0.0486$ | — |
| | | 1 | 212.2138 | $5.6355 \pm 0.0353$ | $-0.6262 \pm 0.0194$ | $0.9828 \pm 0.0493$ | — |
| | | 2 | 212.2151 | $5.6336 \pm 0.0258$ | $-0.6260 \pm 0.0189$ | $1.9821 \pm 0.0487$ | — |
| | | 3 | 212.2152 | $5.6336 \pm 0.0264$ | $-0.6260 \pm 0.0189$ | $2.9825 \pm 0.0486$ | — |
| | | 4 | 212.2161 | $5.6307 \pm 0.0274$ | $-0.6257 \pm 0.0191$ | $3.9823 \pm 0.0491$ | — |
| | | 5 | 212.2143 | $5.6308 \pm 0.0264$ | $-0.6257 \pm 0.0189$ | $4.9809 \pm 0.0486$ | — |
| | | 6 | 212.2143 | $5.6323 \pm 0.0264$ | $-0.6258 \pm 0.0189$ | $5.9822 \pm 0.0486$ | — |
| | | 8 | 212.2163 | $5.6347 \pm 0.0262$ | $-0.6261 \pm 0.0189$ | $7.9812 \pm 0.0486$ | — |
| | | 10 | 212.2151 | $5.6340 \pm 0.0263$ | $-0.6260 \pm 0.0189$ | $9.9829 \pm 0.0486$ | — |
| BCE | SGD | 0 | 393.2500 | $7.1748 \pm 0.1277$ | $-2.8219 \pm 0.1379$ | $3.0789 \pm 0.0489$ | $-0.0120$ |
| | | 1 | 374.9337 | $7.7515 \pm 0.1578$ | $-2.2877 \pm 0.1468$ | $3.6658 \pm 0.0709$ | $-0.0070$ |
| | | 2 | 362.5949 | $8.1822 \pm 0.1525$ | $-1.9121 \pm 0.1604$ | $4.1078 \pm 0.1053$ | $-0.0045$ |
| | | 3 | 355.2978 | $8.5608 \pm 0.1634$ | $-1.6192 \pm 0.1568$ | $4.4557 \pm 0.0981$ | $-0.0030$ |
| | | 4 | 354.6479 | $8.8711 \pm 0.1473$ | $-1.3347 \pm 0.1725$ | $4.7949 \pm 0.1094$ | $-0.0019$ |
| | | 5 | 355.9634 | $9.2305 \pm 0.1503$ | $-1.0452 \pm 0.1960$ | $5.1493 \pm 0.1192$ | $-0.0009$ |
| | | 6 | 361.1938 | $9.5688 \pm 0.1355$ | $-0.7519 \pm 0.1688$ | $5.5084 \pm 0.0869$ | $-0.0002$ |
| | | 8 | 385.6802 | $10.3761 \pm 0.1400$ | $-0.0997 \pm 0.2436$ | $6.3418 \pm 0.0989$ | $0.0007$ |
| | | 10 | 426.3013 | $11.5173 \pm 0.1430$ | $0.7786 \pm 0.3075$ | $7.4858 \pm 0.1021$ | $0.0010$ |
| | AdamW | 0 | 350.4272 | $9.3081 \pm 0.0352$ | $-1.0348 \pm 0.0321$ | $5.2388 \pm 0.0609$ | $-0.0006$ |
| | | 1 | 350.4283 | $9.3015 \pm 0.0345$ | $-1.0340 \pm 0.0321$ | $5.2389 \pm 0.0609$ | $-0.0006$ |
| | | 2 | 350.4292 | $9.3029 \pm 0.0357$ | $-1.0342 \pm 0.0321$ | $5.2388 \pm 0.0609$ | $-0.0006$ |
| | | 3 | 350.4275 | $9.3028 \pm 0.0364$ | $-1.0342 \pm 0.0321$ | $5.2388 \pm 0.0609$ | $-0.0006$ |
| | | 4 | 350.4248 | $9.3039 \pm 0.0362$ | $-1.0343 \pm 0.0320$ | $5.2388 \pm 0.0609$ | $-0.0006$ |
| | | 5 | 350.4250 | $9.3100 \pm 0.0358$ | $-1.0350 \pm 0.0320$ | $5.2388 \pm 0.0608$ | $-0.0006$ |
| | | 6 | 350.4302 | $9.3063 \pm 0.0345$ | $-1.0346 \pm 0.0321$ | $5.2388 \pm 0.0608$ | $-0.0006$ |
| | | 8 | 350.4304 | $9.3094 \pm 0.0356$ | $-1.0349 \pm 0.0321$ | $5.2389 \pm 0.0609$ | $-0.0006$ |
| | | 10 | 350.4330 | $9.3109 \pm 0.0369$ | $-1.0351 \pm 0.0321$ | $5.2388 \pm 0.0609$ | $-0.0006$ |

**The failures in the experiments of neural collapse**. According to the above figures and tables, one can find the models trained with SGD are easily to fail in the experiments of neural collapse, including the ResNet50 trained on MNIST, ResNet50 and DenseNet121 trained on CIFAR10, and the three models trained CIFAR100. The standard deviations of positive/negative decision scores produced by these models are usually larger than $0.5$. These failed models in the neural collapse can be roughly classified into two types:

- The two ResNet50 trained on CIFAR100 with SGD. They are completely failed models. The standard deviations of the decision scores are very high, even more than 20, and, for the

BCE-trained model, the means of the positive and negative decision scores are relatively close, while for the CE-trained model, the mean of positive scores is even less than that of negative ones, indicating that most of the samples were not correctly classified. The classification accuracy $\mathcal{A}$ on the training dataset are only $2.52\%$ and $7.67\%$ with CE and BCE.

- The other failed models trained with SGD, including the ResNet50 trained on MNIST and CIFAR10, DenseNet121 trained on CIFAR10, ResNet18 and DenseNet121 trained on CIFAR100. These models have achieved almost $100\%$ classification accuracy and uniform accuracy on the training dataset. However, according to the standard deviations of decision scores and the NC metrics, we conclude that they do not reach the state of neural collapse.

These failures in the experiments of neural collapse reveal more relationships among classification and neural collapse. In the training, zero classification error appears before zero uniform classification error, which appears before the neural collapse, or, in contrary, the model reaching the neural collapse has the uniform accuracy of $100\%$, and the model with the uniform accuracy of $100\%$ has also the accuracy $100\%$ on the classification. Both the reverses are not true.

Table S-9: The numerical results of ResNet18 trained on MNIST with varying weight decay $\lambda_b$ for the classifier bias.

| Loss | Opt. | $\lambda_b$ | $\hat{\rho}$ | $s_{\text{pos}}$ | $s_{\text{neg}}$ | $\hat{b}$ | $\alpha(\hat{b})$ |
|---|---|---|---|---|---|---|---|
| CE | SGD | $5 \times 10^{-1}$ | 218.6677 | $5.6511 \pm 0.1144$ | $-0.6304 \pm 0.1854$ | $-0.0000 \pm 0.0002$ | — |
| | | $5 \times 10^{-2}$ | 218.6658 | $5.6662 \pm 0.1176$ | $-0.6321 \pm 0.2031$ | $-0.0000 \pm 0.0017$ | — |
| | | $5 \times 10^{-3}$ | 218.5622 | $5.6427 \pm 0.1076$ | $-0.6296 \pm 0.1917$ | $0.0013 \pm 0.0156$ | — |
| | | $5 \times 10^{-4}$ | 219.4882 | $5.6527 \pm 0.1287$ | $-0.6322 \pm 0.2352$ | $4.0998 \pm 0.0796$ | — |
| | | $5 \times 10^{-5}$ | 219.0555 | $5.6526 \pm 0.1407$ | $-0.6310 \pm 0.2192$ | $9.1337 \pm 0.1038$ | — |
| | | $5 \times 10^{-6}$ | 219.2227 | $5.6426 \pm 0.1507$ | $-0.6307 \pm 0.2209$ | $9.8940 \pm 0.1111$ | — |
| | AdamW | $5 \times 10^{-1}$ | 212.2359 | $5.6329 \pm 0.0340$ | $-0.6259 \pm 0.0037$ | $-0.0000 \pm 0.0001$ | — |
| | | $5 \times 10^{-2}$ | 212.2369 | $5.6372 \pm 0.0335$ | $-0.6264 \pm 0.0037$ | $0.0000 \pm 0.0010$ | — |
| | | $5 \times 10^{-3}$ | 212.2328 | $5.6382 \pm 0.0186$ | $-0.6265 \pm 0.0038$ | $0.0000 \pm 0.0083$ | — |
| | | $5 \times 10^{-4}$ | 212.2152 | $5.6339 \pm 0.0257$ | $-0.6260 \pm 0.0128$ | $0.0010 \pm 0.0324$ | — |
| | | $5 \times 10^{-5}$ | 212.2158 | $5.6316 \pm 0.0221$ | $-0.6257 \pm 0.0174$ | $3.4803 \pm 0.0448$ | — |
| | | $5 \times 10^{-6}$ | 212.2147 | $5.6330 \pm 0.0256$ | $-0.6259 \pm 0.0186$ | $8.9169 \pm 0.0480$ | — |
| BCE | SGD | $5 \times 10^{-1}$ | 472.0906 | $4.2473 \pm 0.1306$ | $-5.6495 \pm 0.1260$ | $0.0036 \pm 0.0000$ | $-0.1683$ |
| | | $5 \times 10^{-2}$ | 471.6918 | $4.2916 \pm 0.1134$ | $-5.5975 \pm 0.1029$ | $0.0362 \pm 0.0003$ | $-0.1640$ |
| | | $5 \times 10^{-3}$ | 452.0422 | $4.6706 \pm 0.1199$ | $-5.1987 \pm 0.0936$ | $0.4031 \pm 0.0037$ | $-0.1269$ |
| | | $5 \times 10^{-4}$ | 358.9137 | $9.0244 \pm 0.1190$ | $-0.7897 \pm 0.1281$ | $4.8403 \pm 0.0604$ | $-0.0018$ |
| | | $5 \times 10^{-5}$ | 414.4364 | $11.0715 \pm 0.1306$ | $0.5388 \pm 0.2787$ | $7.0401 \pm 0.0959$ | $0.0008$ |
| | | $5 \times 10^{-6}$ | 424.8451 | $11.4847 \pm 0.1327$ | $0.7536 \pm 0.3067$ | $7.4372 \pm 0.0973$ | $0.0010$ |
| | AdamW | $5 \times 10^{-1}$ | 483.3321 | $4.2399 \pm 0.0308$ | $-5.6315 \pm 0.0215$ | $0.0036 \pm 0.0000$ | $-0.1636$ |
| | | $5 \times 10^{-2}$ | 482.1844 | $4.2698 \pm 0.0306$ | $-5.5977 \pm 0.0213$ | $0.0358 \pm 0.0003$ | $-0.1598$ |
| | | $5 \times 10^{-3}$ | 470.6640 | $4.5928 \pm 0.0281$ | $-5.2753 \pm 0.0201$ | $0.3577 \pm 0.0033$ | $-0.1256$ |
| | | $5 \times 10^{-4}$ | 356.5036 | $7.7870 \pm 0.0130$ | $-2.0822 \pm 0.0285$ | $3.5514 \pm 0.0330$ | $-0.0083$ |
| | | $5 \times 10^{-5}$ | 347.1199 | $9.0726 \pm 0.0303$ | $-1.1593 \pm 0.0304$ | $4.9903 \pm 0.0537$ | $-0.0012$ |
| | | $5 \times 10^{-6}$ | 350.0225 | $9.2915 \pm 0.0372$ | $-1.0489 \pm 0.0319$ | $5.2119 \pm 0.0599$ | $-0.0006$ |

**The bias decay parameter $\lambda_b$.** In Sec. 4, we conducted experiments with fixed $\lambda_b = 0$ and varying $\lambda_b = 0.5, 0.05, 5 \times 10^{-3}, 5 \times 10^{-4}, 5 \times 10^{-5}, 5 \times 10^{-6}$ to further compare CE and BCE in neural collapse. Fig. 3 have visually shown the results, and we here present the numerical results in Tables S-8 and S-9. In our experiments, the classifier weight $\boldsymbol{W}$ and bias $\boldsymbol{b}$ are initialized using "kaiming uniform", i.e., He initialization (He et al., 2015). The initialized classifier bias is with zero-mean, i.e., $\frac{1}{K} \sum_{k=1}^{K} b_k \approx 0$, and we add them with $0, 1, 2, 3, 4, 5, 6, 8, 10$, respectively, to adjust their average value in the experiments with fixed $\lambda_b$.

**The batch size.** In the proof of Theorem 1 and 2, it applied the feature matrix $\boldsymbol{H}$ including the features of all samples, to explore the the lower bounds for the CE and BCE losses, i.e.,

$$\boldsymbol{H} = \left[ h_1^{(1)}, h_2^{(1)}, \cdots, h_n^{(1)}, h_1^{(2)}, h_2^{(2)}, \cdots, h_n^{(2)}, \cdots, h_1^{(K)}, h_2^{(K)}, \cdots, h_n^{(K)} \right]. \tag{56}$$

However, batch algorithm was applied in the practical training of deep models, and the batch size would affect the experimental numerical results. To verify this conclusion, a group of experiments

were conducted with varying batch size. We trained ResNet18 on MNIST using SGD and AdamW using setting-1 and setting-2, while the initial learning rates were adjusted according to the batch size, $0.01 \times \frac{\text{batch size}}{128}$ for SGD and $0.001 \times \frac{\text{batch size}}{128}$ for AdamW. Fig. S-8 visually shows the distributions of the final classifier bias and the positive/negative decision scores, and Table S-10 lists the final numerical results. From these results, one can find that the bias results still conform to our analysis when batch size $\leq 1024$, i.e., the classifier bias converges to zero in the training with CE loss and $\lambda_b > 0$, and the clssifier bias separates the positive and negative decision scores in the training with BCE loss.

Table S-10: The numerical results of ResNet18 trained on MNIST with varying batch size and $\lambda_W = \lambda_H = \lambda_b = 5 \times 10^{-4}$.

| Loss | Opt. | batch size | $\hat{\rho}$ | $s_{\text{pos}}$ | $s_{\text{neg}}$ | $\hat{b}$ | $\alpha(\hat{b})$ |
|---|---|---|---|---|---|---|---|
| CE | SGD | 16 | 100.9731 | $6.7176 \pm 0.3270$ | $-0.7538 \pm 0.1950$ | $-0.0074 \pm 0.0523$ | — |
| | | 32 | 130.1404 | $6.3375 \pm 0.2425$ | $-0.7110 \pm 0.1709$ | $-0.0074 \pm 0.0478$ | — |
| | | 64 | 168.6290 | $6.0159 \pm 0.1562$ | $-0.6737 \pm 0.2052$ | $-0.0074 \pm 0.0547$ | — |
| | | 128 | 219.0960 | $5.6439 \pm 0.1437$ | $-0.6302 \pm 0.2073$ | $-0.0074 \pm 0.0852$ | — |
| | | 256 | 285.6314 | $5.3200 \pm 0.1586$ | $-0.5936 \pm 0.2070$ | $-0.0074 \pm 0.1259$ | — |
| | | 512 | 379.3403 | $4.9776 \pm 0.2735$ | $-0.5535 \pm 0.2921$ | $-0.0073 \pm 0.2526$ | — |
| | | 1024 | 522.5523 | $4.6562 \pm 1.3926$ | $-0.5173 \pm 0.8343$ | $-0.0073 \pm 1.0641$ | — |
| | | 2048 | 473.7898 | $3.5759 \pm 2.6771$ | $-0.3972 \pm 2.0373$ | $-0.0072 \pm 1.8399$ | — |
| | AdamW | 16 | 87.6451 | $6.5511 \pm 0.0110$ | $-0.7279 \pm 0.0089$ | $0.0003 \pm 0.0211$ | — |
| | | 32 | 118.0328 | $6.2558 \pm 0.0101$ | $-0.6951 \pm 0.0104$ | $0.0003 \pm 0.0253$ | — |
| | | 64 | 158.4980 | $5.9506 \pm 0.0106$ | $-0.6612 \pm 0.0117$ | $0.0002 \pm 0.0293$ | — |
| | | 128 | 212.2180 | $5.6331 \pm 0.0120$ | $-0.6259 \pm 0.0127$ | $0.0001 \pm 0.0328$ | — |
| | | 256 | 282.9370 | $5.3168 \pm 0.0148$ | $-0.5908 \pm 0.0133$ | $0.0000 \pm 0.0357$ | — |
| | | 512 | 375.4274 | $4.9968 \pm 0.0209$ | $-0.5552 \pm 0.0140$ | $-0.0001 \pm 0.0380$ | — |
| | | 1024 | 496.6912 | $4.6627 \pm 0.0631$ | $-0.5199 \pm 0.0238$ | $-0.0190 \pm 0.0472$ | — |
| | | 2048 | 668.3063 | $4.3236 \pm 0.3703$ | $-0.4906 \pm 0.2909$ | $-0.0153 \pm 0.2964$ | — |
| BCE | SGD | 16 | 199.6890 | $6.1841 \pm 0.3002$ | $-5.9379 \pm 0.2665$ | $0.7828 \pm 0.0223$ | $-0.0660$ |
| | | 32 | 255.9898 | $6.1508 \pm 0.2184$ | $-5.2761 \pm 0.1932$ | $1.1506 \pm 0.0214$ | $-0.0546$ |
| | | 64 | 324.7408 | $6.2846 \pm 0.1600$ | $-4.4319 \pm 0.1295$ | $1.6456 \pm 0.0254$ | $-0.0399$ |
| | | 128 | 407.1362 | $6.4008 \pm 0.1236$ | $-3.4987 \pm 0.1137$ | $2.2170 \pm 0.0308$ | $-0.0268$ |
| | | 256 | 501.1286 | $6.6422 \pm 0.1347$ | $-2.5493 \pm 0.1501$ | $2.8605 \pm 0.0740$ | $-0.0167$ |
| | | 512 | 631.7796 | $6.6413 \pm 0.2725$ | $-1.9155 \pm 0.2544$ | $3.2338 \pm 0.1859$ | $-0.0127$ |
| | | 1024 | 816.6544 | $6.3274 \pm 0.4653$ | $-1.5393 \pm 0.4515$ | $3.3466 \pm 0.3554$ | $-0.0119$ |
| | | 2048 | 351.9647 | $1.7449 \pm 2.4487$ | $-0.5243 \pm 1.6982$ | $2.6332 \pm 1.5391$ | $0.0077$ |
| | AdamW | 16 | 189.2794 | $6.5169 \pm 0.0240$ | $-5.3841 \pm 0.0215$ | $1.2651 \pm 0.0119$ | $-0.0457$ |
| | | 32 | 242.1592 | $6.7110 \pm 0.0169$ | $-4.5302 \pm 0.0202$ | $1.7885 \pm 0.0167$ | $-0.0322$ |
| | | 64 | 300.8807 | $7.1079 \pm 0.0118$ | $-3.4518 \pm 0.0229$ | $2.5261 \pm 0.0234$ | $-0.0188$ |
| | | 128 | 357.9696 | $7.7460 \pm 0.0113$ | $-2.1233 \pm 0.0291$ | $3.5134 \pm 0.0337$ | $-0.0086$ |
| | | 256 | 455.2137 | $7.6247 \pm 0.0112$ | $-1.6013 \pm 0.0256$ | $3.8010 \pm 0.0325$ | $-0.0068$ |
| | | 512 | 590.9918 | $7.2831 \pm 0.0271$ | $-1.3210 \pm 0.0270$ | $3.8500 \pm 0.0375$ | $-0.0064$ |
| | | 1024 | 790.8874 | $6.6204 \pm 0.1011$ | $-1.3148 \pm 0.1126$ | $3.5830 \pm 0.0899$ | $-0.0089$ |
| | | 2048 | 1019.6438 | $5.9625 \pm 0.2969$ | $-1.2607 \pm 0.2750$ | $3.3303 \pm 0.2111$ | $-0.0122$ |

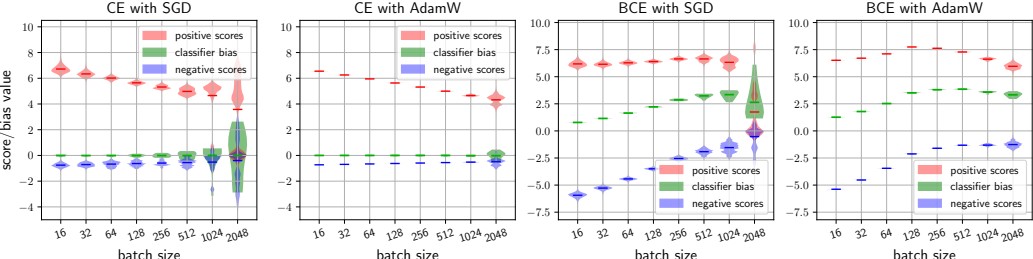

Figure S-8: The distributions of the final classifier bias and positive/negative decision scores for ResNet18 trained on MNIST with different batch sizes, while $\lambda_W = \lambda_H = \lambda_b = 5 \times 10^{-4}$.

The decision score results are very different from that in the experiments with fixed batch size. For examples, in the training with CE loss and fixed batch size $= 128$, the positive and negative decision scores converge to about $5.64$ and $-0.63$, respectively, and the value of $\hat{\rho} = \|\hat{W}\|_F^2$ converge to

about 219 and 212 in the training by SGD and AdamW, respectively, as shown in Tables S-8 and S-9. In contrast, these values varies as the batch size in the experiments with varying batch sizes.

In addition, the positive/negative decision scores did not converge to the theoretical values in Eq. (13) in our experiments; we believe it is resulted from the difference between the batch algorithm and the proof of Theorems. We roughly replaced $n$ with $\frac{\text{batch size}}{K}$ in computing $\alpha(\hat{b})$.

## S-2.3 EXPERIMENTAL RESULTS OF CLASSIFICATION

In the experiments of classification in Sec. 4.2, we train the models for 100 epochs. In each training, the model with best classification accuracy $\mathcal{A}$ is chosen as the final model, which was used to compute the uniform accuracy $\mathcal{A}_{\text{Uni}}$ presented in Table 2. In Table S-11 and S-12, we list their numerical results on the training and test dataset of CIFAR10 and CIFAR100. In these experiments, though the classification accuracy $\mathcal{A}$ of some models on the training datasets have reached $100\%$, neural collapse has not caused during the training. An obvious evidence is that both positive and negative decision scores have not converged, with large standard deviations, whether on the training set or testing set. The small standard deviations of the final classification bias might be more resulted from their initialization.

From Tables S-11 and S-12, one can find that, the gaps between the means of positive and negative decision scores of BCE-trained models are usually larger than that of CE-trained models, while in some cases, the standard deviations of the positive/negative decision scores of BCE-trained models are higher than that of CE-trained models. However, without any modification, the standard deviations and the gap between the positive and negative means cannot be precisely used to evaluate the intra-class compactness and inter-class distinctiveness. The decision score is calculated by the norm of the classifier vector and the feature vector, with the angle between them. The diverse $\hat{\rho}$ of CE-trained and BCE-trained models indicates different norms of the classifier vectors.

In Fig. 4, the all features are first projected into 2-dimension space from $d$-dimension space, and $d = 1024$ for ResNet50, which are then translated and scaled into the region of $[0,1] \times [0,1]$. We finally plot these feature points on the 2D plane.

Table S-11: The numerical results of ResNet18, ResNet50, DenseNet121 trained on CIFAR10 for classification.

| $\mathcal{M}$ | Op. | DA | Loss | classifier | | on training data | | | | on testing data | |
|---|---|---|---|---|---|---|---|---|---|---|---|
| | | | | $\hat{\rho}$ | $\hat{b}$ | $s_{\text{pos}}$ | $s_{\text{neg}}$ | $\mathcal{A}$ | $\mathcal{A}_{\text{Uni}}$ | $s_{\text{pos}}$ | $s_{\text{neg}}$ |
| ResNet18 | SGD | 1 | CE | 34.86 | $-0.01 \pm 0.03$ | $14.9 \pm 3.54$ | $-1.68 \pm 2.64$ | 99.98 | 97.55 | $13.9 \pm 4.73$ | $-1.56 \pm 2.89$ |
| | | | BCE | 52.33 | $2.89 \pm 0.03$ | $12.9 \pm 2.75$ | $-9.70 \pm 2.67$ | 100.00 | 99.99 | $11.3 \pm 4.93$ | $-9.25 \pm 3.30$ |
| | | 1&2 | CE | 12.59 | $-0.01 \pm 0.02$ | $3.23 \pm 0.38$ | $-0.37 \pm 0.62$ | 98.02 | 95.99 | $3.09 \pm 0.61$ | $-0.35 \pm 0.70$ |
| | | | BCE | 19.66 | $2.84 \pm 0.02$ | $3.86 \pm 0.45$ | $-0.86 \pm 0.66$ | 98.71 | 98.01 | $3.66 \pm 0.80$ | $-0.84 \pm 0.77$ |
| | AdamW | 1 | CE | 85.52 | $-0.00 \pm 0.01$ | $12.3 \pm 3.60$ | $-12.4 \pm 4.12$ | 99.99 | 99.57 | $10.9 \pm 5.52$ | $-12.1 \pm 4.58$ |
| | | | BCE | 113.9 | $2.16 \pm 0.02$ | $16.3 \pm 2.95$ | $-20.0 \pm 4.50$ | 100.00 | 100.00 | $14.2 \pm 6.37$ | $-19.0 \pm 5.75$ |
| | | 1&2 | CE | 36.26 | $-0.01 \pm 0.01$ | $2.54 \pm 0.18$ | $-1.13 \pm 0.38$ | 99.96 | 99.88 | $2.41 \pm 0.52$ | $-1.12 \pm 0.50$ |
| | | | BCE | 44.16 | $2.14 \pm 0.01$ | $3.57 \pm 0.20$ | $-1.74 \pm 0.38$ | 99.96 | 99.94 | $3.34 \pm 0.81$ | $-1.72 \pm 0.56$ |
| ResNet50 | SGD | 1 | CE | 18.74 | $0.00 \pm 0.03$ | $17.4 \pm 3.16$ | $-2.00 \pm 3.30$ | 99.99 | 98.09 | $16.1 \pm 4.56$ | $-1.86 \pm 3.73$ |
| | | | BCE | 29.07 | $2.83 \pm 0.04$ | $13.7 \pm 2.33$ | $-12.4 \pm 3.07$ | 99.99 | 99.98 | $11.9 \pm 5.08$ | $-11.8 \pm 3.83$ |
| | | 1&2 | CE | 8.18 | $0.00 \pm 0.04$ | $3.28 \pm 0.35$ | $-0.39 \pm 0.56$ | 98.25 | 96.65 | $3.14 \pm 0.63$ | $-0.37 \pm 0.66$ |
| | | | BCE | 13.86 | $2.65 \pm 0.03$ | $3.68 \pm 0.45$ | $-1.08 \pm 0.61$ | 98.79 | 98.24 | $3.47 \pm 0.85$ | $-1.06 \pm 0.75$ |
| | AdamW | 1 | CE | 143.9 | $0.01 \pm 0.02$ | $16.7 \pm 5.64$ | $-18.6 \pm 6.76$ | 100.00 | 98.95 | $14.9 \pm 7.87$ | $-18.2 \pm 7.23$ |
| | | | BCE | 153.4 | $2.20 \pm 0.01$ | $21.9 \pm 6.82$ | $-28.5 \pm 9.09$ | 99.97 | 99.96 | $19.4 \pm 10.1$ | $-27.2 \pm 10.4$ |
| | | 1&2 | CE | 79.80 | $0.00 \pm 0.01$ | $2.44 \pm 0.25$ | $-1.16 \pm 0.33$ | 99.96 | 99.89 | $2.28 \pm 0.57$ | $-1.16 \pm 0.44$ |
| | | | BCE | 102.6 | $2.14 \pm 0.00$ | $3.35 \pm 0.24$ | $-1.58 \pm 0.44$ | 99.95 | 99.94 | $3.16 \pm 0.72$ | $-1.55 \pm 0.56$ |
| DenseNet121 | SGD | 1 | CE | 48.02 | $0.00 \pm 0.02$ | $10.5 \pm 2.37$ | $-1.16 \pm 2.18$ | 99.30 | 93.29 | $9.57 \pm 3.41$ | $-1.06 \pm 2.41$ |
| | | | BCE | 64.94 | $2.93 \pm 0.03$ | $9.06 \pm 1.76$ | $-6.05 \pm 1.74$ | 99.45 | 99.24 | $7.75 \pm 3.63$ | $-5.71 \pm 2.32$ |
| | | 1&2 | CE | 14.99 | $0.00 \pm 0.02$ | $2.89 \pm 0.67$ | $-0.32 \pm 0.67$ | 91.20 | 86.80 | $2.77 \pm 0.80$ | $-0.30 \pm 0.71$ |
| | | | BCE | 19.60 | $2.86 \pm 0.02$ | $3.69 \pm 0.88$ | $-0.65 \pm 0.73$ | 92.38 | 90.28 | $3.52 \pm 1.08$ | $-0.62 \pm 0.80$ |
| | AdamW | 1 | CE | 139.4 | $0.00 \pm 0.01$ | $10.2 \pm 3.06$ | $-10.6 \pm 4.44$ | 99.97 | 98.48 | $8.70 \pm 5.00$ | $-10.4 \pm 4.86$ |
| | | | BCE | 156.6 | $2.17 \pm 0.01$ | $13.1 \pm 2.78$ | $-15.1 \pm 4.22$ | 99.97 | 99.97 | $10.9 \pm 5.93$ | $-14.4 \pm 5.13$ |
| | | 1&2 | CE | 39.93 | $0.00 \pm 0.01$ | $2.31 \pm 0.28$ | $-1.28 \pm 0.48$ | 98.83 | 98.10 | $2.14 \pm 0.64$ | $-1.26 \pm 0.58$ |
| | | | BCE | 40.53 | $2.18 \pm 0.01$ | $3.40 \pm 0.42$ | $-1.65 \pm 0.52$ | 98.81 | 98.51 | $3.13 \pm 0.94$ | $-1.60 \pm 0.67$ |

Table S-12: The numerical results of ResNet18, ResNet50, DenseNet121 trained on CIFAR100 for classification.

| $\mathcal{M}$ | Opt. | DA | Loss | classifier | | on training data | | | | on testing data | |
|---|---|---|---|---|---|---|---|---|---|---|---|
| | | | | $\hat{\rho}$ | $\hat{b}$ | $s_{\text{pos}}$ | $s_{\text{neg}}$ | $\mathcal{A}$ | $\mathcal{A}_{\text{Uni}}$ | $s_{\text{pos}}$ | $s_{\text{neg}}$ |
| ResNet18 | SGD | 1 | CE | 317.6 | $0.00 \pm 0.02$ | $15.8 \pm 3.15$ | $-0.18 \pm 3.04$ | 99.79 | 76.32 | $13.0 \pm 5.06$ | $-0.15 \pm 3.06$ |
| | | | BCE | 408.8 | $2.89 \pm 0.02$ | $9.39 \pm 2.87$ | $-10.0 \pm 2.94$ | 99.94 | 99.69 | $5.28 \pm 5.95$ | $-9.64 \pm 3.06$ |
| | | 1&2 | CE | 138.8 | $0.00 \pm 0.02$ | $5.82 \pm 1.33$ | $-0.07 \pm 0.98$ | 88.26 | 73.67 | $5.09 \pm 1.67$ | $-0.06 \pm 1.00$ |
| | | | BCE | 163.7 | $2.89 \pm 0.01$ | $3.42 \pm 1.44$ | $-3.22 \pm 0.99$ | 88.56 | 80.23 | $2.64 \pm 1.90$ | $-3.19 \pm 1.02$ |
| | AdamW | 1 | CE | 1007. | $0.00 \pm 0.02$ | $12.5 \pm 4.18$ | $-13.4 \pm 5.16$ | 99.98 | 92.02 | $7.47 \pm 7.81$ | $-13.1 \pm 5.19$ |
| | | | BCE | 1372. | $2.14 \pm 0.02$ | $15.3 \pm 4.85$ | $-21.2 \pm 6.34$ | 99.98 | 99.97 | $7.05 \pm 10.2$ | $-19.7 \pm 6.47$ |
| | | 1&2 | CE | 476.9 | $0.00 \pm 0.02$ | $4.49 \pm 0.82$ | $-2.04 \pm 0.99$ | 99.25 | 95.86 | $3.15 \pm 1.77$ | $-2.14 \pm 1.09$ |
| | | | BCE | 576.1 | $2.18 \pm 0.02$ | $3.67 \pm 0.80$ | $-4.13 \pm 0.84$ | 99.18 | 98.25 | $2.22 \pm 1.84$ | $-4.01 \pm 0.96$ |
| ResNet50 | SGD | 1 | CE | 258.8 | $0.00 \pm 0.01$ | $17.7 \pm 3.12$ | $-0.19 \pm 3.56$ | 99.90 | 79.70 | $14.5 \pm 5.17$ | $-0.16 \pm 3.58$ |
| | | | BCE | 328.3 | $2.87 \pm 0.01$ | $10.0 \pm 2.82$ | $-11.6 \pm 3.40$ | 99.86 | 99.62 | $5.37 \pm 6.20$ | $-10.9 \pm 3.46$ |
| | | 1&2 | CE | 102.7 | $0.00 \pm 0.01$ | $5.97 \pm 1.41$ | $-0.07 \pm 1.07$ | 87.46 | 72.45 | $5.29 \pm 1.71$ | $-0.06 \pm 1.05$ |
| | | | BCE | 118.4 | $2.86 \pm 0.01$ | $3.59 \pm 1.48$ | $-3.33 \pm 0.98$ | 89.17 | 81.80 | $2.75 \pm 1.96$ | $-3.29 \pm 1.02$ |
| | AdamW | 1 | CE | 2157. | $0.00 \pm 0.01$ | $13.9 \pm 5.49$ | $-19.4 \pm 7.18$ | 99.98 | 87.42 | $8.29 \pm 9.45$ | $-19.2 \pm 7.20$ |
| | | | BCE | 2863. | $2.15 \pm 0.02$ | $17.6 \pm 4.92$ | $-25.7 \pm 7.34$ | 99.98 | 99.97 | $8.49 \pm 11.2$ | $-23.5 \pm 7.67$ |
| | | 1&2 | CE | 1334. | $0.00 \pm 0.02$ | $4.42 \pm 0.67$ | $-1.96 \pm 0.87$ | 99.69 | 97.86 | $3.06 \pm 1.90$ | $-2.25 \pm 1.07$ |
| | | | BCE | 1440. | $2.18 \pm 0.02$ | $3.81 \pm 0.82$ | $-4.27 \pm 0.80$ | 99.67 | 99.22 | $2.28 \pm 1.96$ | $-4.21 \pm 0.93$ |
| DenseNet121 | SGD | 1 | CE | 337.9 | $-0.00 \pm 0.02$ | $12.9 \pm 3.33$ | $-0.12 \pm 2.83$ | 92.46 | 56.75 | $10.5 \pm 4.75$ | $-0.10 \pm 2.82$ |
| | | | BCE | 383.8 | $2.95 \pm 0.02$ | $6.03 \pm 2.64$ | $-7.83 \pm 2.81$ | 92.85 | 87.12 | $3.36 \pm 4.28$ | $-7.34 \pm 2.91$ |
| | | 1&2 | CE | 143.2 | $-0.00 \pm 0.02$ | $4.62 \pm 1.67$ | $-0.04 \pm 1.01$ | 67.23 | 47.68 | $4.26 \pm 1.84$ | $-0.04 \pm 1.01$ |
| | | | BCE | 161.5 | $2.90 \pm 0.01$ | $2.14 \pm 1.68$ | $-2.95 \pm 1.04$ | 68.15 | 56.43 | $1.74 \pm 1.85$ | $-2.93 \pm 1.05$ |
| | AdamW | 1 | CE | 1090. | $-0.00 \pm 0.01$ | $9.39 \pm 3.69$ | $-12.3 \pm 4.98$ | 99.89 | 83.67 | $4.74 \pm 6.83$ | $-12.1 \pm 4.99$ |
| | | | BCE | 1146. | $2.17 \pm 0.01$ | $9.78 \pm 2.72$ | $-16.0 \pm 4.77$ | 99.86 | 99.55 | $3.66 \pm 7.20$ | $-14.6 \pm 5.03$ |
| | | 1&2 | CE | 430.2 | $-0.00 \pm 0.01$ | $3.82 \pm 1.13$ | $-2.00 \pm 1.00$ | 91.18 | 80.57 | $2.85 \pm 1.83$ | $-2.07 \pm 1.06$ |
| | | | BCE | 474.5 | $2.20 \pm 0.01$ | $2.70 \pm 1.16$ | $-3.85 \pm 0.89$ | 90.66 | 85.83 | $1.79 \pm 1.83$ | $-3.82 \pm 0.97$ |

## S-3 PROOF OF THEOREM 2

Zhou et al. (2022) have proved that the loss satisfying contrastive property can cause neural collapse. CE loss, focal loss, and label smoothing loss satisfy this property, while BCE does not, and we proof that BCE can also result in the neural collapse in this paper.

**Definition S-1** *(Contrastive property (Zhou et al., 2022)). A loss function $\mathcal{L}(\boldsymbol{z})$ satisfies the contrastive property if there exists a function $\phi$ such that $\mathcal{L}(\boldsymbol{z})$ can be lower bounded by*

$$\mathcal{L}(\boldsymbol{z}) \geq \phi\bigg( \sum_{\substack{j=1 \\ j\neq k}}^{K} \big( z_j - z_k \big) \bigg) \tag{57}$$

*where the equality holds only when $z_j = z_\ell$ for $\forall j, \ell \neq k$, and the function $\phi(x)$ satisfies*

$$x^\star = \arg\min_x \phi(x) + c|x| \tag{58}$$

*is unique for $\forall c > 0$, and $x^\star \leq 0$.* ∎

**Theorem S-3** *(Zhou et al., 2022) Assume that the feature dimension $d$ is larger than the category number $K$, i.e., $d \geq K - 1$, and $\mathcal{L}$ is satisfying the contrastive property. Then any global minimizer $(\boldsymbol{W}^\star, \boldsymbol{H}^\star, \boldsymbol{b}^\star)$ of $f(\boldsymbol{W}, \boldsymbol{H}, \boldsymbol{b})$ defined using $\mathcal{L}$ with Eq. (3) obeys the following properties,*

$$\|\boldsymbol{w}^\star\| = \|\boldsymbol{w}_1^\star\| = \|\boldsymbol{w}_2^\star\| = \cdots = \|\boldsymbol{w}_K^\star\|, \tag{59}$$

$$\boldsymbol{h}_i^{(k)\star} = \sqrt{\frac{\lambda_{\boldsymbol{W}}}{n\lambda_{\boldsymbol{H}}}} \boldsymbol{w}_k^\star, \; \forall \, k \in [K], \; i \in [n], \tag{60}$$

$$\tilde{\boldsymbol{h}}_i^\star := \frac{1}{K} \sum_{j=1}^{K} \boldsymbol{h}_i^{(k)\star} = \boldsymbol{0}, \forall \, i \in [n], \tag{61}$$

$$\boldsymbol{b}^\star = b^\star \mathbf{1}_K, \tag{62}$$

*where either $b^\star = 0$ or $\lambda_{\boldsymbol{b}} = 0$. The matrix $\boldsymbol{W}^{\star T}$ forms a $K$-simplex ETF in the sense that*

$$\frac{1}{\|\boldsymbol{w}^\star\|_2^2} \boldsymbol{W}^{\star T} \boldsymbol{W}^\star = \frac{K}{K-1} \Big( \boldsymbol{I}_K - \frac{1}{K} \mathbf{1}_K \mathbf{1}_K^T \Big), \tag{63}$$

*where $\boldsymbol{I}_K \in \mathbb{R}^{K \times K}$ denotes the identity matrix, $\mathbf{1}_K \in \mathbb{R}^K$ denotes the all ones vector.* ∎

**Theorem S-4** *Assume that the feature dimension $d$ is larger than the number of classes $K$, i.e., $d \geq K - 1$. Then any global minimizer $(\boldsymbol{W}^\star, \boldsymbol{H}^\star, \boldsymbol{b}^\star)$ of*

$$\min_{\boldsymbol{W}, \boldsymbol{H}, \boldsymbol{b}} f_{\text{bce}}(\boldsymbol{W}, \boldsymbol{H}, \boldsymbol{b}) := g_{\text{bce}}(\boldsymbol{W}\boldsymbol{H} - \boldsymbol{b}\mathbf{1}^T) + \frac{\lambda_{\boldsymbol{W}}}{2} \|\boldsymbol{W}\|_F^2 + \frac{\lambda_{\boldsymbol{H}}}{2} \|\boldsymbol{H}\|_F^2 + \frac{\lambda_{\boldsymbol{b}}}{2} \|\boldsymbol{b}\|_2^2 \tag{64}$$

*with*

$$g_{\text{bce}}(\boldsymbol{W}\boldsymbol{H} - \boldsymbol{b}\mathbf{1}^T) := \frac{1}{N} \sum_{k=1}^{K} \sum_{i=1}^{n} \mathcal{L}_{\text{bce}}(\boldsymbol{W}\boldsymbol{h}_i^{(k)} - \boldsymbol{b}, \boldsymbol{y}_k), \tag{65}$$

*obeys the following*

$$\|\boldsymbol{w}^\star\| = \|\boldsymbol{w}_1^\star\| = \|\boldsymbol{w}_2^\star\| = \cdots = \|\boldsymbol{w}_K^\star\|, \;\; \text{and} \;\; \boldsymbol{b}^\star = b^\star \mathbf{1}, \tag{66}$$

$$\boldsymbol{h}_i^{(k)\star} = \sqrt{\frac{\lambda_{\boldsymbol{W}}}{n\lambda_{\boldsymbol{H}}}} \boldsymbol{w}_k^\star, \; \forall \, k \in [K], \; i \in [n], \;\; \text{and} \;\; \tilde{\boldsymbol{h}}_i^\star := \frac{1}{K} \sum_{j=1}^{K} \boldsymbol{h}_i^{(k)\star} = \boldsymbol{0}, \forall \, i \in [n], \tag{67}$$

*and the matrix $\frac{1}{\|\boldsymbol{w}^\star\|_2} \boldsymbol{W}^{\star T}$ forms a $K$-simplex ETF in the sense that*

$$\frac{1}{\|\boldsymbol{w}^\star\|_2^2} \boldsymbol{W}^{\star T} \boldsymbol{W}^\star = \frac{K}{K-1} \Big( \boldsymbol{I}_K - \frac{1}{K} \mathbf{1}_K \mathbf{1}_K^T \Big), \tag{68}$$

*where $b^\star$ is the solution of equation*

$$\lambda_b b = \frac{K-1}{K\left(1 + \exp\left(b + \sqrt{\frac{\lambda_W}{n\lambda_H}}\frac{\rho}{K(K-1)}\right)\right)} - \frac{1}{K\left(1 + \exp\left(\sqrt{\frac{\lambda_W}{n\lambda_H}}\frac{\rho}{K} - b\right)\right)}. \tag{69}$$

**Proof**  *According to Lemma 1, any critical point $(\boldsymbol{W}, \boldsymbol{H}, \boldsymbol{b})$ of $f(\boldsymbol{W}, \boldsymbol{H}, \boldsymbol{b})$ satisfies*

$$\boldsymbol{W}^T\boldsymbol{W} = \frac{\lambda_H}{\lambda_W}\boldsymbol{H}^T\boldsymbol{H}. \tag{70}$$

*Let $\rho = \|\boldsymbol{W}\|_F^2$ for any critical point $(\boldsymbol{W}, \boldsymbol{H}, \boldsymbol{b})$. Then, according to Lemma 3, for any $c_1, c_2 \geq 0$,*

$$f_{\text{bce}}(\boldsymbol{W}, \boldsymbol{H}, \boldsymbol{b})$$

$$\geq \left[\lambda_W - \left(\frac{2K-1}{N(1+c_2)} - \frac{1}{N(1+c_1)}\right)\sqrt{\frac{n\lambda_W}{\lambda_H}}\right]\rho - \frac{1}{2K\lambda_b}\left(\frac{K-1}{1+c_2} - \frac{1}{1+c_1}\right)^2 + C \tag{71}$$

*where*

$$C = \frac{c_1}{1+c_1}\log\left(\frac{1+c_1}{c_1}\right) + \frac{\log(1+c_1)}{1+c_1} + \frac{K-1}{1+c_2}\left[c_2\log\left(\frac{1+c_2}{c_2}\right) + \log(1+c_2)\right]. \tag{72}$$

*According to Lemma 4, the inequality (71) achieves its equality when*

$$\|\boldsymbol{w}_1\| = \|\boldsymbol{w}_2\| = \cdots = \|\boldsymbol{w}_K\|, \quad and \quad \boldsymbol{b} = b^\star\mathbf{1}, \tag{73}$$

$$\boldsymbol{h}_i^{(k)} = \sqrt{\frac{\lambda_W}{n\lambda_H}}\boldsymbol{w}_k, \; \forall\, k \in [K], \; i \in [n], \quad and \quad \tilde{\boldsymbol{h}}_i = \frac{1}{K}\sum_{k=1}^K \boldsymbol{h}_i^{(k)} = \mathbf{0}, \forall\, i \in [n], \tag{74}$$

$$\boldsymbol{W}\boldsymbol{W}^T = \frac{\rho}{K-1}\left(\boldsymbol{I}_K - \frac{1}{K}\mathbf{1}_K\mathbf{1}_K^T\right), \tag{75}$$

$$c_1 = \exp\left(\sqrt{\frac{\lambda_W}{n\lambda_H}}\frac{\rho}{K} - b^\star\right), \quad and \quad c_2 = \exp\left(b^\star + \sqrt{\frac{\lambda_W}{n\lambda_H}}\frac{\rho}{K(K-1)}\right), \tag{76}$$

*where $b^\star$ is the solution of equation*

$$\lambda_b b = \frac{K-1}{K\left(1 + \exp\left(b + \sqrt{\frac{\lambda_W}{n\lambda_H}}\frac{\rho}{K(K-1)}\right)\right)} - \frac{1}{K\left(1 + \exp\left(\sqrt{\frac{\lambda_W}{n\lambda_H}}\frac{\rho}{K} - b\right)\right)}. \tag{77}$$

*According to Lemma 5, the equation (77) in terms of $b$ has only one solution $b^\star$.*

*Given $\lambda_W, \lambda_H, \lambda_b > 0$, $f_{\text{bce}}(\boldsymbol{W}, \boldsymbol{H}, \boldsymbol{b})$ is convex function, which achieves its minimum with finite $\boldsymbol{W}, \boldsymbol{H}, \boldsymbol{b}$. Therefore, the right side of inequality (71) is a consistent when $\lambda_W, \lambda_H, \lambda_b$ are fixed and Eqs. (73, 74, 75, 76) hold, which finishes the proof.* ∎

**Lemma 1**  *Any critical point $(\boldsymbol{W}, \boldsymbol{H}, \boldsymbol{b})$ of Eq. (64) obeys*

$$\boldsymbol{W}^T\boldsymbol{W} = \frac{\lambda_H}{\lambda_W}\boldsymbol{H}\boldsymbol{H}^T, \quad and \quad \|\boldsymbol{W}\|_F^2 = \frac{\lambda_H}{\lambda_W}\|\boldsymbol{H}\|_F^2. \tag{78}$$

**Proof**  *See Lemma D.2 in reference (Zhu et al., 2021).* ∎

**Lemma 2**  *For any $\boldsymbol{h}_i^{(k)}$ with $c_1, c_2 > 0$, the BCE loss is lower bounded by*

$$\mathcal{L}_{\text{bce}}(\boldsymbol{W}\boldsymbol{h}_i^{(k)}, \boldsymbol{y}_k) \geq \frac{1}{1+c_1}\left(-\boldsymbol{w}_k^T\boldsymbol{h}_i^{(k)} + b_k\right) + \frac{1}{1+c_2}\sum_{\substack{j=1\\j\neq k}}^K\left(\boldsymbol{w}_j^T\boldsymbol{h}_i^{(k)} - b_j\right) + C, \tag{79}$$

*where*

$$C = \frac{c_1}{1+c_1}\log\left(\frac{1+c_1}{c_1}\right) + \frac{\log(1+c_1)}{1+c_1} + \frac{K-1}{1+c_2}\left[c_2\log\left(\frac{1+c_2}{c_2}\right) + \log(1+c_2)\right]. \tag{80}$$

The inequality becomes an equality when

$$\boldsymbol{w}_j^T \boldsymbol{h}_i^{(k)} - b_j = \boldsymbol{w}_\ell^T \boldsymbol{h}_i^{(k)} - b_\ell, \ \ \forall j, \ell \neq k, \tag{81}$$

and

$$c_1 = \exp\left(\boldsymbol{w}_k^T \boldsymbol{h}_i^{(k)} - b_k\right), \tag{82}$$

$$c_2 = \exp\left(b_j - \boldsymbol{w}_j^T \boldsymbol{h}_i^{(k)}\right), \ \ j \neq k. \tag{83}$$

**Proof** *By the concavity of the* $\log(1 + e^x)$, *we have,*

$$\sum_{k=1}^K \log\left(1 + \exp(x_k)\right) \geq K \log\left(1 + \exp\left(\frac{\sum_{k=1}^K x_k}{K}\right)\right), \ \ \forall x_k \in \mathbb{R}. \tag{84}$$

*Then,*

$$\mathcal{L}_{\text{bce}}(\boldsymbol{W}\boldsymbol{h}_i^{(k)} + \boldsymbol{b}, \boldsymbol{y}_k) \tag{85}$$

$$= \log\left(1 + \exp(-\boldsymbol{w}_k^T \boldsymbol{h}_i^{(k)} + b_k)\right) + \sum_{\substack{j=1 \\ j \neq k}}^K \log\left(1 + \exp(\boldsymbol{w}_j^T \boldsymbol{h}_i^{(k)} - b_j)\right) \tag{86}$$

$$\geq \log\left(1 + \exp(-\boldsymbol{w}_k^T \boldsymbol{h}_i^{(k)} + b_k)\right) + (K-1)\log\left[1 + \exp\left(\frac{\sum_{\substack{j=1 \\ j \neq k}}^K \left(\boldsymbol{w}_j^T \boldsymbol{h}_i^{(k)} - b_j\right)}{K-1}\right)\right] \tag{87}$$

$$= \log\left(\frac{c_1}{1+c_1}\frac{1+c_1}{c_1} + \frac{1+c_1}{1+c_1}\exp\left(-\boldsymbol{w}_k^T \boldsymbol{h}_i^{(k)} + b_k\right)\right)$$

$$+ (K-1)\log\left[\frac{c_2}{1+c_2}\frac{1+c_2}{c_2} + \frac{1+c_2}{1+c_2}\exp\left(\frac{\sum_{\substack{j=1 \\ j \neq k}}^K \left(\boldsymbol{w}_j^T \boldsymbol{h}_i^{(k)} - b_j\right)}{K-1}\right)\right] \tag{88}$$

$$\geq \frac{c_1}{1+c_1}\log\left(\frac{1+c_1}{c_1}\right) + \frac{1}{1+c_1}\log\left((1+c_1)\exp\left(-\boldsymbol{w}_k^T \boldsymbol{h}_i^{(k)} + b_k\right)\right)$$

$$+ (K-1)\left\{\frac{c_2}{1+c_2}\log\left(\frac{1+c_2}{c_2}\right) + \frac{1}{1+c_2}\log\left[(1+c_2)\exp\left(\frac{\sum_{\substack{j=1 \\ j \neq k}}^K \left(\boldsymbol{w}_j^T \boldsymbol{h}_i^{(k)} - b_j\right)}{K-1}\right)\right]\right\} \tag{89}$$

$$= \frac{1}{1+c_1}\left(-\boldsymbol{w}_k^T \boldsymbol{h}_i^{(k)} + b_k\right) + \frac{1}{1+c_2}\sum_{\substack{j=1 \\ j \neq k}}^K \left(\boldsymbol{w}_j^T \boldsymbol{h}_i^{(k)} - b_j\right)$$

$$+ \underbrace{\frac{c_1}{1+c_1}\log\left(\frac{1+c_1}{c_1}\right) + \frac{\log(1+c_1)}{1+c_1} + \frac{K-1}{1+c_2}\left[c_2\log\left(\frac{1+c_2}{c_2}\right) + \log(1+c_2)\right]}_{C}. \tag{90}$$

*The first inequality is derived from the concavity of* $\log(1 + e^x)$, *i.e., Eq. (84), which achieves the equality if and only if*

$$\boldsymbol{w}_j^T \boldsymbol{h}_i^{(k)} - b_j = \boldsymbol{w}_\ell^T \boldsymbol{h}_i^{(k)} - b_\ell, \ \ \forall j, \ell \neq k \in [K]. \tag{91}$$

*The second inequality is derived from the concavity of* $\log(x)$,

$$\log\left(tx_1 + (1-t)x_2\right) \geq t\log(x_1) + (1-t)\log(x_2), \ \ \forall x_1, x_2 \in \mathbb{R} \ \text{and} \ t \in [0,1], \tag{92}$$

*which achieves its equality if and only if* $x_1 = x_2$, *or* $t = 0$, *or* $t = 1$. *Then, the second inequality holds for any* $c_1, c_2 \geq 0$, *and it becomes an equality if and only if*

$$\frac{1+c_1}{c_1} = (1+c_1)\exp\left(-\boldsymbol{w}_k^T \boldsymbol{h}_i^{(k)} + b_k\right) \ \text{or} \ c_1 = 0 \ \text{or} \ c_1 = +\infty, \ \text{and} \tag{93}$$

$$\frac{1+c_2}{c_2} = (1+c_2)\exp\left(\frac{\sum_{\substack{j=1 \\ j \neq k}}^K \left(\boldsymbol{w}_j^T \boldsymbol{h}_i^{(k)} - b_j\right)}{K-1}\right) \ \text{or} \ c_1 = 0 \ \text{or} \ c_1 = +\infty. \tag{94}$$

*It is trivial when $c_1 = 0$ or $c_1 = +\infty$ or $c_2 = 0$ or $c_2 = +\infty$. Then, we get*

$$c_1 = \exp\left(\boldsymbol{w}_k^T \boldsymbol{h}_i^{(k)} - b_k\right), \tag{95}$$

$$c_2 = \exp\left(\frac{\sum_{\substack{j=1 \\ j \neq k}}^{K} \left(b_j - \boldsymbol{w}_j^T \boldsymbol{h}_i^{(k)}\right)}{K-1}\right) \stackrel{(91)}{=} \exp\left(b_j - \boldsymbol{w}_j^T \boldsymbol{h}_i^{(k)}\right), \; j \neq k, \tag{96}$$

*which are desired.* ∎

**Lemma 3** *Let*

$$\boldsymbol{W} = \left[\boldsymbol{w}_1, \boldsymbol{w}_2, \cdots, \boldsymbol{w}_K\right]^T \in \mathbb{R}^{K \times d}, \tag{97}$$

$$\boldsymbol{H} = \left[h_1^{(1)}, \cdots, h_n^{(1)}, \cdots, h_1^{(K)}, \cdots, h_n^{(K)}\right] \in \mathbb{R}^{d \times N} \tag{98}$$

*with $N = nK$. Then, for any critical point $(\boldsymbol{W}, \boldsymbol{H}, \boldsymbol{b})$ of Eq. (64) and any $c_1, c_2 \geq 0$, we have*

$$f_{\text{bce}}(\boldsymbol{W}, \boldsymbol{H}, \boldsymbol{b})$$

$$\geq \left[\lambda_{\boldsymbol{W}} - \left(\frac{1}{N(1+c_2)} + \frac{1}{N(1+c_1)}\right)\sqrt{\frac{n\lambda_{\boldsymbol{W}}}{\lambda_{\boldsymbol{H}}}}\right]\rho - \frac{1}{2K\lambda_{\boldsymbol{b}}}\left(\frac{K-1}{1+c_2} - \frac{1}{1+c_1}\right)^2 + C \tag{99}$$

*with $C = \frac{c_1}{1+c_1}\log\left(\frac{1+c_1}{c_1}\right) + \frac{\log(1+c_1)}{1+c_1} + \frac{K-1}{1+c_2}\left[c_2\log\left(\frac{1+c_2}{c_2}\right) + \log(1+c_2)\right]$.*

**Proof** *According to Lemma 1, Eq. (79) holds for any $c_1, c_2 > 0$ and any $\boldsymbol{h}_i^{(k)}$ with $k \in [K]$, $i \in [n]$. We take the same $c_1$ and $c_2$ for all $\boldsymbol{h}_i^{(k)}$, then*

$$(1+c_1)(1+c_2)\left[g_{\text{bce}}(\boldsymbol{W}\boldsymbol{H} + \boldsymbol{b}\boldsymbol{1}^T) - C\right] \tag{100}$$

$$= (1+c_1)(1+c_2)\left[\frac{1}{N}\sum_{k=1}^{K}\sum_{i=1}^{n}\mathcal{L}_{\text{bce}}(\boldsymbol{W}\boldsymbol{h}_i^{(k)} + \boldsymbol{b}, \boldsymbol{y}_k) - C\right] \tag{101}$$

$$\geq \frac{1}{N}\sum_{k=1}^{K}\sum_{i=1}^{n}\left[(1+c_2)\left(-\boldsymbol{w}_k^T\boldsymbol{h}_i^{(k)} + b_k\right) + (1+c_1)\sum_{\substack{j=1 \\ j \neq k}}^{K}\left(\boldsymbol{w}_j^T\boldsymbol{h}_i^{(k)} - b_j\right)\right] \tag{102}$$

$$= \frac{1+c_1}{N}\sum_{k=1}^{K}\sum_{i=1}^{n}\sum_{\substack{j=1 \\ j \neq k}}^{K}\left(\boldsymbol{w}_j^T\boldsymbol{h}_i^{(k)} - b_j\right) - \frac{1+c_2}{N}\sum_{k=1}^{K}\sum_{i=1}^{n}\left(\boldsymbol{w}_k^T\boldsymbol{h}_i^{(k)} - b_k\right) \tag{103}$$

$$= \frac{1+c_1}{N}\sum_{k=1}^{K}\sum_{i=1}^{n}\left(\sum_{j=1}^{K}\left(\boldsymbol{w}_j^T\boldsymbol{h}_i^{(k)} - b_j\right) - \boldsymbol{w}_k^T\boldsymbol{h}_i^{(k)} + b_k\right) - \frac{1+c_2}{N}\sum_{k=1}^{K}\sum_{i=1}^{n}\left(\boldsymbol{w}_k^T\boldsymbol{h}_i^{(k)} - b_k\right) \tag{104}$$

$$= \frac{1+c_1}{N}\sum_{k=1}^{K}\sum_{i=1}^{n}\sum_{j=1}^{K}\left(\boldsymbol{w}_j^T\boldsymbol{h}_i^{(k)} - b_j - \boldsymbol{w}_k^T\boldsymbol{h}_i^{(k)} + b_k\right) + \frac{1+c_1}{N}\sum_{k=1}^{K}\sum_{i=1}^{n}\sum_{\substack{j=1 \\ j \neq k}}^{K}\left(\boldsymbol{w}_k^T\boldsymbol{h}_i^{(k)} - b_k\right)$$

$$- \frac{1+c_2}{N}\sum_{k=1}^{K}\sum_{i=1}^{n}\left(\boldsymbol{w}_k^T\boldsymbol{h}_i^{(k)} - b_k\right) \tag{105}$$

$$= \frac{1+c_1}{N}\left[\sum_{k=1}^{K}\sum_{i=1}^{n}\sum_{j=1}^{K}\left(\boldsymbol{w}_j^T\boldsymbol{h}_i^{(k)} - b_j\right) - \sum_{k=1}^{K}\sum_{i=1}^{n}\sum_{j=1}^{K}\left(\boldsymbol{w}_k^T\boldsymbol{h}_i^{(k)} - b_k\right)\right]$$

$$+ \left(\frac{1+c_1}{N}(K-1) - \frac{1+c_2}{N}\right)\sum_{k=1}^{K}\sum_{i=1}^{n}\boldsymbol{w}_k^T\boldsymbol{h}_i^{(k)} - \left(\frac{1+c_1}{N}(K-1) - \frac{1+c_2}{N}\right)\sum_{k=1}^{K}\sum_{i=1}^{n}b_k \tag{106}$$

$$= \frac{1+c_1}{N}\sum_{i=1}^{n}\left[\sum_{k=1}^{K}\left(\sum_{j=1}^{K}\boldsymbol{w}_k^T\boldsymbol{h}_i^{(j)} - K\boldsymbol{w}_k^T\boldsymbol{h}_i^{(k)}\right) - \underbrace{\sum_{k=1}^{K}\sum_{j=1}^{K}b_j + \sum_{k=1}^{K}\sum_{j=1}^{K}b_k}_{0}\right]$$

$$+ \left( \frac{1 + c_1}{N} (K - 1) - \frac{1 + c_2}{N} \right) \sum_{k=1}^{K} \sum_{i=1}^{n} \boldsymbol{w}_k^T \boldsymbol{h}_i^{(k)} - \left( \frac{1 + c_1}{K} (K - 1) - \frac{1 + c_2}{K} \right) \sum_{k=1}^{K} b_k \tag{107}$$

$$= \frac{1 + c_1}{n} \sum_{i=1}^{n} \sum_{k=1}^{K} \boldsymbol{w}_k^T \left( \tilde{\boldsymbol{h}}_i - \boldsymbol{h}_i^{(k)} \right) + \left( \frac{1 + c_1}{N} (K - 1) - \frac{1 + c_2}{N} \right) \sum_{k=1}^{K} \sum_{i=1}^{n} \boldsymbol{w}_k^T \boldsymbol{h}_i^{(k)}$$

$$- \left( \frac{1 + c_1}{K} (K - 1) - \frac{1 + c_2}{K} \right) \sum_{k=1}^{K} b_k \tag{108}$$

where $\tilde{\boldsymbol{h}}_i = \frac{1}{K} \sum_{k=1}^{K} \boldsymbol{h}_i^{(k)}$.

*According to the AM-GM inequality, we have*

$$\boldsymbol{u}^T \boldsymbol{v} \geq -\frac{c}{2} \|\boldsymbol{u}\|_2^2 - \frac{1}{2c} \|\boldsymbol{v}\|_2^2, \ \forall \ \boldsymbol{u}, \ \boldsymbol{v} \in \mathbb{R}^d, \ \forall \ c \geq 0. \tag{109}$$

*Then,*

$$(1 + c_1)(1 + c_2) \left[ g_{\text{bce}}(\boldsymbol{W}\boldsymbol{H} + \boldsymbol{b}\mathbf{1}^T) - C \right]$$

$$\geq -\frac{1 + c_1}{n} \left( \frac{c_3}{2} \sum_{i=1}^{n} \sum_{k=1}^{K} \|\boldsymbol{w}_k\|_2^2 + \frac{1}{2c_3} \sum_{i=1}^{n} \sum_{k=1}^{K} \left\| \tilde{\boldsymbol{h}}_i - \boldsymbol{h}_i^{(k)} \right\|_2^2 \right)$$

$$- \left( \frac{1 + c_1}{N} (K - 1) - \frac{1 + c_2}{N} \right) \left( \frac{c_4}{2} \sum_{k=1}^{K} \sum_{i=1}^{n} \|\boldsymbol{w}_k\|_2^2 + \frac{1}{2c_4} \sum_{k=1}^{K} \sum_{i=1}^{n} \|\boldsymbol{h}_i^{(k)}\|_2^2 \right)$$

$$- \left( \frac{1 + c_1}{K} (K - 1) - \frac{1 + c_2}{K} \right) \sum_{k=1}^{K} b_k \tag{110}$$

$$= -\frac{1 + c_1}{n} \left[ \frac{c_3}{2} \sum_{i=1}^{n} \sum_{k=1}^{K} \|\boldsymbol{w}_k\|_2^2 + \frac{1}{2c_3} \sum_{i=1}^{n} \left( \sum_{k=1}^{K} \left\| \boldsymbol{h}_i^{(k)} \right\|_2^2 - K \|\tilde{\boldsymbol{h}}_i\|_2^2 \right) \right]$$

$$- \left( \frac{1 + c_1}{N} (K - 1) - \frac{1 + c_2}{N} \right) \left( \frac{c_4}{2} \sum_{k=1}^{K} \sum_{i=1}^{n} \|\boldsymbol{w}_k\|_2^2 + \frac{1}{2c_4} \sum_{k=1}^{K} \sum_{i=1}^{n} \|\boldsymbol{h}_i^{(k)}\|_2^2 \right)$$

$$- \left( \frac{1 + c_1}{K} (K - 1) - \frac{1 + c_2}{K} \right) \sum_{k=1}^{K} b_k \tag{111}$$

$$= -\frac{1 + c_1}{n} \left( \frac{c_3}{2} \sum_{i=1}^{n} \sum_{k=1}^{K} \|\boldsymbol{w}_k\|_2^2 + \frac{1}{2c_3} \sum_{i=1}^{n} \sum_{k=1}^{K} \left\| \boldsymbol{h}_i^{(k)} \right\|_2^2 \right)$$

$$- \left( \frac{1 + c_1}{N} (K - 1) - \frac{1 + c_2}{N} \right) \left( \frac{c_4}{2} \sum_{k=1}^{K} \sum_{i=1}^{n} \|\boldsymbol{w}_k\|_2^2 + \frac{1}{2c_4} \sum_{k=1}^{K} \sum_{i=1}^{n} \|\boldsymbol{h}_i^{(k)}\|_2^2 \right)$$

$$- \left( \frac{1 + c_1}{K} (K - 1) - \frac{1 + c_2}{K} \right) \sum_{k=1}^{K} b_k + \frac{1 + c_1}{2nc_3} \sum_{i=1}^{n} K \|\tilde{\boldsymbol{h}}_i\|_2^2 \tag{112}$$

$$= -\frac{1 + c_1}{n} \left( \frac{nc_3}{2} \|\boldsymbol{W}\|_F^2 + \frac{1}{2c_3} \|\boldsymbol{H}\|_F^2 \right)$$

$$- \left( \frac{1 + c_1}{N} (K - 1) - \frac{1 + c_2}{N} \right) \left( \frac{nc_4}{2} \|\boldsymbol{W}\|_F^2 + \frac{1}{2c_4} \|\boldsymbol{H}\|_F^2 \right)$$

$$- \left( \frac{1 + c_1}{K} (K - 1) - \frac{1 + c_2}{K} \right) \sum_{k=1}^{K} b_k + \frac{1 + c_1}{2nc_3} \sum_{i=1}^{n} K \|\tilde{\boldsymbol{h}}_i\|_2^2 \tag{113}$$

*and the inequality becomes an equality if and only if*

$$c_3 \boldsymbol{w}_k = \boldsymbol{h}_i^{(k)} - \tilde{\boldsymbol{h}}_i, \quad \forall \, k \in [K], \; i \in [n], \quad \text{and} \tag{114}$$

$$c_4 \boldsymbol{w}_k = -\boldsymbol{h}_i^{(k)}, \qquad \forall \, k \in [K], \; i \in [n], \tag{115}$$

*which can be achieved only when* $\tilde{\boldsymbol{h}}_i = \mathbf{0}$.

*Let* $\rho = \|\boldsymbol{W}\|_F^2$. *Then, by using Lemma 1, we have* $\|\boldsymbol{H}\|_F^2 = \frac{\lambda_{\boldsymbol{W}}}{\lambda_{\boldsymbol{H}}} \rho$, *and*

$$f_{\text{bce}}(\boldsymbol{W}, \boldsymbol{H}, \boldsymbol{b})$$

$$= g_{\text{bce}}(\boldsymbol{W}\boldsymbol{H} + \boldsymbol{b}\mathbf{1}^T) + \frac{\lambda_{\boldsymbol{W}}}{2}\|\boldsymbol{W}\|_F^2 + \frac{\lambda_{\boldsymbol{H}}}{2}\|\boldsymbol{H}\|_F^2 + \frac{\lambda_{\boldsymbol{b}}}{2}\|\boldsymbol{b}\|_2^2 \tag{116}$$

$$\geq \; -\frac{1}{n(1+c_2)}\left(\frac{nc_3}{2}\|\boldsymbol{W}\|_F^2 + \frac{1}{2c_3}\|\boldsymbol{H}\|_F^2\right)$$

$$-\left(\frac{K-1}{N(1+c_2)} - \frac{1}{N(1+c_1)}\right)\left(\frac{nc_4}{2}\|\boldsymbol{W}\|_F^2 + \frac{1}{2c_4}\|\boldsymbol{H}\|_F^2\right)$$

$$-\left(\frac{K-1}{K(1+c_2)} - \frac{1}{K(1+c_1)}\right)\sum_{k=1}^K b_k + \frac{1}{2nc_3(1+c_2)}\sum_{i=1}^n K\|\tilde{\boldsymbol{h}}_i\|_2^2 + C$$

$$+\frac{\lambda_{\boldsymbol{W}}}{2}\rho + \frac{\lambda_{\boldsymbol{H}}}{2}\frac{\lambda_{\boldsymbol{W}}}{\lambda_{\boldsymbol{H}}}\rho + \frac{\lambda_{\boldsymbol{b}}}{2}\|\boldsymbol{b}\|_2^2 \tag{117}$$

$$= \; -\frac{1}{n(1+c_2)}\left(\frac{nc_3}{2}\rho + \frac{1}{2c_3}\frac{\lambda_{\boldsymbol{W}}}{\lambda_{\boldsymbol{H}}}\rho\right) - \left(\frac{K-1}{N(1+c_2)} - \frac{1}{N(1+c_1)}\right)\left(\frac{nc_4}{2}\rho + \frac{1}{2c_4}\frac{\lambda_{\boldsymbol{W}}}{\lambda_{\boldsymbol{H}}}\rho\right)$$

$$-\left(\frac{K-1}{K(1+c_2)} - \frac{1}{K(1+c_1)}\right)\sum_{k=1}^K b_k + \frac{1}{2nc_3(1+c_2)}\sum_{i=1}^n K\|\tilde{\boldsymbol{h}}_i\|_2^2 + C + \lambda_{\boldsymbol{W}}\rho + \frac{\lambda_{\boldsymbol{b}}}{2}\|\boldsymbol{b}\|_2^2 \tag{118}$$

$$= \left[\lambda_{\boldsymbol{W}} - \frac{1}{n(1+c_2)}\left(\frac{nc_3}{2} + \frac{1}{2c_3}\frac{\lambda_{\boldsymbol{W}}}{\lambda_{\boldsymbol{H}}}\right) - \left(\frac{K-1}{N(1+c_2)} - \frac{1}{N(1+c_1)}\right)\left(\frac{nc_4}{2} + \frac{1}{2c_4}\frac{\lambda_{\boldsymbol{W}}}{\lambda_{\boldsymbol{H}}}\right)\right]\rho$$

$$+\frac{\lambda_{\boldsymbol{b}}}{2}\|\boldsymbol{b}\|_2^2 - \left(\frac{K-1}{K(1+c_2)} - \frac{1}{K(1+c_1)}\right)\sum_{k=1}^K b_k + \frac{1}{2nc_3(1+c_2)}\sum_{i=1}^n K\|\tilde{\boldsymbol{h}}_i\|_2^2 + C \tag{119}$$

$$= \left[\lambda_{\boldsymbol{W}} - \frac{1}{n(1+c_2)}\left(\frac{nc_3}{2} + \frac{1}{2c_3}\frac{\lambda_{\boldsymbol{W}}}{\lambda_{\boldsymbol{H}}}\right) - \left(\frac{K-1}{N(1+c_2)} - \frac{1}{N(1+c_1)}\right)\left(\frac{nc_4}{2} + \frac{1}{2c_4}\frac{\lambda_{\boldsymbol{W}}}{\lambda_{\boldsymbol{H}}}\right)\right]\rho$$

$$+\frac{\lambda_{\boldsymbol{b}}}{2}\sum_{k=1}^K\left[b_k - \frac{1}{\lambda_{\boldsymbol{b}}}\left(\frac{K-1}{K(1+c_2)} - \frac{1}{K(1+c_1)}\right)\right]^2 - \frac{1}{2\lambda_{\boldsymbol{b}}}\sum_{k=1}^K\left(\frac{K-1}{K(1+c_2)} - \frac{1}{K(1+c_1)}\right)^2$$

$$+\frac{1}{2nc_3(1+c_2)}\sum_{i=1}^n K\|\tilde{\boldsymbol{h}}_i\|_2^2 + C \tag{120}$$

$$\geq \left[\lambda_{\boldsymbol{W}} - \frac{1}{n(1+c_2)}\left(\frac{nc_3}{2} + \frac{1}{2c_3}\frac{\lambda_{\boldsymbol{W}}}{\lambda_{\boldsymbol{H}}}\right) - \left(\frac{K-1}{N(1+c_2)} - \frac{1}{N(1+c_1)}\right)\left(\frac{nc_4}{2} + \frac{1}{2c_4}\frac{\lambda_{\boldsymbol{W}}}{\lambda_{\boldsymbol{H}}}\right)\right]\rho$$

$$+\frac{\lambda_{\boldsymbol{b}}}{2}\sum_{k=1}^K\left[b_k - \frac{1}{\lambda_{\boldsymbol{b}}}\left(\frac{K-1}{K(1+c_2)} - \frac{1}{K(1+c_1)}\right)\right]^2 - \frac{1}{2K\lambda_{\boldsymbol{b}}}\left(\frac{K-1}{1+c_2} - \frac{1}{1+c_1}\right)^2 + C \tag{121}$$

$$\geq \left[\lambda_{\boldsymbol{W}} - \frac{1}{n(1+c_2)}\left(\frac{nc_3}{2} + \frac{1}{2c_3}\frac{\lambda_{\boldsymbol{W}}}{\lambda_{\boldsymbol{H}}}\right) - \left(\frac{K-1}{N(1+c_2)} - \frac{1}{N(1+c_1)}\right)\left(\frac{nc_4}{2} + \frac{1}{2c_4}\frac{\lambda_{\boldsymbol{W}}}{\lambda_{\boldsymbol{H}}}\right)\right]\rho$$

$$-\frac{1}{2K\lambda_{\boldsymbol{b}}}\left(\frac{K-1}{1+c_2} - \frac{1}{1+c_1}\right)^2 + C, \tag{122}$$

*where the inequality (121) achieves its equality if and only if*

$$\tilde{\boldsymbol{h}}_i = \mathbf{0}, \quad \forall i \in [n], \tag{123}$$

and the inequality (122) becomes an equality whenever either

$$\lambda_{\boldsymbol{b}} = 0 \ \ or \ \ b_k = \frac{1}{\lambda_{\boldsymbol{b}}} \left( \frac{K-1}{K(1+c_2)} - \frac{1}{K(1+c_1)} \right), \ \ \forall k \in [K]. \tag{124}$$

Due to $\lambda_{\boldsymbol{b}} > 0$ and $c_1, c_2$ are same for any $k \in [K]$, therefore

$$b_k = b_j, \ \ \forall k, j \in [K]. \tag{125}$$

Based on Eqs. (114) and (123), we have

$$c_3 \boldsymbol{w}_k = \boldsymbol{h}_i^{(k)} \Rightarrow c_3^2 = \frac{\sum_{i=1}^n \sum_{k=1}^K \|\boldsymbol{h}_i^{(k)}\|_2^2}{\sum_{i=1}^n \sum_{k=1}^K \|\boldsymbol{w}_k\|_2^2} = \frac{\|\boldsymbol{H}\|_F^2}{n\|\boldsymbol{W}\|_F^2} = \frac{\lambda_{\boldsymbol{W}}}{n\lambda_{\boldsymbol{H}}} \Rightarrow c_3 = \sqrt{\frac{\lambda_{\boldsymbol{W}}}{n\lambda_{\boldsymbol{H}}}}; \tag{126}$$

similarly, from Eq. (115), we get

$$c_4 \boldsymbol{w}_k = -\boldsymbol{h}_i^{(k)} \Rightarrow c_4^2 = \frac{\sum_{i=1}^n \sum_{k=1}^K \|\boldsymbol{h}_i^{(k)}\|_2^2}{\sum_{i=1}^n \sum_{k=1}^K \|\boldsymbol{w}_k\|_2^2} = \frac{\|\boldsymbol{H}\|_F^2}{n\|\boldsymbol{W}\|_F^2} = \frac{\lambda_{\boldsymbol{W}}}{n\lambda_{\boldsymbol{H}}} \Rightarrow c_4 = -\sqrt{\frac{\lambda_{\boldsymbol{W}}}{n\lambda_{\boldsymbol{H}}}}. \tag{127}$$

Plugging them into Eq. (119), we get

$$f_{\mathrm{bce}}(\boldsymbol{W}, \boldsymbol{H}, \boldsymbol{b})$$

$$\geq \left[ \lambda_{\boldsymbol{W}} - \frac{1}{n(1+c_2)} \left( \frac{nc_3}{2} + \frac{1}{2c_3} \frac{\lambda_{\boldsymbol{W}}}{\lambda_{\boldsymbol{H}}} \right) - \left( \frac{K-1}{N(1+c_2)} - \frac{1}{N(1+c_1)} \right) \left( \frac{nc_4}{2} + \frac{1}{2c_4} \frac{\lambda_{\boldsymbol{W}}}{\lambda_{\boldsymbol{H}}} \right) \right] \rho$$

$$- \frac{1}{2K\lambda_{\boldsymbol{b}}} \left( \frac{K-1}{1+c_2} - \frac{1}{1+c_1} \right)^2 + C \tag{128}$$

$$= \left[ \lambda_{\boldsymbol{W}} - \left( \frac{1}{n(1+c_2)} - \frac{K-1}{N(1+c_2)} + \frac{1}{N(1+c_1)} \right) \left( \frac{n}{2} \sqrt{\frac{\lambda_{\boldsymbol{W}}}{n\lambda_{\boldsymbol{H}}}} + \frac{1}{2} \sqrt{\frac{n\lambda_{\boldsymbol{H}}}{\lambda_{\boldsymbol{W}}}} \frac{\lambda_{\boldsymbol{W}}}{\lambda_{\boldsymbol{H}}} \right) \right] \rho$$

$$- \frac{1}{2K\lambda_{\boldsymbol{b}}} \left( \frac{K-1}{1+c_2} - \frac{1}{1+c_1} \right)^2 + C \tag{129}$$

$$= \left[ \lambda_{\boldsymbol{W}} - \left( \frac{1}{n(1+c_2)} - \frac{K-1}{N(1+c_2)} + \frac{1}{N(1+c_1)} \right) \sqrt{\frac{n\lambda_{\boldsymbol{W}}}{\lambda_{\boldsymbol{H}}}} \right] \rho$$

$$- \frac{1}{2K\lambda_{\boldsymbol{b}}} \left( \frac{K-1}{1+c_2} - \frac{1}{1+c_1} \right)^2 + C \tag{130}$$

$$= \left[ \lambda_{\boldsymbol{W}} - \left( \frac{1}{N(1+c_2)} + \frac{1}{N(1+c_1)} \right) \sqrt{\frac{n\lambda_{\boldsymbol{W}}}{\lambda_{\boldsymbol{H}}}} \right] \rho - \frac{1}{2K\lambda_{\boldsymbol{b}}} \left( \frac{K-1}{1+c_2} - \frac{1}{1+c_1} \right)^2 + C \tag{131}$$

which is desired. ∎

**Lemma 4** *Under the same assumptions of Lemma 3, the lower bound in Eq. (99) is achieved for any critical point $(\boldsymbol{W}, \boldsymbol{H}, \boldsymbol{b})$ of Eq. (64) if and only if the following hold*

$$\|\boldsymbol{w}_1\| = \|\boldsymbol{w}_2\| = \cdots = \|\boldsymbol{w}_K\|, \ \ and \ \ \boldsymbol{b} = b^\star \boldsymbol{1}, \tag{132}$$

$$\boldsymbol{h}_i^{(k)} = \sqrt{\frac{\lambda_{\boldsymbol{W}}}{n\lambda_{\boldsymbol{H}}}} \boldsymbol{w}_k, \ \forall \, k \in [K], \ i \in [n], \ \ and \ \ \tilde{\boldsymbol{h}}_i = \frac{1}{K} \sum_{k=1}^K \boldsymbol{h}_i^{(k)} = \boldsymbol{0}, \forall \, i \in [n], \tag{133}$$

$$\boldsymbol{W}\boldsymbol{W}^T = \frac{\rho}{K-1} \left( \boldsymbol{I}_K - \frac{1}{K} \boldsymbol{1}_K \boldsymbol{1}_K^T \right), \tag{134}$$

$$c_1 = \exp\left( \sqrt{\frac{\lambda_{\boldsymbol{W}}}{n\lambda_{\boldsymbol{H}}}} \frac{\rho}{K} - b^\star \right), \ \ and \ \ c_2 = \exp\left( b^\star + \sqrt{\frac{\lambda_{\boldsymbol{W}}}{n\lambda_{\boldsymbol{H}}}} \frac{\rho}{K(K-1)} \right), \tag{135}$$

where $b^\star$ is the solution of equation

$$\lambda_{\boldsymbol{b}} b = \left[ \frac{K-1}{K \left( 1 + \exp\left( b + \sqrt{\frac{\lambda_{\boldsymbol{W}}}{n\lambda_{\boldsymbol{H}}}} \frac{\rho}{K(K-1)} \right) \right)} - \frac{1}{K \left( 1 + \exp\left( \sqrt{\frac{\lambda_{\boldsymbol{W}}}{n\lambda_{\boldsymbol{H}}}} \frac{\rho}{K} - b \right) \right)} \right]. \tag{136}$$

**Proof**   *With the proof of Lemma 3, to achieve the lower bound, it needs at least Eqs. (114), (115), and (123) to hold, i.e.,*

$$\tilde{\boldsymbol{h}}_i = \frac{1}{K}\sum_{k=1}^{K}\boldsymbol{h}_i^{(k)} = \boldsymbol{0}, \ \ \forall\, i \in [n], \ \ and \ \ \sqrt{\frac{\lambda_{\boldsymbol{W}}}{n\lambda_{\boldsymbol{H}}}}\boldsymbol{w}_k = \boldsymbol{h}_i^{(k)}, \ \ \forall\, k \in [K], \ i \in [n], \tag{137}$$

*and further implies*

$$\sum_{k=1}^{K}\boldsymbol{w}_k = \sqrt{\frac{n\lambda_{\boldsymbol{H}}}{\lambda_{\boldsymbol{W}}}}\sum_{k=1}^{K}\boldsymbol{h}_i^{(k)} = \boldsymbol{0}. \tag{138}$$

*Then,*

$$c_1 = \exp\Big(\boldsymbol{w}_k^T\boldsymbol{h}_i^{(k)} - b_k\Big) = \exp\Big(\sqrt{\frac{\lambda_{\boldsymbol{W}}}{n\lambda_{\boldsymbol{H}}}}\big\|\boldsymbol{w}_k\big\|_2^2 - b_k\Big), \ \ \forall k \in [K], \tag{139}$$

$$c_2 = \exp\Big(b_j - \boldsymbol{w}_j^T\boldsymbol{h}_i^{(k)}\Big) = \exp\Big(b_j - \sqrt{\frac{\lambda_{\boldsymbol{W}}}{n\lambda_{\boldsymbol{H}}}}\boldsymbol{w}_k^T\boldsymbol{w}_j\Big), \ \ \forall j \neq k \in [K], \tag{140}$$

*Since that $c_1, c_2$ are chosen to be the same for any $j \neq k \in [K]$, therefore,*

$$\sqrt{\frac{\lambda_{\boldsymbol{W}}}{n\lambda_{\boldsymbol{H}}}}\big\|\boldsymbol{w}_k\big\|_2^2 - b_k = \sqrt{\frac{\lambda_{\boldsymbol{W}}}{n\lambda_{\boldsymbol{H}}}}\big\|\boldsymbol{w}_j\big\|_2^2 - b_j, \ \ \forall k, j \in [K], \tag{141}$$

$$\sqrt{\frac{\lambda_{\boldsymbol{W}}}{n\lambda_{\boldsymbol{H}}}}\boldsymbol{w}_k^T\boldsymbol{w}_j - b_j = \sqrt{\frac{\lambda_{\boldsymbol{W}}}{n\lambda_{\boldsymbol{H}}}}\boldsymbol{w}_k^T\boldsymbol{w}_\ell - b_\ell, \ \ \forall j \neq \ell \in [K], \forall k \in [K], \tag{142}$$

*With the proof of Lemma 2, to achieve the lower bound, it needs at least Eqs. (91) to hold, then,*

$$\sqrt{\frac{\lambda_{\boldsymbol{W}}}{n\lambda_{\boldsymbol{H}}}}\big\|\boldsymbol{w}_k\big\|_2^2 - b_k$$

$$\overset{(138)}{=} \quad -\sqrt{\frac{\lambda_{\boldsymbol{W}}}{n\lambda_{\boldsymbol{H}}}}\sum_{\substack{j=1\\j\neq k}}\boldsymbol{w}_j^T\boldsymbol{w}_k - b_k \tag{143}$$

$$\overset{(142)}{=} \quad -\sqrt{\frac{\lambda_{\boldsymbol{W}}}{n\lambda_{\boldsymbol{H}}}}\sum_{\substack{\ell=1\\\ell\neq k}}^{K}\boldsymbol{w}_k^T\boldsymbol{w}_\ell - b_k + \sum_{\substack{j=1\\j\neq k}}(b_\ell - b_j) \tag{144}$$

$$= \quad -(K-1)\sqrt{\frac{\lambda_{\boldsymbol{W}}}{n\lambda_{\boldsymbol{H}}}}\underbrace{\boldsymbol{w}_k^T\boldsymbol{w}_\ell}_{\ell\neq k} - 2b_k + (K-1)b_\ell - K\bar{b} \tag{145}$$

$$\overset{(141,142)}{\Longrightarrow} \quad -2b_k + (K-1)b_\ell - K\bar{b} = -2b_\ell + (K-1)b_j - K\bar{b} \tag{146}$$

$$\Longleftrightarrow \quad b_k = b_\ell, \ \ \forall \ell \neq k \in [K], \tag{147}$$

*which is conforming to Eq. (125) when $\lambda_{\boldsymbol{b}} > 0$. Then, combining with Eqs. (141) and (138),*

$$\big\|\boldsymbol{w}_k\big\|_2^2 = \big\|\boldsymbol{w}_j\big\|_2^2 = \frac{\|\boldsymbol{W}\|_F^2}{K} = \frac{\rho}{K}, \ \ \forall k, j \in [K], \tag{148}$$

$$\big\|\boldsymbol{w}_k\big\|_2^2 = -(K-1)\sum_{\substack{j=1\\j\neq k}}^{K}\boldsymbol{w}_k^T\boldsymbol{w}_j \Rightarrow \boldsymbol{w}_k^T\boldsymbol{w}_j = -\frac{1}{K-1}\frac{\rho}{K}, \ \ \forall j \neq k \in [K]. \tag{149}$$

*Therefore,*

$$\boldsymbol{W}\boldsymbol{W}^T = \frac{\rho}{K-1}\Big(\boldsymbol{I}_K - \frac{1}{K}\boldsymbol{1}_K\boldsymbol{1}_K^T\Big). \tag{150}$$

*Plugging (148) and (149) into (139) and (140)*

$$c_1 = \exp\Big(\sqrt{\frac{\lambda_{\boldsymbol{W}}}{n\lambda_{\boldsymbol{H}}}}\frac{\rho}{K} - b\Big), \tag{151}$$

$$c_2 = \exp\Big(b + \sqrt{\frac{\lambda_{\boldsymbol{W}}}{n\lambda_{\boldsymbol{H}}}}\frac{\rho}{K(K-1)}\Big), \tag{152}$$

where $b = b_k = b_j$. When $\lambda_b > 0$, substitute Eqs. (151) and (152) into (124), we have

$$b = \frac{1}{\lambda_b}\left(\frac{K-1}{K(1+c_2)} - \frac{1}{K(1+c_1)}\right) \tag{153}$$

$$= \frac{1}{\lambda_b}\left[\frac{K-1}{K\left(1 + \exp\left(b + \sqrt{\frac{\lambda_W}{n\lambda_H}}\frac{\rho}{K(K-1)}\right)\right)} - \frac{1}{K\left(1 + \exp\left(\sqrt{\frac{\lambda_W}{n\lambda_H}}\frac{\rho}{K} - b\right)\right)}\right]. \tag{154}$$

When $\lambda_b = 0$, substitute Eq. (137) into

$$\frac{\partial f_{\text{bce}}}{\partial b_k} = \frac{1}{nK}\left(n - \sum_{j=1}^{K}\sum_{i=1}^{n}\frac{1}{1 + \mathrm{e}^{-\boldsymbol{w}_k \boldsymbol{h}_i^{(j)} + b_k}}\right) = 0, \ \ \forall k \in [K], \tag{155}$$

we have

$$0 = \frac{K-1}{K\left(1 + \exp\left(b + \sqrt{\frac{\lambda_W}{n\lambda_H}}\frac{\rho}{K(K-1)}\right)\right)} - \frac{1}{K\left(1 + \exp\left(\sqrt{\frac{\lambda_W}{n\lambda_H}}\frac{\rho}{K} - b\right)\right)}, \tag{156}$$

by combining with Eqs. (148) and (149). ∎

**Lemma 5** *The equation*

$$\lambda_b b = \frac{K-1}{K\left(1 + \exp\left(b + \sqrt{\frac{\lambda_W}{n\lambda_H}}\frac{\rho}{K(K-1)}\right)\right)} - \frac{1}{K\left(1 + \exp\left(\sqrt{\frac{\lambda_W}{n\lambda_H}}\frac{\rho}{K} - b\right)\right)} \tag{157}$$

*has only one solution.*

**Proof** *A number $b^\star$ is a solution of equation (157) if and only if it is a solution of*

$$\overbrace{\lambda_b K b + \frac{1}{1 + \exp\left(\sqrt{\frac{\lambda_W}{n\lambda_H}}\frac{\rho}{K} - b\right)}}^{\beta_1(b)} = \overbrace{\frac{K-1}{1 + \exp\left(b + \sqrt{\frac{\lambda_W}{n\lambda_H}}\frac{\rho}{K(K-1)}\right)}}^{\beta_2(b)}. \tag{158}$$

*When $\lambda_b > 0$,*

$$\beta_1(b) \to -\infty, \ \ \beta_2(b) \to K-1 \quad as \ \ b \to -\infty \tag{159}$$
$$\beta_1(b) \to +\infty, \ \ \beta_2(b) \to 0 \qquad as \ \ b \to +\infty, \tag{160}$$

*and if $\lambda_b = 0$,*

$$\beta_1(b) = 0, \qquad \beta_2(b) \to K-1 \quad as \ \ b \to -\infty \tag{161}$$
$$\beta_1(b) \to +\infty, \ \ \beta_2(b) \to 0 \qquad as \ \ b \to +\infty. \tag{162}$$

*Therefore, the curves of $\beta_1(b)$ and $\beta_2(b)$ must intersect at least once in the plane, i.e., the equations (157) and (158) have solutions.*

*In addition,*

$$\frac{\mathrm{d}\beta_1(b)}{\mathrm{d}b} = \lambda_b K + \frac{\exp\left(\sqrt{\frac{\lambda_W}{n\lambda_H}}\frac{\rho}{K} - b\right)}{\left(1 + \exp\left(\sqrt{\frac{\lambda_W}{n\lambda_H}}\frac{\rho}{K} - b\right)\right)^2} > 0, \tag{163}$$

$$\frac{\mathrm{d}\beta_2(b)}{\mathrm{d}b} = -\frac{(K-1)\exp\left(b + \sqrt{\frac{\lambda_W}{n\lambda_H}}\frac{\rho}{K(K-1)}\right)}{\left(1 + \exp\left(b + \sqrt{\frac{\lambda_W}{n\lambda_H}}\frac{\rho}{K(K-1)}\right)\right)^2} < 0, \tag{164}$$

*i.e., $\beta_1(b)$ is strictly increasing, while $\beta_2(b)$ is strictly decreasing. Therefore, they can intersect at only one point.* ∎

**Lemma 6** *When the class number $K > 2$ and*

$$\lambda_{\boldsymbol{b}}\sqrt{\frac{\lambda_{\boldsymbol{W}}}{n\lambda_{\boldsymbol{H}}}}\frac{\rho}{K-1} + \frac{1}{2(K-1)} > \frac{1}{1 + \exp\left(\sqrt{\frac{\lambda_{\boldsymbol{W}}}{n\lambda_{\boldsymbol{H}}}}\frac{\rho}{K-1}\right)}, \tag{165}$$

*the final critical bias $b^\star$ could uniformly separate the all positive decision scores*

$$\left\{\boldsymbol{w}_k^{\star T}\boldsymbol{h}_i^{(k)\star} : k \in [K], i \in [n]\right\} \tag{166}$$

*and the all negative decision scores*

$$\left\{\boldsymbol{w}_j^{\star T}\boldsymbol{h}_i^{(k)\star} : k, j \in [K], i \in [n], k \neq j\right\}, \tag{167}$$

*where*

$$\boldsymbol{W}^\star = \left[\boldsymbol{w}_1^\star, \boldsymbol{w}_2^\star, \cdots, \boldsymbol{w}_K^\star\right]^T \tag{168}$$

$$\boldsymbol{H}^\star = \left[\boldsymbol{h}_1^{(1)\star}, \cdots, \boldsymbol{h}_n^{(1)\star}, \cdots, \boldsymbol{h}_1^{(K)\star}, \cdots, \boldsymbol{h}_n^{(K)\star}\right] \tag{169}$$

$$\boldsymbol{b}^\star = (b^\star, b^\star, \cdots, b^\star)^T = b^\star \mathbf{1}_K \tag{170}$$

*form the critical point of function $f(\boldsymbol{W}, \boldsymbol{H}, \boldsymbol{b})$ in Eq. (64).*

**Proof** *According to Lemma 4, for the critical point $(\boldsymbol{W}^\star, \boldsymbol{H}^\star, \boldsymbol{b}^\star)$, we have*

$$\boldsymbol{w}_k^{\star T}\boldsymbol{h}_i^{(k)\star} = \sqrt{\frac{\lambda_{\boldsymbol{W}}}{n\lambda_{\boldsymbol{H}}}}\frac{\rho}{K}, \quad \forall k \in [K], i \in [n] \tag{171}$$

$$\boldsymbol{w}_j^{\star T}\boldsymbol{h}_i^{(k)\star} = -\sqrt{\frac{\lambda_{\boldsymbol{W}}}{n\lambda_{\boldsymbol{H}}}}\frac{\rho}{K(K-1)}, \quad \forall k, j \in [K], i \in [n], k \neq j. \tag{172}$$

*Let $b_{\text{neg}} = -\sqrt{\frac{\lambda_{\boldsymbol{W}}}{n\lambda_{\boldsymbol{H}}}}\frac{\rho}{K(K-1)}, b_{\text{pos}} = \sqrt{\frac{\lambda_{\boldsymbol{W}}}{n\lambda_{\boldsymbol{H}}}}\frac{\rho}{K}$. Then, the critical $b^\star$ separating the all positive and negative score if and only if*

$$b_{\text{neg}} = -\sqrt{\frac{\lambda_{\boldsymbol{W}}}{n\lambda_{\boldsymbol{H}}}}\frac{\rho}{K(K-1)} < b^\star < \sqrt{\frac{\lambda_{\boldsymbol{W}}}{n\lambda_{\boldsymbol{H}}}}\frac{\rho}{K} = b_{\text{pos}} \tag{173}$$

*which, according to the proof of Lemma 5, is equivalent to*

$$\beta_1(b_{\text{neg}}) < \beta_2(b_{\text{neg}}) \text{ and } \beta_1(b_{\text{pos}}) > \beta_2(b_{\text{pos}}). \tag{174}$$

*Due to*

$$\beta_1(b_{\text{neg}}) < \beta_2(b_{\text{neg}}) \Leftrightarrow -\lambda_{\boldsymbol{b}}\sqrt{\frac{\lambda_{\boldsymbol{W}}}{n\lambda_{\boldsymbol{H}}}}\frac{\rho}{K-1} + \frac{1}{1 + \exp\left(\sqrt{\frac{\lambda_{\boldsymbol{W}}}{n\lambda_{\boldsymbol{H}}}}\frac{\rho}{K-1}\right)} < \frac{K-1}{2}$$

$$\Leftarrow \frac{1}{1 + \mathrm{e}^0} < \frac{K-1}{2} \Leftarrow 2 < K \tag{175}$$

$$\beta_1(b_{\text{pos}}) > \beta_2(b_{\text{pos}}) \Leftrightarrow \lambda_{\boldsymbol{b}}\sqrt{\frac{\lambda_{\boldsymbol{W}}}{n\lambda_{\boldsymbol{H}}}}\rho + \frac{1}{2} > \frac{K-1}{1 + \exp\left(\sqrt{\frac{\lambda_{\boldsymbol{W}}}{n\lambda_{\boldsymbol{H}}}}\frac{\rho}{K-1}\right)} \tag{176}$$

$$\Leftrightarrow \lambda_{\boldsymbol{b}}\sqrt{\frac{\lambda_{\boldsymbol{W}}}{n\lambda_{\boldsymbol{H}}}}\frac{\rho}{K-1} + \frac{1}{2(K-1)} > \frac{1}{1 + \exp\left(\sqrt{\frac{\lambda_{\boldsymbol{W}}}{n\lambda_{\boldsymbol{H}}}}\frac{\rho}{K-1}\right)}, \tag{177}$$

*it completes the proof.* ∎

## S-4 MORE DISCUSSION ABOUT DECISION SCORES IN THE TRAINING

In the training, the decision scores are updated along the negative direction of their gradients during the back propagation stage, i.e.,

$$\boldsymbol{w}_k\boldsymbol{h}^{(k)} \;\leftarrow\; \boldsymbol{w}_k\boldsymbol{h}^{(k)} - \eta\frac{\partial f_\mu(\boldsymbol{W},\boldsymbol{H},\boldsymbol{b})}{\partial\big(\boldsymbol{w}_k\boldsymbol{h}^{(k)}\big)},\;\; \forall k \in [K], \tag{178}$$

$$\boldsymbol{w}_j\boldsymbol{h}^{(k)} \;\leftarrow\; \boldsymbol{w}_j\boldsymbol{h}^{(k)} - \eta\frac{\partial f_\mu(\boldsymbol{W},\boldsymbol{H},\boldsymbol{b})}{\partial\big(\boldsymbol{w}_j\boldsymbol{h}^{(k)}\big)},\;\; \forall j \neq k \in [K], \tag{179}$$

where $\eta$ is the learning rate, and $\mu \in \{\mathrm{ce}, \mathrm{bce}\}$.

In the training with CE, the updating formulas are

$$\boldsymbol{w}_k\boldsymbol{h}^{(k)} \;\leftarrow\; \boldsymbol{w}_k\boldsymbol{h}^{(k)} + \eta\bigg(1 - \frac{\mathrm{e}^{\boldsymbol{w}_k\boldsymbol{h}^{(k)}-b_k}}{\sum_\ell \mathrm{e}^{\boldsymbol{w}_\ell\boldsymbol{h}^{(k)}-b_\ell}}\bigg), \tag{180}$$

$$\boldsymbol{w}_j\boldsymbol{h}^{(k)} \;\leftarrow\; \boldsymbol{w}_j\boldsymbol{h}^{(k)} - \eta\frac{\mathrm{e}^{\boldsymbol{w}_j\boldsymbol{h}^{(k)}-b_j}}{\sum_\ell \mathrm{e}^{\boldsymbol{w}_\ell\boldsymbol{h}^{(k)}-b_\ell}}. \tag{181}$$

Then, for the samples with diverse initial decision scores, it is difficult to update their decision scores to the similar level, if they own the similar predicted probabilities belong to each categories.

In the training with BCE, the updating formulas are

$$\boldsymbol{w}_k\boldsymbol{h}^{(k)} \;\leftarrow\; \boldsymbol{w}_k\boldsymbol{h}^{(k)} + \eta\bigg(1 - \frac{1}{1+\mathrm{e}^{-\boldsymbol{w}_k\boldsymbol{h}^{(k)}+b_k}}\bigg), \tag{182}$$

$$\boldsymbol{w}_j\boldsymbol{h}^{(k)} \;\leftarrow\; \boldsymbol{w}_j\boldsymbol{h}^{(k)} - \eta\frac{1}{1+\mathrm{e}^{-\boldsymbol{w}_j\boldsymbol{h}^{(k)}+b_j}}. \tag{183}$$

Then, for the sample with small positive decision score $\boldsymbol{w}_k\boldsymbol{h}^{(k)}$, its predicted probability $\frac{1}{1+\mathrm{e}^{-\boldsymbol{w}_k\boldsymbol{h}^{(k)}+b_k}}$ to its category will be also small, and the score updating amplitude $\eta\big(1 - \frac{1}{1+\mathrm{e}^{-\boldsymbol{w}_k\boldsymbol{h}^{(k)}+b_k}}\big)$ will be large; in contrary, for the sample with large positive score $\boldsymbol{w}_k\boldsymbol{h}^{(k)}$, the probability $\frac{1}{1+\mathrm{e}^{-\boldsymbol{w}_k\boldsymbol{h}^{(k)}+b_k}}$ will be also large, and the updating amplitude $\eta\big(1 - \frac{1}{1+\mathrm{e}^{-\boldsymbol{w}_k\boldsymbol{h}^{(k)}+b_k}}\big)$ will be small. This property helps to update the all positive decision scores to be in uniform high level.

Similarly, for the sample with large negative decision score $\boldsymbol{w}_j\boldsymbol{h}^{(k)}$, its predicted probability $\frac{1}{1+\mathrm{e}^{-\boldsymbol{w}_j\boldsymbol{h}^{(k)}+b_j}}$ to other category will be also large, so is the score updating amplitude $\eta\frac{1}{1+\mathrm{e}^{-\boldsymbol{w}_j\boldsymbol{h}^{(k)}+b_j}}$; in contrary, for the sample with small negative score $\boldsymbol{w}_j\boldsymbol{h}^{(k)}$, the probability $\frac{1}{1+\mathrm{e}^{-\boldsymbol{w}_j\boldsymbol{h}^{(k)}+b_k}}$ will be small, so is the updating amplitude $\eta\big(1 - \frac{1}{1+\mathrm{e}^{-\boldsymbol{w}_j\boldsymbol{h}^{(k)}+b_k}}\big)$. This property helps to update the all negative decision scores to be in uniform low level.