# OpenReview forum: "BCE vs. CE in Deep Feature Learning"
_ICLR.cc/2025/Conference — Submitted to ICLR 2025_

### Official Review · Reviewer_bpu4 · 2024-11-01

**Soundness:** 3
**Presentation:** 3
**Contribution:** 3
**Rating:** 6
**Confidence:** 4

**Summary:**

The paper shows that the features learned by deep neural network architectures trained with binary cross entropy (BCE) loss maximize intra-class compactness and inter-class distinctiveness when the loss reaches its minimum, thereby leading to the well-known phenomenon of neural collapse. The paper then illustrates that both classification performance and search/retrieval performance, as previously empirically evaluated by other works, are due to the capability of BCE compared to CE in learning more compact and separated features.

**Strengths:**

1. The paper broadens the loss functions that can lead to neural collapse including a novel ene into the list. It is well written and organized.

2. The formal proof, similar to previous studies on neural collapse, is very interesting. The use of lower bounds for the BCE loss and identifying the conditions for achieving these lower bounds is a significant result.

3. The specific result with respect to the CE case shows that the global minimizers' biases satisfy a particular equation.

**Weaknesses:**

1. The paper is hard to follow despite the very good proof. Even in the insightful parts, it does not clearly demonstrate the advantage of BCE over CS. However, it provides a solid analysis of neural collapse in the context of BCE loss. As an example, the papers Wen et al. (2022) and Zhou et al. (2023) more effectively presented the geometric relationships of the involved entities.

2. In the reviewer's opinion, the bounded aspect of BCE does not seem to provide a clear advantage over the unbounded CE. Typically, features learned with cross-entropy (CE) are used in a normalized form. This operation projects the unbounded features onto a hypersphere, placing them within a bounded hyperspherical region around the prototype.

3. The related work section seems to be generic with respect to the addressed topic. The section concludes by stating that none of the existing works clearly reveal the advantage of BCE loss, However the insights into these advantages are still difficult to perceive at this stage. The works Wen et al. (2022) and Zhou et al. (2023) should be introduced earlier and revisited later in light of the paper's main findings on bias and decision boundaries. The current presentation makes it difficult to grasp the paper's main point.

4. Typically, papers about Neural Collapse with CE, including the two mentioned in this paper (Yang et al., 2022; Kim & Kim, 2024), show that the final classifier geometry of a regular simplex can be fixed at the beginning of training by setting the classifier biases to zero. Does BCE follow a similar pattern in this case, or do the biases play a significant role?

5. The contrastive property seems to be very important throughout the paper and should have been discussed more thoroughly.

6. In Figure 1, features within the hyperball radius are shown as fully bounded features, with two of the hyperspheres being tangent at contact points while two are not. This needs clarification on the significance of the tangency and non-tangency.

7. Line 427: Why is the bias set to 0, 1, 2, 3, 4, etc., for visual identification purposes in the figure?

**Questions:**

Please refer to the weaknesses box.

---

> ### Author Response · Authors · 2024-11-20
> **response to W1-W2**
>
> **W1: the advantage of BCE over CE**
>
> **R-W1:** We thank the reviewer for the positive comments on our good proof and solid analysis that BCE leads to neural collapse.
>
> Our paper focuses on the theorical comparison of BCE and CE in deep feature learning when they are used for multi-class classification tasks. Based on our proof and analysis, we found that the classifier bias in BCE plays a substantial role in the model training, which constrains all positive decision scores to a uniform high level and all negative decision scores to a uniform low level. However, for CE, its classifier bias does not have this effect. This advantage allows BCE to achieve better intra-class compactness and inter-class distinctiveness of features compared to CE when training models in practice, although they both theoretically lead to neural collapse, i.e., maximizing the compactness and distinctiveness of features. The experiments in the paper also fully justify our claim.
>
> Both SphereFace2 and UniFace in (Wen et al., 2022) and (Zhou et al., 2023) were designed for face recognition, and they took the unified bias integrated normalized BCE. They actually perform feature comparison/contrastive learning on the hypersphere, and their papers also clearly present the geometric relationship of the involved entities on the hypersphere. Our paper focuses on the comparison of BCE and CE in deep feature learning when they are used for multi-class classifications. Without the normalizations on the classifier and features, Fig. 1 in our paper presents their geometric comparison for CE and BCE.
>
> ---
> ---
>
> **W2: BCE and CE in normalized form**
>
> **R-W2:** Thanks for the comments. During the training of muti-class classification models, CE usually does not use normalization. Without the normalization, constraining the features within the same class to a bounded area is beneficial for learning features with better properties.  According to our paper's proof and analysis, the classifier bias in BCE can provides this constrain, which improve the classification performance of the models. The experimental results in the paper also verify our conclusions.
>
> When used for feature comparison tasks such as face recognition, the classifier and features in CE are usually normalized, and the features are learned on the hypersphere. The references of (Wen et al., 2022) and (Zhou et al., 2023) have shown that the normalized BCE can also be used in these tasks and achieve superior face recognition results.
>
> After both the classifier and feature vectors are normalized, the inner product in the decision scores becomes the cosine of the angle between the two vectors. At this point, on the hypersphere, the features learned by both CE and BCE are contained within the bounded regions. Nevertheless, when minimizing BCE and CE, it is also necessary to make the gradient of classifier bias converge to 0. In this process, the classifier bias of BCE can still substantially constrain the positive and negative decision scores, while the classifier bias of CE does not. In fact, on the hypersphere, the positive and negative decision scores of CE can take values within $[-1,1]$, and it only requires that the cosine value (positive decision score) between a sample feature vector and its corresponding classifier vector is relatively greater than that (negative decision scores) between the feature vector and other classifier vectors. With normalized CE, the classifier bias still provides a compensation for these cosine values, but does not play a substantial role. In contrast, BCE expects that the cosine value of the angle between the feature vector $h^{(k)}$ from the $k$-th class and its classifier vector $w_k$, i.e., positive decision score, is greater than the classifier bias $b_k$, and the cosine value between the feature vector and the other classifier vector $w_j$, i.e., negative decision score, is less than the classifier bias $b_j$. Therefore, the normalized BCE still has the potential to obtain better features than the normalized CE, as its classifier bias provides a constrain on its feature learning.
>
> When the class number $K=2$, one can image the feature distribution of CE and BCE on a sphere in 3D space. If the north pole and south pole on the sphere were used as the classifier vectors of the first and second classes, then CE can correctly classify all the samples as long as the features of the first class are distributed in the northern hemisphere and that of the second class are distributed in the southern hemisphere. In contrast, for BCE, it requires that the feature points of these two classes are distributed within a spherical circle centered at the north and south poles, with a radius related to the classifier bias, which implies higher intra-class compactness and inter-class distinctiveness.

---

> ### Author Response · Authors · 2024-11-20
> **responses to W3-W4**
>
> **W3: the related work**
>
> **R-W3:** Thanks for the comments. Our paper focuses on comparing BCE and CE in feature learning when they are used to train the multi-class classification models. For the first time, we theoretically prove that BCE, like CE, can also lead to neural collapse by reaching its minimum, which maximizes the intra-class compactness and inter-class distinctiveness of features. Then, our analysis shows that in practice, when using BCE to train the models, the classifier bias plays a substantial impact on the feature learning, which helps to enhance the compactness and distinctiveness of the features. CE does not have this advantage. Therefore, BCE can obtain better compactness and distinctiveness for sample features than CE. Our experimental results have validated the above analysis.
>
> In the first part of related works, we briefly reviewed the current applications of CE and BCE in training multi-class classification models. Though CE still dominates this field, BCE is also gaining increasing attention, making it necessary to analyze the potential advantages that BCE may have. The second part of related works reviews the neural collapse and briefly describes its current researches. There is currently no published work that focuses on whether BCE can lead to neural collapse.
>
> The works of Wen et al. (2022) and Zhou et al. (2023) did apply BCE to build their superior face recognition methods, but they mainly focused on the practical application of BCE in the task, so we did not emphasize these two works, although we have mentioned the work of Wen et al. (2022) in line 56 of the Introduction.
>
> ---
> ---
>
> **W4: BCE with fixed classifier**
>
> **R-W4:** Thanks for the comments. When CE loss reaches its minimum, the classifier vectors form an equiangular tight frame (ETF), which is the basis for existing works that fix the classifier vectors into a regular simplex before the model training. Similarly, as our proof, when BCE loss reaches its minimum point, the classifier vectors also form an ETF. Therefore, before training a model using BCE, the classifier vectors can also be fixed as a regular simplex. However, only fixing the classifier to ETF before the model training, may prevent the classifier from aligning with the final learned sample features, resulting in decrease of the model performance, and various sophisticated techniques have been developed for improving the performance of models with the fixed classifiers.
>
> When the classifier vectors are fixed, setting the classifier biases of CE to 0 has no significant impact on the model training. Because all biases equal to 0, can also make CE reach a minimum, whether to set them to 0 before the model training has no substantial impact on the training. However, at this point, if the classifier biases of BCE are set to 0, there is a lack of the requirement for BCE to have a gradient towards 0 with respect to the biases during the model training. As a result, the explicit constraint of biases on the positive and negative decision scores of features does not exist, which can reduce the learned feature properties and might reduce the model performance.
>
> Even setting the classifier biases of BCE to 0, the zero-bias still implicitly constrain the feature learning; because during the minimization of the BCE loss, it is still desirable for the positive and negative decision scores to satisfy Eqs. 15 and 16 in the paper, i.e.,
> $$w_k h^{(k)}> 0 ~~ \text{and} ~~ w_j h^{(k)} < 0 ~~ \text{for}~~ \forall j\neq k$$
> regardless of whether the classifier vectors $[w_j]_{j=1}^K$ are fixed or not. In contrast, for CE, as long as its classifier biases are equal to each other, i.e., $b_k=b_j$, whether they are equal to 0 or not, they will not affect the decision results by the decision scores, seeing Eq. 14.

---

> ### Author Response · Authors · 2024-11-20
> **responses to W5-W7**
>
> **W5: the contrastive property**
>
> **R-W5:** Thanks for the comments. The contrastive property was induced by Zhou et al. (2022), when they analyzed the neural collapse of CE, focal loss, and label smoothing loss in a unified way. We described it in our supplementary, i.e., Definition S-1. As discussed by Zhou et al. (2022), the contrastive property describes the relative values between the positive and negative decision scores. We repeat the description of the contrastive property in (Zhou et al., 2022) as follows,
>
> $$L_{CE}(z,y_k)\geq\log\Big(1+(K-1)\exp⁡\big(\frac{\sum_{j\neq k}(z_j-z_k)}{K-1})\Big)= \phi_{CE}\big(\sum_{j\neq k}(z_j-z_k)\big)\qquad\qquad(8)$$
> where $\phi_{CE}(t) = \log\big(1+(K-1)\exp(\frac{t}{K-1})\big)$. $\cdots \cdots$ Since $\phi_{CE}$ is an increasing function, minimizing the CE loss $L_{CE}(z,y_k)$ is equivalent to maximizing $(K-1)z_k - \sum_{j\neq k}z_j$ , which contrasts the $k$-th output $z_k$ simultaneously to all the other outputs $z_j$ for all $j\neq k$. Thus, we call (8) as a contrastive property. Maximizing $(K-1)z_k - \sum_{j\neq k}z_j$  would lead to a positive (and relatively large) $z_k$ and negative (and relatively small) $z_j$.
>
>
> The above statements are excerpted from page 5 of the reference (Zhou et al., 2022). In the above statements, $z_k= w_k h^{(k)}$ denotes the positive decision score and $z_j= w_j h^{(k)}, j\neq k$ denote the negative decision scores. Motivated by the above discussion, Zhou et al., (2022) introduced the contrastive property for the general losses, and they proved the loss will lead to the neural collapse if it satisfies the contrastive property.
>
> BCE constraint the absolute value of the exponential decision scores, and we found that it does not satisfy the above contrastive property. Nevertheless, we still proved that BCE can lead to neural collapse, which is a contribution of our paper.
>
> ---
> ---
>
> **W6: tangency and non-tangency in Fig. 1**
>
> **R-W6:** Thanks for the comments. The two hyperspheres are tangent, indicating that the features of the two classes represented by the two hyperspheres have critical inter-class distinctiveness. The two hyperspheres are not tangent, indicating that there is a margin between the features of the two corresponding classes, i.e., the better inter-class distinctiveness.
>
> ---
> ---
>
> **W7: the average value of initial biases in Fig. 3**
>
> **R-W7:** Thanks for the comments. According to our analysis, the classifier biases learned by CE are not affected by the decision scores of the features, and in contrary, it leads us to conclude that the classifier bias of CE does not play a substantial role in its feature learning. To verify this conclusion, on MNIST ($K=10$), we set the average value $\frac{1}{10}\sum_{j=1}^{10} b_j$ of the initial values of the 10 learnable classifier biases $[b_j]_{j=1}^{10}$ to 0, 1, 2, 3, 4, 5, 6, 8, and 10, respectively, and trained **18** models using SGD and AdamW optimizers with CE. As the first two subfigures in Fig.3(top) show, after the training, the classifier biases of these models are still approximately 0, 1, 2, 3, 4, 5, 6, 8, and 10, respectively, while, no matter the optimizer, the initial average value of the classifier biases, or the final classifier biases, the positive decision scores converge to around 5.64 and the negative ones converge to around -0.63. These results support our conclusion.
>
> In contrast, with the same settings, we also train **18** models using BCE, and the last two subfigures of Fig.3(top) show the results. One can find that, there exists a strong correlation between the final classifier biases of these models and the final positive and negative decision scores of the features, consistent with that the classifier bias of BCE plays a substantial role in its feature learning.

---

> ### Comment · Reviewer_bpu4 · 2024-12-02
> **Comment to the Authors' Response**
>
> The reviewer appreciated several responses from the authors, although the specific properties the reviewer wanted to know more about are not fully positive. Below is a summary of the responses for which no entirely positive results emerged:
>
> 1. **Contrastive Property**: BCE **does not** satisfy (while CE does) the contrastive property, which involves balancing positive and negative decision scores, yet it still achieves effective feature separation through neural collapse.
>
> 2. **Tangency Between Hyperspheres**: The discussion on tangency between hyperspheres shows that tangent hyperspheres indicate critical inter-class distinctiveness, while non-tangent hyperspheres signify a better separation with a margin. The reviewer would expect **more discussion** about **tangency**, even though the geometry is not simple. The response is reasonable, but a more formal one would be appreciated.
>
> 3. **Fixed Classifier**: The use of a fixed classifier is explored, noting that fixing the classifier as a regular simplex can be effective but may reduce performance if alignment with learned features is not achieved properly. This seems to be a **limitation** of BCE with respect to CE, as many recent papers have used this property for various problems (e.g., Federated Learning [1], Continual Learning [2], Compatible Learning [3], Imbalanced Learning [4], and more [5]). Some discussion should be provided in the final revision.
>    - [1] Li, Zexi, et al. "No fear of classifier biases: Neural collapse inspired federated learning with synthetic and fixed classifier." ICCV 2023.
>    - [2] Pernici, Federico, et al. "Class-incremental learning with pre-allocated fixed classifiers." ICPR 2021.
>    - [3] Biondi, Niccolò, et al. "Stationary Representations: Optimally Approximating Compatibility and Implications for Improved Model Replacements." CVPR 2024.
>    - [4] Yang, Yibo, et al. "Inducing neural collapse in imbalanced learning: Do we really need a learnable classifier at the end of deep neural network?" NeurIPS 2022.
>    - [5] Kim, Hoyong, and Kangil Kim. "Fixed Non-negative Orthogonal Classifier: Inducing Zero-mean Neural Collapse with Feature Dimension Separation." ICLR 2024.
>
> 4. **Gradient of Classifier Bias**: The gradient of the classifier bias in BCE plays a critical role in constraining decision scores, driving more compact intra-class features and well-separated inter-class features, unlike CE where classifier bias has less impact on feature learning. It is not entirely clear why it is necessary for the gradient of the classifier bias to converge to zero. Consider elaborating on **why** this condition **is important** and how it impacts feature learning in BCE.
>
> The reviewer is willing to increase their rating and speak positively with other reviewers about the work. The authors are kindly invited to provide final feedback on the reviewer comments, specifically addressing the four points mentioned above, to further strengthen the paper.

---

> > ### Author Response · Authors · 2024-12-03
> > **The further responses to Reviewer bpu4 -- Part I**
> >
> > We are very grateful to Reviewer bpu4 for the discussion, which has been very helpful in improving our paper. Below are our responses to the comments from Reviewer bpu4.
> >
> > **1. Contrastive Property with BCE**
> >
> > **A-1**: To be precise, we cannot prove that BCE fulfills the contrastive property, i.e., after trying hard, we did not identify a function $\phi_{bce}$ satisfying
> > $$\qquad\mathcal L_{bce}(\textbf{z})\geq \phi_{bce}\Big(\sum_{j=1,j\neq k}(z_j – z_k)\Big).$$
> > While preparing this response, we tried again but still couldn’t find a function satisfies the condition, even though we couldn’t prove that BCE doesn’t satisfy the contrastive property.
> >
> > Since the contrastive property describes the relative value of the positive and negative decision scores, while BCE measures the absolute values of the exponential decision scores, we **conjecture** that BCE does not satisfy the contrastive property. We will clarify the statements about BCE and contrastive property in the final version of our paper.
> >
> > The reference Zhou et al. (2022) demonstrated that when a loss function satisfies the contrastive property, it would lead to neural collapse, such as CE, focal loss, and label smoothing loss, etc. In the paper, we have proven that BCE can also lead to neural collapse without invoking the contrastive property, which broaden the range of losses that can lead to neural collapse.
> >
> > **Reference**:
> >
> > Jinxin Zhou, et al. Are all losses created equal: a neural collapse perspective. NeurIPS 2022.
> >
> > ---
> > ---
> > **2. Tangency Between Hyperspheres**
> >
> > **A-2**:  In the distance space, if the two hyperspheres of features of two classes intersect and overlap, then the distance between their centers is less than the sum of their radii, i.e., $||w_k-w_j||_2 \leq b’_k + b’_j$, then there is **no** hyperplane that can distinguish between the features of the two classes, and there is no inter-class distinctiveness between them.
> >
> > If the two feature regions represented by the two hyperspheres are tangent, then their distance is zero, and the distance between their centers is equal to their radius sum. Meanwhile, their common tangent plane is the **only** hyperplane that can distinguish between the features of the two classes. As Fig. 1(b) shows, the two hyperspheres centered at $w_1$ and $w_2$
> > $$\qquad\mathcal S_1 = \big[ \{h^{(1)}: ||w_1-h^{(1)}||_2 \leq {b’}_1\}\big]$$
> > $$\qquad\mathcal S_2 = \big[ \{h^{(2)}: ||w_2-h^{(2)}||_2 \leq {b’}_2\}\big]$$
> > tangent, their distance
> > $$\qquad\text{distance}(\mathcal S_1, \mathcal S_2) = \min\big[||h^{(1)}-h^{(2)}||_2: \forall h^{(1)}\in \mathcal S_1, \forall h^{(2)}\in\mathcal S_2 \big] = 0$$
> > and $||w_1-w_2||_2 = b’_1+b’_2$. Their common tangent plane, $(w_1-w_2)^T(h-p)=0$, is the only hyperplane that can distinguish between their features, where $p$ denotes the tangent point of the two hyperspheres and $h$ denotes any feature point. If and only if the feature $h$ coincides with the tangent point $p$, the tangent plane cannot correctly classify the feature, i.e., the features of these two classes have the critical inter-class distinctiveness.
> >
> > If two disjoint feature hyperspheres are not tangent, then there are **infinite** hyperplanes that can completely distinguish between the features of the two classes. As Fig. 1(b) shows, the two hyperspheres centered at $w_2$ and $w_3$
> > $$\qquad \mathcal S_2 = \big[h^{(2)}: ||w_2-h^{(2)}||_2 \leq {b’}_2\big]$$
> > $$\qquad \mathcal S_3 = \big[h^{(3)}: ||w_3-h^{(3)}||_2 \leq {b’}_3\big]$$
> > do not tangent. The distance between them
> > $$\qquad \text{distance}(\mathcal S_3, \mathcal S_2) = \min\big[||h^{(3)}-h^{(2)}||: \forall h^{(3)}\in \mathcal S_3, \forall h^{(2)}\in\mathcal S_2\big] > 0$$
> > and $||w_3-w_2||_2 > b’_3+b’_2$. Obviously, there are countless hyperplanes that can distinguish the features represented by these two hyperspheres, and the features of the two classes have better inter-class differences.

---

> > ### Author Response · Authors · 2024-12-03
> > **The further responses to Reviewer bpu4 -- Part II**
> >
> > ---
> > ---
> > **3. Fixed Classifier**
> >
> > **A-3**: As the reviewer pointed out, after the neural collapse phenomenon was discovered, the fixed classifiers with simplex equiangular tight frame (ETF) structure were applied into many fields, including federated learning, continual learning, and long-tail recognition (LTR), etc. However, the fixed ETF classifier only considers the geometric structure of the classifier, without considering the direction of the classifier, which may not align well with the final learned features, making it difficult to fully utilize the performance of the ETF classifier.
> >
> > We believe that this defect will exist regardless of using CE or BCE losses, which is **not** the limitation of BCE with respect to CE. For example, when studying the LTR with the CE loss, Peifeng et al. (2023) argued that “the bad initialization of ETF is harmful to the generalization of the deep model”. They took a learnable orthogonal matrix to adjust the direction of the ETF classifier based on the CE loss, achieving better LTR results.
> > The fixed ETF classifier can also be used with the BCE loss, and we believe that a bad initialization of the fixed ETF classifier will also harm the generalization of the final models.
> >
> > **Reference**:
> >
> > Gao Peifeng, et al. Feature directions matter: long-tailed learning via rotated balanced representation, ICML 2023.
> >
> > ---
> > ---
> > **4. Gradient of Classifier Bias**
> >
> > **A-4**: Both the CE and BCE losses are differentiable with respect to their variables, including the classifier vectors, classifier biases, and the sample features. Thus, their minimum points must be stationary points regarding the classifier biases. Therefore, having the gradient of the classifier bias equal to zero is a necessary condition for reaching a minimum point. For univariate differentiable functions, this conclusion is expressed as **Fermat's Theorem in Calculus**. When using gradient descent algorithms (SGD, AdamW, and others) to optimize a model, it searches for the point where the gradient of the loss function is zero. Therefore, when optimizing the model along the gradient descent direction, the classifier bias constrains the positive and negative decision scores.
> >
> > For the CE loss, its gradient with respect to the classifier bias is shown in Eq. (20) of the paper, i.e.,
> > $$\qquad \frac{\partial f_{ce}}{\partial b_k}
> >     = \frac{1}{nK}\bigg(n - \sum_{j=1}^K\sum_{i=1}^n  \frac{\exp(w_k h_i^{(j)}-b_k)}{\sum_{\ell}\exp(w_\ell h_i^{(j)}-b_\ell)}\bigg) + \lambda_{b} b_k.$$
> > According to Theorem 1, when the CE loss reaches a minimum point and leads to neural collapse, the unbiased positive and negative decision scores converge to fixed values (i.e., $\sqrt{\frac{\lambda_{W}}{n \lambda_{H}}}\frac{\rho}{K}$ and $-\sqrt{\frac{\lambda_{W}}{n \lambda_{H}}}\frac{\rho}{K(K-1)}$), and $b_k=b_\ell$. When they are substituted into Eq. (20), the first term naturally equals 0, which indicates that the necessary condition for achieving a minimum — specifically, the gradient of the classifier bias being equal to 0 — does not provide any constraint on the classifier bias and the decision scores. A reasonable guess is that before reaching the minimum point, the requirement for the gradient to tend towards 0 only weakly constrains the classifier biases and the decision scores.
> >
> > In contrast, for BCE loss, its gradient with respect to the classifier bias is shown in Eq. (21) in the paper, i.e.,
> > $$\qquad\frac{\partial f_{{bce}}}{\partial b_k}
> >     = \frac{1}{nK}\bigg(n - \sum_{j=1}^K\sum_{i=1}^n \frac{1}{1+\exp(- w_k h_i^{(j)}+b_k)}\bigg) + \lambda_{b} b_k.$$
> > The minimum point of BCE is also point where the gradient of the classifier bias equals zero. Similarly, according to Theorem 2, the unbiased positive and negative decision scores converge to the fixed values and $b_k=b_\ell$. When substituted them into Eq. (21), they yield an equation regarding the classifier bias $b$, where the first term does not equal zero, which substantially constrains the final classifier bias $b$ and the positive and negative decision scores. A reasonable conjecture is that, before reaching this state, the requirement for the gradient to tend towards zero always substantially constrains the classifier bias and the decision scores.
> >
> > ---
> > ---
> > The above is our responses to Reviewer bpu4 item-by-item. If there are any deficiencies, we hope that the reviewer can further discuss with us after reviewing.

---

### Official Review · Reviewer_Eks9 · 2024-11-04

**Soundness:** 2
**Presentation:** 3
**Contribution:** 1
**Rating:** 3
**Confidence:** 4

**Summary:**

This paper compares the binary cross-entropy (BCE) loss with the traditional cross-entropy (CE) loss in the context of deep feature learning. The key idea is that both losses can lead to neural collapse, maximizing intra-class compactness and inter-class distinctiveness.

**Strengths:**

1. The paper is well-written and organized, clearly explaining the key ideas and providing a detailed comparison between BCE and CE.
2. The experimental results are presented systematically, supporting the claims made in the paper.

**Weaknesses:**

1. The volume of data used for the experiments is small, and for data-dependent deep neural networks, there will be a gap between the theoretical and practical effects.
2. The paper mentions that data augmentation techniques like Mixup and CutMix can improve classification performance, but it would be helpful to further investigate how these techniques interact with BCE and CE losses and whether they have a similar impact on feature properties.
3. The visualization in Figure 4 is not very explicit, and only the comparison of the two categories is clearer, which is not enough to support the authors' conclusion that “the features of BCE-trained ResNet18 for these categories are distributed in more compact areas and have signiﬁcant gaps between them, implying better feature compactness and distinctiveness.”

**Questions:**

1. Sigmoid loss and BCE loss are both commonly used loss functions for binary classification tasks, what are the differences and connections between them.
2. What is the classification performance of BCE loss on medium to large scale datasets? For example, ImageNet.
3. Would BCE loss have a theoretical advantage over sigmoid loss for comparison learning tasks?

---

> ### Author Response · Authors · 2024-11-20
> **responses to the Weaknesses**
>
> **W1: The gap between the theoretical and practical effects.**
>
> **R-W1:** Thanks for the comments. Our paper focuses on the theoretical comparison of BCE and CE in deep feature learning when they are used for training multi-class classification models. For the first time, the paper reveals through theoretical analysis that BCE can also lead to neural collapse, i.e., maximizing the intra-class compactness and inter-class distinctiveness of sample features. Moreover, our paper reveals the advantages of BCE over CE in deep feature learning. That is, the classifier bias in BCE plays a substantial role, i.e., explicitly constraining its feature learning, while the classifier bias in CE does not have this effect. Therefore, BCE can achieve better compactness and distinctiveness for sample features than CE. Though our experiments were conducted on the small datasets, such as MNIST, CIFAR10, and CIFAR100, the experimental results clearly verified the advantages of BCE over CE. At present, the published researches (Papyan et al., 2020; Zhu et al., 2021; Zhou et al., 2022; Liu et al., 2023; Guo et al., 2024) on neural collapse, revealing the theoretical properties of loss functions such as CE, MSE, focal loss, and label smoothing loss, etc., are commonly evaluated on such small datasets.
>
> On the more complex tasks, we believe that BCE can still achieve better results than CE, and we are trying to apply BCE on the large-scale or medium-scale datasets, such as ImageNet.
>
> **Reference:**
>
> Vardan Papyan, et al. Prevalence of neural collapse during the terminal phase of deep learning training. PNAS 2020.
>
> Zhihui Zhu, et al. A geometric analysis of neural collapse with unconstrained features. NeurIPS 2021.
>
> Jinxin Zhou, et al. Are all losses created equal: A neural collapse perspective. NeurIPS 2022.
>
> Weiyang Liu, et al. Generalizing and decoupling neural collapse via hyperspherical uniformity gap. ICLR 2023.
>
> Pengyu Li, et al. Neural collapse in multi-label learning with pick-all-label loss, ICML 2024.
>
> -----------------------------------------------------------------------
> -----------------------------------------------------------------------
>
> **W2: Mixup and CutMix**
>
> **R-W2:** Thanks for the comments. We mainly focus on the theoretical comparison of BCE and CE, while the classification experiments using different data augmentation techniques in Sec. 4.2 show that BCE achieves better classification results compared to CE, regardless of whether Mixup or CutMix is used. These results suggests that the advantage of BCE may stem from the fact that BCE itself can learn better features with higher intra-class compactness and inter-class distinctiveness compared to CE, rather than that BCE aligns with Mixup and CutMix as conjectured in the previous literatures (Wightman et al., 2021, Touvron et al., 2022).
>
> At present, we do not consider how CutMix and Mixup interact with BCE and CE through theoretical analysis. However, the experimental results in our paper still reveal some empirical conclusions about them. According to the results in Tables 2 and 3, when using the AdamW optimizer, DA2, i.e., Mixup and CutMix, can improve the classification accuracy and feature compactness and distinctiveness of different models on CIFAR10 and CIFAR100 in most cases. However, when using the SGD optimizer, DA2 may reduce the classification accuracy and feature properties in most cases. These results indicate that Mixup and CutMix require the suitable optimizer for better performance. In contrast, in the most cases, BCE can achieve better results than CE, regardless of the optimizer, model, or data augmentation technique.
>
> **Reference:**
>
> Ross Wightman, et al. ResNet strikes back: An improved training procedure in timm. NeurIPS 2021 Workshop, 2021.
>
> Hugo Touvron, et al. DeiT III: Revenge of the ViT. ECCV, 2022.
>
> -----------------------------------------------------------------------
> -----------------------------------------------------------------------
>
> **W3: Fig. 4**
>
> **R-W3:** Thanks for the comments. In Fig. 4, BCE clearly shows better intra-class compactness and inter-class distinctiveness than CE only on the class 3 and class 5, i.e., 'cat' and 'dog', and this visualization example still illustrates that the ResNet18 trained on CIFAR10 with BCE has better feature properties than the one trained with CE.
>
> More objectively, we define the intra-class compactness $\mathcal E_{com}$ and inter-class distinctiveness $\mathcal E_{dis}$ for the sample features, as shown in Eqs. 47 and 48 in Supplementary. We have compared the two properties of the sample features corresponding to BCE and CE under different optimizers, models, datasets, and data augmentations. The results in Table 3 show that in most cases, BCE achieves better intra-class compactness and inter-class distinctiveness for sample features than CE.

---

> ### Author Response · Authors · 2024-11-20
> **responses to the Questions**
>
> **Q1: Sigmoid loss and BCE loss**
>
> **R-Q1:** Thanks for your question. 'Sigmoid loss' does not seem to be a widely used standard term in deep learning. We may not have accurately understood this concept you mentioned in the following response.
>
> In deep learning, the commonly used loss for training of multi-class classification models is the cross-entropy (CE) loss, which has been analyzed in our paper. In the classification task, when the class number $K$ is equal to 2, the Softmax used in CE reduces to the Sigmoid, and CE also reduces to the ordinary binary CE, termed oBCE. Is this ordinary BCE the Sigmoid loss you mentioned?
>
> Different from the ordinary BCE used for binary classification tasks, our paper discusses the BCE used for multi-class classification. It is obtained by converting a multi-class classification task from $K$ classes into $K$ binary classification tasks, and then adding up the $K$ ordinary BCEs. For a sample $X$, in the training, the model $\mathcal M$ first extracts its feature $h$, and the classifier (full connection layer usually used in deep networks) converts the feature into $K$ metrics, $[w_j^T h-b_j]_{j=1}^K$. Then, Sigmoid is used to transform the $j$-th metric into the probabilities of the sample $X$ belongs to the class $j$, $\text{Sigmoid}(w_j^T h-b_j )=\frac{1}{1+\exp⁡(-w_j^T h+b_j)}$. For the training of model and classifier, the $j$-th ordinary BCE is
> $$-\big[p_j  \log⁡(\text{Sigmoid}(w_j^T h-b_j ))+(1-p_j )  \log⁡(1-\text{Sigmoid}(w_j^T h-b_j )) \big] =p_j \log⁡(1+\exp⁡(-w_j^T h+b_j))+(1-p_j )  \log⁡(1+\exp⁡(w_j^T h-b_j))$$
>  where $p_j$ represents the true probability that the sample belongs to the class $j$.
>
> When the sample is from class $k$, we denote it as $X^{(k)}$, and its feature as $h^{(k)}$. Then, $p_k=1$ and $p_j=0$ for $j\neq k$ for this sample, and the sum of its $K$ ordinary BCEs is
> $$\log(1+\exp(-w_k^T h^{(k)}+b_k ))+ \sum_{j=1,j\neq k}^K\log(1+\exp(w_j^T h^{(k)}-b_j )),$$
> which is the BCE used for multi-class classification tasks and analyzed in our paper.	Even $K=2$, the BCE discussed in our paper is different from the ordinary BCE (oBCE), which is the sum of two oBCEs, including one oBCE corresponding to a positive term and $K-1=1$ oBCE corresponding to negative term.
>
> ----------------------------------------
> ----------------------------------------
>
> **Q2: BCE on medium to large datasets.**
>
> **R-Q2:** Thanks for the question. We believe that BCE performs also better than CE on medium- or large-scale datasets. We conduct a group of experiments on ImageNet-1k. On ImageNet-1k, we apply BCE and CE to train ResNet50, ResNet101, and DenseNet161, respectively. Table 3 presents the classification results on the validation set of ImageNet-1k. While $\mathcal A$ denotes the classification accuracy, $\mathcal A_{Uni}$ denotes the uniform accuracy. For saving the training times, we did not train these three models from scratch. Instead, we fine-tune the models that have been pretrained for 90 epochs with CE, and each fine-tuning runs 30 epochs. Despite that, BCE clearly shows a consistent advantage over CE on this medium-scale dataset.
>
> **Table 3.**  Comparison of BCE and CE on ImageNet-1K
>
> | |R50|	R50|	R101|	R101|	D161|	D161|
> | --- | --- | --- | --- | --- | --- | --- |
> |    |$\mathcal A$| $\mathcal A_{Uni}$ |$\mathcal A$| $\mathcal A_{Uni}$ |$\mathcal A$| $\mathcal A_{Uni}$ |
> |CE	  |76.74|34.48|78.47|38.85|78.58|43.56|
> |BCE |**77.12**|**66.92**|**78.88**|**70.46**|	**79.19**|**69.08**|
>
> In addition, in the response to the reviewer **tKaJ**, we also briefly compared BCE and CE on the long-tailed recognition tasks, and BCE also showed clear advantages over CE. In future, we will systematically verify the performance of BCE on the more complex tasks.
>
> -------------------------------------
> -------------------------------------
>
> **Q3: BCE and sigmoid loss on comparison learning.**
>
> **R-Q3:** Thanks for your question. If the Sigmoid loss you mentioned refers to the ordinary BCE used for binary classification, then when it is used for comparison learning/contrastive learning of sample features, all samples will be combined into positive and negative sample pairs for training the model. In practice, even on small datasets such as CIFAR, the number of combinations of the sample pairs will increase dramatically, making it difficult to use all sample pairs to train the model in a short period of time, which may limit the performance of the trained model in the contrastive learning.
>
> The multi-class classification models trained using CE or BCE can also be used for the feature comparison learning, such as face recognitions, because the good classification results usually require high intra-class compactness and inter-class distinctiveness of features. Our paper has theoretically analyzed and experimentally verified that BCE can achieve better compactness and distinctiveness of features than CE. Therefore, BCE has great potential in contrastive/comparison learning tasks.

---

> ### Comment · Reviewer_tKaJ · 2024-11-21
> **Don’t you think BCE loss and sigmoid loss are essentially the same?**
>
> In binary classification, BCE loss pretty much assumes you’re using sigmoid.

---

> > ### Author Response · Authors · 2024-11-22
> > **the Sigmoid loss**
> >
> > Yes, we also think that the 'Sigmoid loss' mentioned by reviewer **Eks9** is the ordinary BCE used for binary classification tasks. If we have misunderstood, we hope that reviewer Eks9 can clarify the meaning of the Sigmoid loss mentioned, so that we can further discuss its relationship and differences with the BCE used for multi-classification tasks and analyzed in our paper.

---

> ### Author Response · Authors · 2024-11-27
> **Official Comment by Authors**
>
> Dear Reviewer Eks9,
>
> We have meticulously addressed your comment with comprehensive responses during the rebuttal phase. At present, we have not yet received new comments from you. We are looking forward to your valuable feedback and insights, and grateful for your effort throughout this process.
>
> Best regards,
>
> Submission 2929 authors

---

> ### Author Response · Authors · 2024-12-02
>
> Dear Reviewer Eks9,
>
> We are still waiting for your further reply and hope to discuss with you in order to improve our paper and future research.
>
> Best regards,
>
> Submission 2929 authors

---

> ### Author Response · Authors · 2024-12-02
>
> Dear Reviewer Eks9,
>
> Hope our response clarify the concerns raised and we are open for further discussion.
>
> Best regards,
>
> Submission 2929 authors

---

### Official Review · Reviewer_tKaJ · 2024-11-06

**Soundness:** 3
**Presentation:** 3
**Contribution:** 3
**Rating:** 8
**Confidence:** 3

**Summary:**

This paper explores whether the binary cross-entropy (BCE) loss function can induce neural collapse, a phenomenon in which deep learning models achieve tightly clustered intra-class features and well-separated inter-class features. Through both theoretical analysis and empirical validation, the authors show that, like cross-entropy (CE) loss, BCE loss can indeed facilitate neural collapse, even in multi-class tasks. Furthermore, the experiments demonstrate that models trained with BCE loss perform better than those trained with CE loss. This advantage is from the classifier bias introduced by BCE, which explicitly optimizes feature alignment towards class centers, enhancing intra-class compactness and inter-class separation. These findings highlight that BCE loss not only fosters a beneficial feature distribution but also boosts model performance.

**Strengths:**

1. Strong theoretical analysis of BCE’s ability to achieve neural collapse
This paper provides a detailed theoretical analysis comparing BCE and CE on neural collapse. It shows that BCE, like CE, can achieve high intra-class compactness and inter-class separation, indicating that BCE has strong theoretical potential in feature learning.

2. Practical analysis of decision score and boundary updates
Beyond the theory, the paper also explains the practical differences between BCE and CE regarding decision score and boundary updates. BCE is shown to provide consistent updates to decision scores, leading to stronger feature constraints. This insight helps us understand how BCE can improve feature representation.

3. Solid experimental validation
The experiments effectively support the theoretical points, showing BCE’s advantages over CE across different models and datasets.

**Weaknesses:**

1. Some practical training factors not considered: The paper doesn’t fully address certain practical training factors, such as weight decay and class weight initialization, which might impact the formation of a simplex equiangular tight frame and the contrastive properties.

2. Assumptions could be generalized: Some theorems are based on the assumption that feature dimension $d$ is larger than the category number $K$. In large-scale representation tasks (e.g., face recognition), the category number can be very large, and in practice, BCE is still effective even when $d<K$. It would strengthen the paper in discussing or generalizing this assumption, considering that BCE might be useful in larger-scale scenarios. (Although the generalization is difficult.)

**Questions:**

1. BCE’s performance in more complex tasks: These researches and experiments focus mainly on image classification, especially when feature dimension $d$ is greater than the number of categories $K$. For more complex tasks (like simple sequence classification or large-scale classification where $K$ might be larger than $d$), do you expect BCE to show similar advantages? Are there any plans to test BCE’s applicability in these settings?

2. Explicit vs. implicit constraints in feature learning: You’ve shown that BCE’s explicit constraints give it a clear edge in feature learning, and the experiments confirm this strong feature constraint effect. Would it be possible to further derive the performance limit (Upper-boundary) of CE’s implicit constraints? This might help to make a more direct comparison of their feature learning strengths.

---

> ### Author Response · Authors · 2024-11-20
> **responses to the Weaknesses**
>
> **[W1: Some practical training factors.]**
>
> **R-W1:** Thanks for the comments. As pointed out by the reviewer, there are many practical training factors, such as weight decay and class weight initialization, that can influence the process of reaching the minimum of the loss functions and forming simplex equiangular tight frames. Since our paper primarily focuses on comparing BCE and CE from the perspectives of neural collapse and deep feature learning, following the analysis mode in the previous literatures, such as (Zhu et al, 2021) and (Zhou et al, 2022), we mainly consider the weight decay on the classifier weight vector, classifier bias, and sample features. For the first time, our theoretical analysis reveals that the main difference between BCE and CE is that the classifier bias of BCE plays a substantial role in its feature learning, whereas the classifier bias in CE does not. Consequently, in our experiments, we compared CE and BCE with different weight decay and different initialization for the classifier bias.
>
> We believe that proper weight decay and initialization for the other parameters in the model aid its classifier vectors in forming a simplex equiangular tight frame, and for BCE and CE they may differ, which are also topics worth investigating for comparing BCE and CE.
>
> **Reference:**
>
> Zhihui Zhu, Tianyu Ding, Jinxin Zhou, Xiao Li, Chong You, Jeremias Sulam, and Qing Qu. A geometric analysis of neural collapse with unconstrained features. Advances in Neural Information Processing Systems, 34:29820–29834, 2021.
>
> Jinxin Zhou, Chong You, Xiao Li, Kangning Liu, Sheng Liu, Qing Qu, and Zhihui Zhu. Are all losses created equal: A neural collapse perspective. Advances in Neural Information Processing Systems, 35:31697–31710, 2022.
>
> -------------------------------
> -------------------------------
>
> **[W2: Assumptions could be generalized.]**
>
> **R-W2:** Thanks for the comments. As stated by the reviewer, it would be beneficial to consider the case where the feature dimension $d$ is smaller than the number of classes $K$, which is more in line with the application of deep network models in the practical tasks such as face recognition. We noticed that some scholars (Liu et al., 2023, Jiang et al., 2023) have analyzed and discussed the neural collapse in deep network models when d<K by using CE or designing other loss functions, and proposed the concept of generalized neural collapse (GNC). We are also analyzing the feature learning of BCE with GNC for the case of $d<K$, but we have not achieved any solid theoretical result. Despite that, we believe that in this situation, the classifier bias of BCE still plays a substantial role in its feature learning, but when BCE reaches its minimum, its classifier vectors may not form a standard equiangular tight frame.
>
> **Reference:**
>
> Weiyang Liu, Longhui Yu, Adrian Weller, Bernhard Scholkopf. Generalizing and decoupling neural collapse via hyperspherical uniformity gap. ICLR, 2023.
>
> Jiachen Jiang, Jinxin Zhou, Peng Wang, Qing Qu, Dustin Mixon, Chong You, Zhihui Zhu, Generalized neural collapse for a large number of classes. Arxiv: 2310.05351, 2023.

---

> ### Author Response · Authors · 2024-11-20
> **responses to the Questions**
>
> **[Q1: BCE's performance in complex tasks]**
>
> **R-Q1:** Thanks for the question. We believe that BCE can also perform better than CE on the more complex classification tasks, and we will definitely try to experimentally verify its performance on the more complex tasks. We here briefly describe the experimental results of using BCE by us or other scholars on other more complex tasks:
>
>   **a. SphereFace2 and UniFace in face recognition.** In the face recognition task, the number of classes $K$ is much larger than the feature dimension $d$. Both SphereFace2 (Wen et al., 2022) and UniFace (Zhou et al., 2023) adopted the normalized BCE loss integrated with a unified classifier bias, and they both achieved SOTA results on face recognition and face verification at the time of their publication.
>
>   **b. ImageNet-1k.** We trained ResNet50, ResNet101, and DenseNet161 on ImageNet-1k by using CE and BCE, respectively. For saving the training time, we did not train these three models from scratch. Instead, we applied the two losses to fine-tune the models that have been pretrained for 90 epochs with CE, and each fine-tuning runs 30 epochs. Table 1 presents the classification results on the validation set of ImageNet-1k. While $\mathcal A$ denotes the classification accuracy, $\mathcal A_{Uni}$ denotes the uniform accuracy. From the table, one can find that on the medium-scale ImageNet-1k, BCE still shows a consistent and clear advantage over CE.
>
> **Table 1.** Comparison of BCE and CE on ImageNet-1K
>
> |     |R50| R50 | R101 |R101| D161|D161 |
> | --- | --- | --- | --- | --- | --- | --- |
> |       | $\mathcal A$|$\mathcal A_{Uni}$|$\mathcal A$|$\mathcal A_{Uni}$|$\mathcal A$|$\mathcal A_{Uni}$|
> | CE | 76.74 | 34.48 | 78.47 | 38.85 | 78.58 | 43.56 |
> | BCE | **77.12** | **66.92** | **78.88** | **70.46** | **79.19** | **69.08** |
>
> **c. Long-tailed recognition (LTR).** In the paper, we compared BCE and CE based on balanced dataset and found the advantages of BCE over CE in feature learning. On the imbalanced dataset, we find that BCE also show clear advantages over CE. On the long-tailed dataset CIFAR100-LT with imbalanced factor (IF) of 10, 50, and 100, we trained a customized ResNet32 using CE and BCE, respectively. The recognition results on the balanced test set are shown in Table 2. As our analysis, BCE could suppress the imbalance effect by decoupling the $K$ imbalanced metrics using Sigmoid, while they are coupled together on the Softmax’s denominator in CE.
>
> **Table 2.** The top-1 accuracy of BCE and CE on CIFAR100-LT with three different IF.
>
> |IF | 10| 50 | 100 |
> | --- | --- | --- | --- |
> |CE	| 70.91|57.59|	51.48|
> |BCE| **71.54**|**58.49**|**52.88**|
>
> **Reference:**
>
> Yandong Wen, et al. Sphereface2: Binary classification is all you need for deep face recognition. ICLR 2022.
>
> Jiancan Zhou, et al. Uniface: Unified cross-entropy loss for deep face recognition. ICCV, 2023.
>
> ----------------------------------------------
> ----------------------------------------------
>
> **[Q2: Explicit vs. implicit constraints in feature learning.]**
>
> **R-Q2:** Thanks for the question. Though CE implicitly constrains the feature learning, it also ultimately leads to neural collapse, i.e., maximizing the intra-class compactness and inter-class distinctiveness of its final features. Therefore, theoretically, the performance upper boundary of CE and BCE is the same.
>
> We are not sure if the "feature learning strengths" mentioned by the reviewer can be understood as the convergence speed of BCE and CE in model training? According to our empirical results, BCE in the model training converges to neural collapse faster than CE, which indicates that BCE has a faster convergence speed. We have tried to prove it through theoretical analysis, but did not obtain solid results. During our theorical analysis, we define a naïve loss
> $$L_{naive}(z^{(k)}) = - (w_k^Th^{(k)} - b_k) + \frac{1}{K-1}\sum_{j=1,j\neq k}^K(w_j^Th^{(k)}-b_j)$$
> which could present a lower bound  for the CE loss,
> $$L_{naive}(z^{(k)}) \leq -\frac{K}{K-1}\log\frac{\exp(w_k^Th^{(k)}-b_k)}{\sum_{j=1}^K\exp(w_j^Th^{(k)}-b_j)} - \frac{K\log K}{K-1}=\frac{K}{K-1}L_{ce}(z^{(k)}) - \frac{K\log K}{K-1}.$$
> The above inequality can be proved by using the well-known inequality of arithmetic and geometric means.
>
> We hope that the reviewer could further provide information or literatures about feature learning strengths, which will be very helpful for our future research.

---

### Author Response · Authors · 2024-11-25
**Thanks to all the reviewers.**

We thank all the reviewers for their time and efforts in reviewing our paper and providing comments. We especially thank the reviewers **tKaJ** and **bpu4** for their positive comments on our paper, and deeply appreciate the reviewer **tKaJ** for the recognition of our paper's contributions. We also thank the reviewer **Eks9** for the comments on our paper.

We have replied point-by-point all the weaknesses and questions raised by the reviewers regarding the paper. We hope that our responses have adequately addressed these issues, and we look forward to further discussions with the reviewers. We are particularly eager to discuss further with reviewer **Eks9** to resolve our differing views on the Sigmoid loss and BCE loss.

---

### Meta-Review · Area_Chair_ozQG · 2024-12-23

**Metareview:**

The paper shows that deep neural networks trained with binary cross entropy (BCE) loss exhibit neural collapse by maximizing intra-class compactness and inter-class distinctiveness at the loss minimum. This phenomenon highlights BCE's effectiveness over categorical cross entropy (CE) in learning more compact and separated features, leading to improved classification and search/retrieval performance as observed in previous studies.

**Additional Comments On Reviewer Discussion:**

After the rebuttal, the submission received accept, borderline accept, and reject recommendations from the reviewers, which remained very mixed. The ACs have carefully considered this paper and all the discussions. We are aware that there is a confusing point about the BCE and sigmoid losses, which we did not consider a valid reason to reject the paper.

Considering all the reviews, rebuttals, and discussions, both the theoretical and experimental analyses are interesting. One major weakness is the experiments. This submission is theoretically convincing while experimentally still somewhat weak. It would be good to see if the same analysis and conclusions still hold for Transformer-based architectures, given that such models are gradually dominating more and more tasks. Meanwhile, the experiments on ImageNet-1k (during rebuttal) were conducted using pre-trained models with CE loss. Given the prevalence of strong self-supervised foundation models, such as DINOv2, it would be very helpful to show the results of leveraging such pre-trained models as well.

Therefore, we are afraid that the submission is not ready to be accepted in its current form. The authors are encouraged to further improve the paper for future submission, considering all the review comments.

---

### Decision · Program_Chairs · 2025-01-22

Reject